# Macrophage metabolic reprogramming during dietary stress influences adult body size in *Drosophila*

Anusree Mahanta [1,8 ✉], Sajad Ahmad Najar[1,2,8], Nivedita Hariharan[1], Ajit Bhowmick[1], Syed Iqra Rizvi[1], Manisha Goyal [1,3], Preethi Parupalli[1,4], Ramaswamy Subramanian [5,6], Angela Giangrande [7], Dasaradhi Palakodeti [1] & Tina Mukherjee [1 ✉]

## Abstract

Immune cells are increasingly recognized as nutrient sensors; however, their developmental role in regulating growth under homeostasis or dietary stress remains elusive. Here, we show that *Drosophila* larval macrophages, in response to excessive dietary sugar (HSD), reprogram their metabolic state by activating glycolysis, thereby enhancing TCA-cycle flux, and increasing lipogenesis —while concurrently maintaining a lipolytic state. Although this immune-metabolic configuration correlates with growth retardation under HSD, our genetic analyses reveal that enhanced lipogenesis supports growth, whereas glycolysis and lipolysis are growth-inhibitory. Notably, promoting immune-driven lipogenesis offsets early growth inhibition in imaginal discs caused by glycolytic and lipolytic immune-metabolic states. Our findings reveal a model of immune-metabolic imbalance, where growth-suppressive states (glycolysis, lipolysis) dominate over a growth-supportive lipogenic state, thereby impairing early organ size control and ultimately affecting adult size. Overall, this study provides important insights into dietary stress-induced immune-metabolic reprogramming and its link to organ size regulation and early developmental plasticity.

**Keywords** Macrophages; Glycolysis; Lipogenesis; Animal Growth; Dietary Stress
**Subject Categories** Development; Immunology; Metabolism

## Introduction

Body growth is a highly orchestrated process that ensures the formation of adults with correct size and proportions to finally influence survival and reproduction (Baron et al, 2015; Boulan et al, 2015; Nijhout et al, 2014). A complex integration of environmental and developmental cues governs the rate and duration of juvenile growth, which determines the final adult body size (Penzo-Méndez and Stanger, 2015). It is also essential that these growth mechanisms are plastic to allow adaptation of developing animals to environmental challenges like infection and fluctuations in nutrition. The evolutionary conservation of mammalian growth control pathways in fruit flies has facilitated numerous studies revealing intricate communication between organs for systemic growth regulation in homeostasis and under varying environmental conditions (Koyama et al, 2020). Inter-organ communication among *Drosophila* nutrient sensor and responder tissues—including the fat body, brain, imaginal discs, muscle, and gut—is pivotal in regulating organ and body growth. Hormones, cytokines, and morphogens serve as the signaling molecules orchestrating this crosstalk (Reviewed in Chatterjee and Perrimon, 2021; Droujinine and Perrimon, 2016; Boulan and Léopold, 2021). Understanding how these tissues sense environmental cues and adjust growth accordingly provides insights into the systemic growth control axis.

In this context, the functioning of the immune system with consequences on systemic growth is documented, where examples of immune modulation and its impact on animal sizes have been described. Heightened immunity correlates with stunted growth, while the opposite is true with animals with a weak immune system (van der Most et al, 2011). The importance of maintaining immune cell numbers to enable systemic growth has also been recently described (Bakopoulos et al, 2020; Cho et al, 2020; Ramond et al, 2020). These evidence have alluded to immune cell functioning and its trade-off with growth homeostasis. However, when it comes to the growth axis, immune cells are seldom mentioned. Perhaps because these examples of growth modulation are described in conditions of infection, we consider changes in animal growth as a consequence of altered immunity as opposed to their direct contribution to the larger scheme of developmental control of growth. The fact that immune cells are emerging as key nutrient sensors (Martínez-Micaelo et al, 2016; Newsholme, 2021), much like the fat body and brain, and implicated in developmental decisions (Juarez-Carreño and Geissmann, 2023), their role in growth homeostasis from a developmental standpoint does not seem unrealistic.

[1]Institute for Stem Cell Science and Regenerative Medicine (inStem), GKVK post, Bellary Road, Bangalore, Karnataka 560065, India. [2]The Shanmugha Arts, Science, Technology & Research Academy (SASTRA), Thanjavur, Tamil Nadu 613401, India. [3]The University of Trans Disciplinary Health Sciences & Technology (TDU), Bengaluru, Karnataka 560064, India. [4]Department of Biological Sciences, University of Texas, Dallas, TX, USA. [5]Department of Biological Sciences, Purdue University, West Lafayette, IN, USA. [6]Bindley Biosciences Centre, Purdue University, West Lafayette, IN, USA. [7]Institut de Génétique et de Biologie Moléculaire et Cellulaire, Strasbourg, France. [8]These authors contributed equally: Anusree Mahanta, Sajad Ahmad Najar. ✉E-mail: anusreem@instem.res.in; tinam@instem.res.in

It is now increasingly appreciated that macrophages in response to their surrounding environment undergo metabolic rewiring which in turn, determines their functional responses (Batista-Gonzalez et al, 2019; El Kasmi and Stenmark, 2015). The recent advances in high-throughput transcriptomics and metabolic analysis have aided a deeper understanding of macrophage heterogeneity, revealing distinct phenotypes that rely on metabolic pathways involving lactate (Geeraerts et al, 2021), purine (Li et al, 2022), and arginine (Viola et al, 2019). This is in addition to the already established M1 and M2 macrophage types employing aerobic glycolysis and fatty acid oxidation, respectively (Galván-Peña and O'Neill, 2014). Whether these macrophage functional types are different subsets or one subset with potential for plasticity remains to be understood (Remmerie and Scott, 2018). Nonetheless, the link between the metabolic heterogeneity of macrophages and their functions has been widely implicated in both health and disease. Recent studies have in fact also shown *Drosophila* macrophage-like plasmatocytes to be highly heterogeneous with regard to adopting comparable metabolic remodeling (Cattenoz et al, 2020; Cho et al, 2020; Coates et al, 2021; Girard et al, 2021). In fact, a study has reported enhanced glucose uptake accompanied by aerobic glycolysis in *Drosophila* macrophages post bacterial infection, similar to mammalian M1 type of macrophages (Krejčová et al, 2019). In addition, perturbation of *Drosophila* macrophage oxidative phosphorylation, a metabolic signature of M2 macrophages, resulted in increased immunocompetence against wasp infection (Vesala et al, 2024).

To that end, our work from the recent past has implicated *Drosophila* larval immune cells as regulators of animal growth (P et al, 2020). *Drosophila* blood cells, called hemocytes, akin to vertebrate myeloid cells (Evans et al, 2003), contributed significantly towards coordinating growth in conditions of dietary sugar stress. Growth retarding effects of excessive dietary sugar (high sugar diet, HSD) have been observed across species from flies to mammals, and the foremost underlying reason implicated in this pathological outcome is the development of insulin resistance or inhibition of growth hormone signaling (Mitchell, 2017; Giannini et al, 2014). We however found that animals with depleted number of immune cells grew poorly in conditions of dietary sugar stress. Intriguingly, animals with more active immune cells developed unexpectedly well on HSD and were comparable to flies on a regular diet. These findings highlighted immune cells as key modifiers of growth homeostasis in stress conditions. The work proposed immune cell state changes as a key paradigm for growth adaptation in stress conditions (P et al, 2020). Thus, immune control of animal growth both in homeostasis and in stress conditions which remains a poorly understood area, led us to take on board the current investigation. The immune underpinnings of systemic growth homeostasis, specifically with respect to growth retardation evident in high sugar intake forms the central focus of our investigation.

To investigate how immune cells respond to a high-sugar diet (HSD) and how their metabolic state influences organismal growth, we conducted time-resolved dietary stress experiments in *Drosophila* larvae. Using a multipronged, unbiased approach, this study addresses this central question. A key finding is that immune metabolic rewiring induced by high sugar intake impacts early organ growth, which in turn affects overall organismal growth homeostasis. These results lead us to propose a deterministic role

for immune cells in regulating organ size, ultimately influencing adult body size. Our findings place the cellular immune system at the core of the growth control paradigm, as active participants rather than passive bystanders in stress-induced physiological responses.

# Results

## Dietary sugar overload impacts immune cell physiology and function

The central question of the study is to discern intracellular immune cell states governing body size control in a high sugar diet (HSD containing 25% sucrose whereas regular diet (RF) contains 5% sucrose only) induced stress. Therefore, we first characterized the status of immune cells themselves, namely, cell numbers, basal metabolic state, morphology, and function when exposed to HSD. To accomplish this, we utilized fly line $Hml^\Delta>GFP$ ($Hml^\Delta>GFP$ crossed to $w^{1118}$) in which Hml (Hemolectin) marks the differentiating immune cells. Next, we exposed the animals to two different dietary regimes: one, short-term exposure to HSD for four hours (referred as 4 hr.HSD, henceforth) as the means to gauge immediate changes induced in immune cells by short-term intake of high sugar and second, a long-term, constitutive HSD feeding (referred as Ct.HSD, henceforth) to identify cell states established as a consequence of sustained high sugar intake by the animal (for details, see "Methods" section).

To assess HSD-induced changes in immune cell numbers, we specifically monitored Hemolectin-positive (Hml⁺) and Hemolectin-negative (Hml⁻) cell populations across circulatory and sessile pools (see "Methods" for further details on their assessment). For metabolic changes, we characterized immune cells for their intracellular redox state, glucose, and lipid levels. For functional characterization, phagocytic bead uptake ability was measured (Hao et al, 2018) and finally for morphological changes, phalloidin staining was undertaken to assess changes in cell morphology, size, shape, and length of immune cell filopodia extensions.

We observed that high sugar treatment severely impacted larval immune cell numbers in the long-term, Ct.HSD condition (Fig. 1A–D). We observed that while short-term, 4 hr.HSD treatment of larvae did not reveal any changes in immune cell numbers, Ct.HSD animals showed a significant decline in total immune cell numbers (Fig. 1C,C'). Specifically, a significant decline in sessile Hml+ cell population was apparent (Fig. EV1A–A'''), while circulating cell numbers remained comparable across RF, 4 hr.HSD and Ct.HSD (Fig. EV1B,B'''). The number of Hml− cells however remained comparable between RF, 4 hr.HSD and Ct.HSD condition (Figs. 1D and EV1A''',B''), which implied a specific sensitivity of the Hml+ cell population to high sugar diet exposure.

Next, we compared the metabolic states and assessed for reactive oxygen species (ROS) levels by dihydroethidium (DHE) staining and observed non-significant change with high sugar exposure in 4 hr.HSD and Ct.HSD immune cells (Figs. 1E–E'', K and EV1C–C''). Biochemical means to estimate glucose revealed a significant rise in immune cell glucose levels following the short-term 4 hr.HSD exposure, which remains comparable in the long-term Ct.HSD regime (Fig. 1F). This implied that immune cell

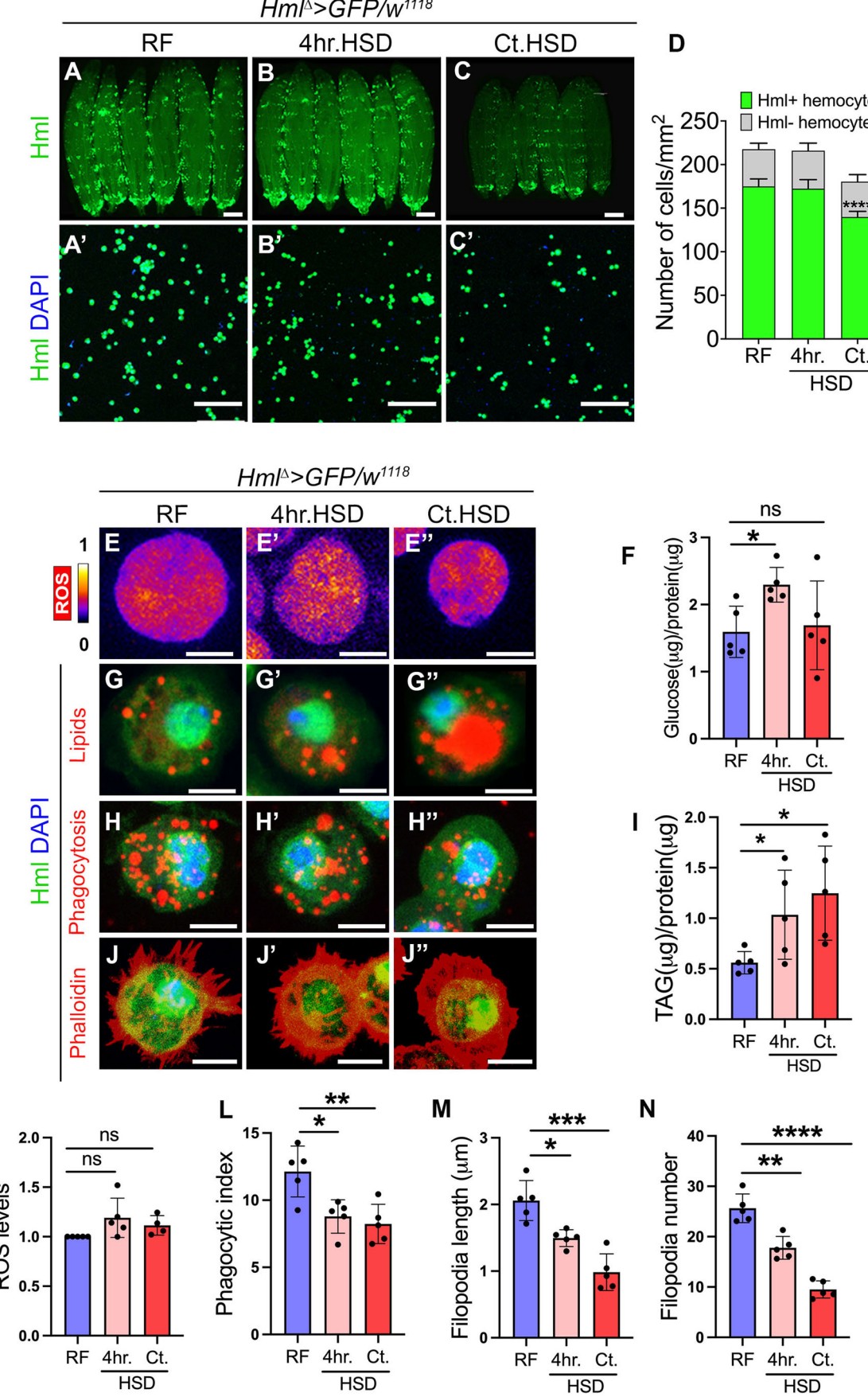

**Figure 1. Dietary sugar stress affects larval macrophage physiology.**

(A–D) High sugar diet affects macrophage number. (A–C') Representative images of third-instar larvae and macrophages under RF, 4 hr.HSD, and Ct.HSD. Macrophage numbers are unchanged at 4 hr.HSD (B, B') but reduced at Ct.HSD (C, C') compared to RF (A, A'). (D) Quantification of Hml-positive cell numbers: RF ($N = 3$, $n = 18$), 4 hr.HSD ($N = 3$, $n = 18$, $P = 0.9949$), Ct.HSD ($N = 3$, $n = 18$, $P < 0.0001$). Quantification of Hml-negative cell numbers: RF ($N = 3$, $n = 18$), 4 hr.HSD ($N = 3$, $n = 18$, $P = 0.9999$), Ct.HSD ($N = 3$, $n = 18$, $P = 0.9974$). (E–E") ROS levels in macrophages remain unchanged across dietary conditions. (E) Regular food (RF); (E') 4 h. high-sugar diet (4 hr.HSD); (E") chronic high-sugar diet (Ct.HSD). These panels are higher magnification views taken from the corresponding lower magnification images shown in Fig. EV1C–C". Regions displayed here are marked with white boxes in Fig. EV1C–C". (F) Intracellular glucose levels increase at 4 hr.HSD and remains unchanged at Ct.HSD compared to RF. RF ($N = 5$, $n = 150$), 4 hr.HSD ($N = 5$, $n = 150$, $P = 0.0114$), Ct.HSD ($N = 5$, $n = 150$, $P = 0.7878$). (G–G") Lipid content in macrophages increases under 4 hr.HSD and Ct.HSD compared to RF. Nile Red staining shows gradual accumulation (G–G"). Ct.HSD panel is higher magnification view taken from the corresponding lower magnification image shown in Fig. EV1D". Regions displayed here are marked with white boxes in Fig. EV1D". (H–H") Phagocytic activity decreases under sugar stress. Bead uptake is reduced at 4 hr.HSD and Ct.HSD vs. RF. (I) Quantification of triacylglycerol: RF ($N = 5$, $n = 165$), 4 hr.HSD ($N = 5$, $n = 165$, $P = 0.0476$), Ct.HSD ($N = 5$, $n = 165$, $P = 0.0124$). Triglyceride levels increase at 4 hr.HSD and Ct.HSD vs. RF. (J–J") Filopodia length and number decrease under a sugar diet. RF vs. 4 hr.HSD and Ct.HSD. These panels are higher magnification views taken from the corresponding lower magnification images shown in Fig. EV1F–F". Regions displayed here are marked with white boxes in Fig. EV1F–F". (K) Quantification of ROS: RF ($N = 4$, $n = 30$), 4 hr.HSD ($N = 5$, $n = 30$, $P = 0.0995$), Ct.HSD ($N = 4$, $n = 30$, $P = 0.0587$). No significant change both at 4 hr.HSD and Ct.HSD. (L) Quantification of bead uptake: RF ($N = 5$, $n = 50$), 4 hr.HSD ($N = 5$, $n = 50$, $P = 0.0108$), Ct.HSD ($N = 5$, $n = 50$, $P = 0.0065$). Bead uptake decreases at 4 hr.HSD and Ct.HSD vs. RF. (M) Quantification of filopodia length: RF ($N = 5$, $n = 50$); 4 hr.HSD ($N = 5$, $n = 50$, $P = 0.0102$); Ct.HSD ($N = 5$, $n = 50$, $P = 0.0004$). Filopodia length decreases at 4 hr.HSD and Ct.HSD compared to RF. (N) Quantification of filopodia number: RF ($N = 5$, $n = 50$); 4 hr.HSD ($N = 5$, $n = 50$, $P = 0.0013$); Ct.HSD ($N = 5$, $n = 50$, $P < 0.0001$). Filopodia number decreases at 4 hr.HSD and Ct.HSD compared to RF. Data information: DNA is labeled with DAPI (blue); macrophages are marked by GFP (green; $Hml^\Delta$>UAS-GFP/$w^{1118}$). Reactive oxygen species (ROS) shown in spectral mode; E–E"). (A–C) Scale bar: 0.5 mm; (A'–C'), scale bar: 100 μm; (E–J"), scale bar: 5 μm. In bar graphs, data are presented as mean ± SD. In graphs (F, I, K–N), each dot represents one experimental repeat. $N$ indicates the number of independent biological replicates, and $n$ refers to the total number of larvae analyzed. Statistical comparisons were performed against regular food (RF); asterisks mark statistically significant differences (*$P < 0.05$; **$P < 0.01$; ***$P < 0.001$; ****$P < 0.0001$). Two-way ANOVA with Sidak's multiple comparisons test was used for (D); unpaired $t$ test for (F, I, K–N). RF, 4 hr.HSD, and Ct.HSD indicate larvae fed regular food, 4 h high sugar diet and constitutive high sugar diet, respectively. Source data are available online for this figure.

glucose levels increased immediately to high sugar exposure, but gradually plateaued in the long-term HSD.

We observed that high sugar diet also resulted in an overall increase in lipid levels inside immune cells, much more evident in the long-term Ct.HSD than 4 hr.HSD condition (Figs. 1G–G",I and EV1D–D",G–G"). For lipid measurements, we employed nile red staining to mark lipid droplets, TAG (triacylglycerol) biochemical measurements to assess total TAG and UAS-LSD2-GFP genetic reporter line to assess their lipogenic state (Fauny et al, 2005). Specifically, we observed a gradual increase in the number of lipid droplets (Figs. 1G–G" and EV1D–D") and total TAG level in immune cells from 4 hr.HSD and Ct.HSD when compared to RF condition (Fig. 1I). These signatures of increasing levels of TAG in the cells corroborated with their lipogenic potential as seen with increasing LSD2-GFP reporter expression (Fig. EV1G–G"). The data highlighted the sensitivity of immune cells to HSD, and the overall impact on their internal lipid homeostasis when faced with dietary sugar stress. Altogether, the increased glucose levels following high sugar diet exposure and the gradual increase in lipid levels were suggestive of induction of metabolic programs to accommodate excessive sugar as reported for the fat body (Musselman et al, 2011).

Functionally, high sugar exposure severely impaired immune cell phagocytic abilities. A gradual decrease in the number of internalized beads was evident in the immune cells. This was seen as early as in 4 hr.HSD treatment and dramatically reduced in Ct.HSD condition (Figs. 1H–H",L and EV1E–E"). Morphologically, compared to numerous filopodia seen protruding from the immune cell surface from RF larvae (Figs. 1J and EV1F), a reduction in the number and length of filopodia was evident even at 4 hr.HSD exposure which was further pronounced in immune cells when subjected to Ct.HSD condition (Figs. 1J–J",M,N and EV1F–F"). Also, this dietary regime did not lead to any aberrant formation of lamellocytes, which are discriminated as large

flattened shaped cells against other immune cells (Fig. EV1F–F") which are much smaller in size (Madhwal et al, 2020).

The overall temporal profiling of immune cell number, cytoskeleton dynamics, phagocytosis and metabolism revealed manifestation in metabolic and functional capabilities in immune cells as early as 4 hr.HSD feeding. A clear decline in immune cell phagocytic ability with increased lipogenesis was evident, and these changes were exaggerated with longer exposure to dietary sugar. The relevance of such sugar-induced immune cell state changes on adult body size was investigated next. To do this, we undertook a genetic screening approach as a means to identify specific candidates whose function in immune cells affected animal growth on Ct.HSD.

## Immune-specific screen identifies key growth regulators

For the genetic screen, males from a comprehensive RNAi library (>1000) maintained at the National Center for Biological Sciences (NCBS) Fly Stock Center (Bangalore) were crossed to females carrying the $Hml^\Delta$-GAL4,UAS-GFP transgenes to achieve macrophage-specific RNAi-mediated knockdown. This RNAi collection is a repository of VDRC (Vienna Drosophila Resource Center) and BDSC (Bloomington Drosophila Stock Center) lines curated from multiple Indian research laboratories, not only at NCBS but across other national laboratories in India. Thus, the collection is a diverse set of RNAi lines which was available to us and therefore used for systematically probing gene function in the context of basal immune homeostasis.

The crosses were set up on regular food and 35–40 embryos from the progeny/F1 generation were collected, transferred to a high sugar diet (Ct.HSD) and reared at 29 °C until adult flies eclosed, whose sizes were thereafter scored (refer to "Methods" for details). In each experimental set, the RNAi crosses were tested in two batches as biological replicates for enhanced accuracy. For

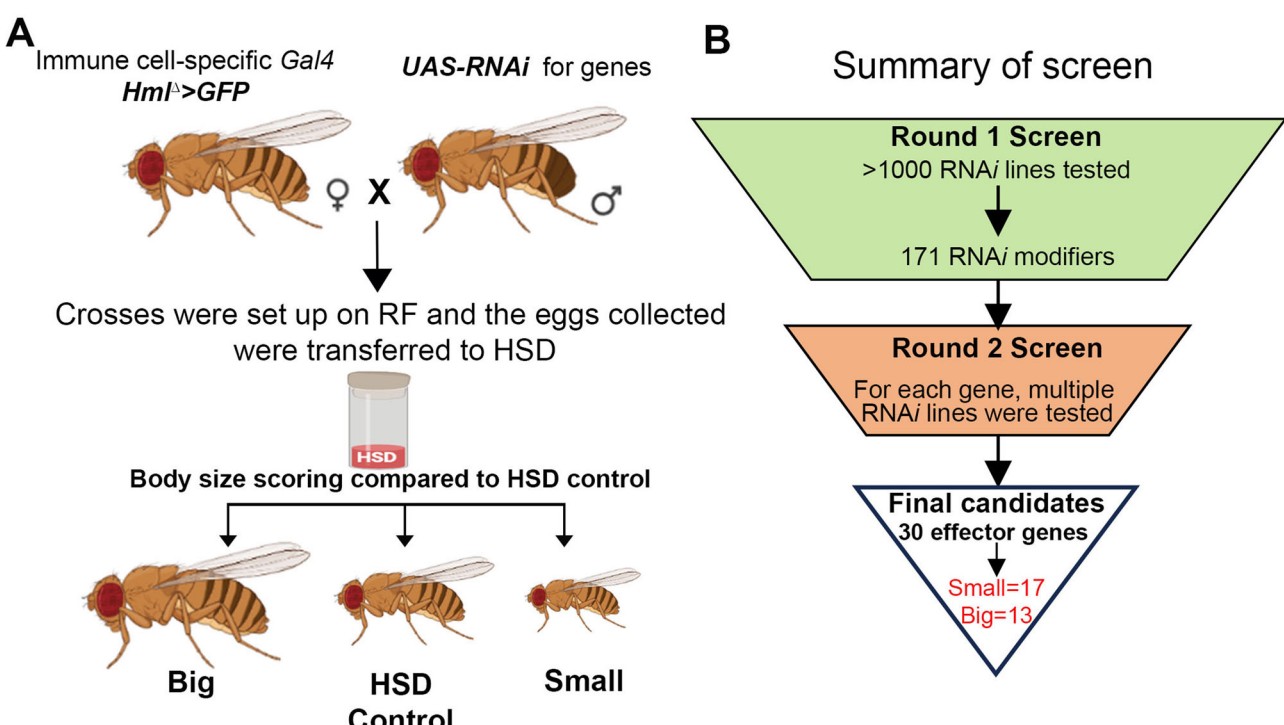

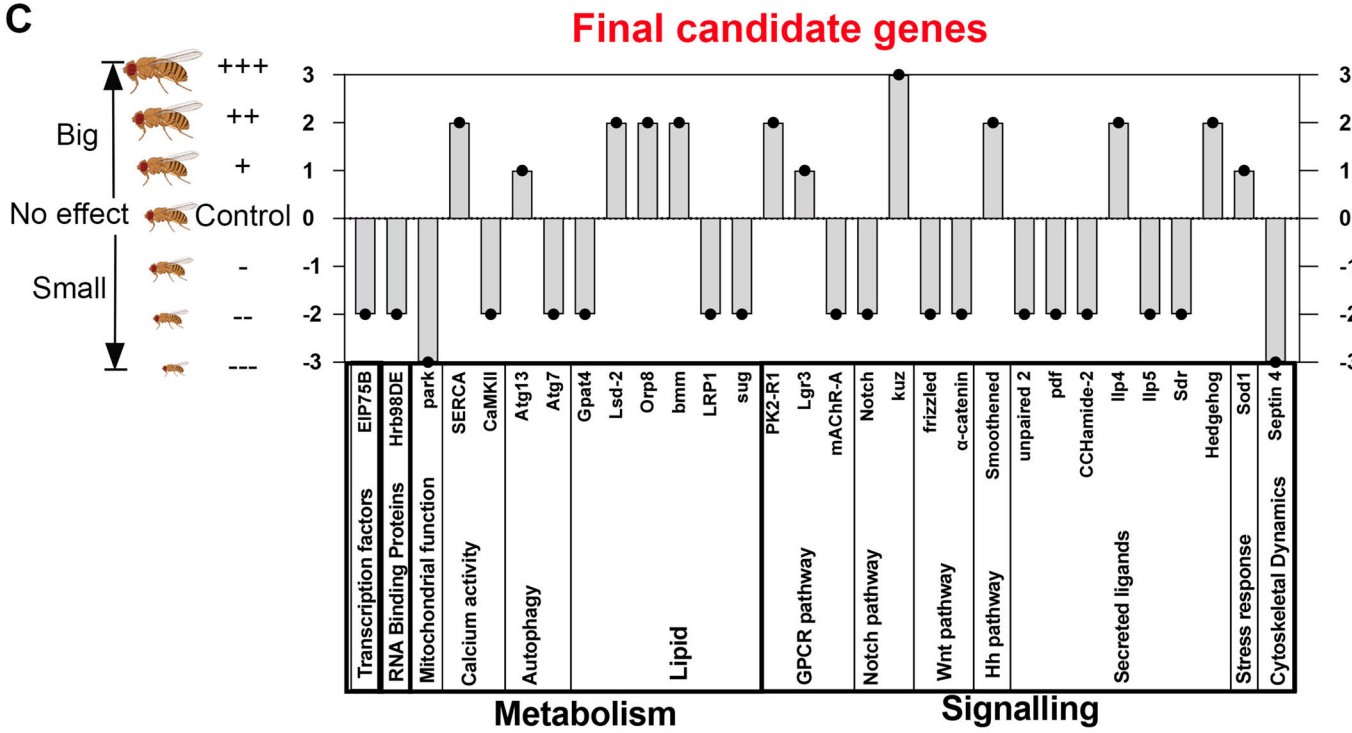

comparison to assess changes in adult sizes, $Hml^\Delta>UAS\text{-}GFP/w^{1118}$ adults were grown on RF and on Ct.HSD conditions were used as controls that demonstrated normal adult fly size and growth retarded HSD flies, respectively (Fig. 2A).

For scoring, 1-day-old adult flies obtained post eclosion were phenotypically screened and scored for their body size. To increase the robustness of the screening process, the size scoring was performed independently by three different individuals in a blind

Figure 2.    Identification of immune-specific modulators of animal growth in dietary sugar stress condition.

(A) Schematic representation of in vivo *RNAi* screen. Females of the immune cell-specific Gal4 driver (*Hml^Δ>GFP*) line were crossed to *UAS-RNAi* males. Eggs collected on regular food were transferred to a high sugar diet (HSD). Eclosed flies were scored for body size phenotype. (B) Summary of the results from in vivo *RNAi* screen. In the first round, >1000 *RNAi* lines were tested. Modifiers obtained from the first round were further tested in the second round with multiple *RNAi* lines. Finally, we arrived at 30 effector genes with 17 genes being positive regulators of growth and 13 as negative regulators (see Appendix Table S1 for details on the lines tested). (C) Summary of the final effector/candidate genes obtained from the screen. The genes were categorized based on their biological functions. "Big" and "Small" phenotypes, were graded into mild (+) or (−), moderate (++) or (−−) or severe (+++) or (−−−) categories. Source data are available online for this figure.

manner. As an initial body size scoring paradigm, the adults from the *RNAi* crosses were scored either for any further reduction in their size than seen in HSD flies and were marked "Small", or any recovery in their size if they appeared any closer to size seen in regular food-raised controls and were marked "Big", respectively (Fig. 2A).

Based on this assessment criteria, in the first round, a total of 171 *RNAi* lines were identified as "modifiers" of adult size on HSD condition. Of these, 101 lines showed size reduction and were smaller than HSD control adults, implying that these lines are positive regulators of growth on HSD conditions. Very interestingly, 18 *RNAi* strains restored the adult size defect seen in HSD. The emerging adults from these *RNAi* crosses, were larger in size than observed in HSD controls and were rather closer in their size to RF controls. These "Big" genes were designated as negative regulators of growth. The remaining 797 *RNAi* lines did not demonstrate any deviation from HSD controls and were recorded as "no effect" (NE) and were listed as non-modifiers (Fig. 2B).

For the total 171 modifier lines identified from Round 1, we next undertook a second round of screening. Here, we tested multiple *RNAi* lines against each candidate and only those candidates, where we observed consistent growth phenotypes across two or more *RNAi* lines, they were finally selected. The majority of the *RNAi* lines chosen here are published lines validated for their function. Depending on the extent of size modulation observed, they were further graded. For "Big" and "Small" phenotypes, they were also graded into mild (+) or (−), moderate (++) or (−−) or severe (+++) or (−−−) categories (Fig. 2C; Appendix Table S1).

We found a total of 30 genes that showed consistent phenotype with more than one *RNAi* line and were listed as "final candidate genes" (Fig. 2C; Appendix Table S1). Of these 17 were identified as "Small" and 13 were identified as "Big", and the majority of these lines were moderate modifiers of the growth phenotype, while only a few were mild effectors. This finding implied a robust contribution by the immune cells on growth in HSD condition. Subsequently, FlyBase was used to determine the known or predicted functions of these genes (Fig. 2C; Appendix Table S1). When functionally categorized, these top 30 candidate genes came under the categories that included diverse cellular functions ranging from transcription factors to metabolic and signaling genes (Fig. 2C).

As the major cohort important for animal growth, the signaling genes seemed expected; however, we observed an unexpected influence on growth in this category. Signaling pathway components of the Notch, Wnt, and JAK/STAT pathways were identified as necessary for growth, as their downregulation in immune cells caused a retardation in adult fly size compared to HSD controls. Hedgehog (hh) signaling, contrarily, was identified as a negative modulator of growth. Interestingly, both hh and its receptor, smoothened (Alcedo et al, 1996) appeared in the screen and

blocking their expression in blood cells, resulted in growth recovery. This finding was indeed surprising as it implied both hh and smo (smoothened) operated in Hml⁺ blood cells to control growth. While this could argue for an autocrine mode of functioning as described in tracheal progenitor cell (Yin et al, 2022), it could also arise as a consequence of heterogeneity (Cattenoz et al, 2020) where a subpopulation of immune cells are responsive to hh ligand and controlled growth on HSD. Nevertheless, the data implied that in HSD condition, immune cell signaling exerted dual control on growth, where players like Notch (N), upd2 (unpaired 2), and Wnt enabled growth, but hh and it's signaling inhibited growth.

The other biological process that was overrepresented was the "metabolism category". Under this, majority encoded functions related to "lipid metabolism". Specifically, lipogenic genes like *Glycerol 3-phosphate acyltransferase 4 (Gpat4)*, involved in triglyceride synthesis (Heier and Kühnlein, 2018) and transcription factors, *sugarbabe (sug), Oxysterol receptor protein 8 (Orp8)* (Kokki et al, 2021; Mattila et al, 2015; Repa et al, 2000), known to promote lipogenic expression on high sugar diet were identified as positive growth regulators. Importantly, the screen also identified *brummer (bmm)*, a key lipolytic gene (Grönke et al, 2005), whose downregulation, showed growth recovery with adult flies much larger in size than HSD control. These data revealed a significant role for immune lipid levels on systemic growth and implied growth-promoting functions for immune lipogenesis but growth inhibitory consequences for immune cell lipid turnover (Fig. 2C). Overall, the screen revealed unexpected and opposing functional states within the Hml⁺ immune cell population—some promoting growth, such as Notch, upd2, and Gpat4, while others, including hh, smo, and bmm, acted as growth suppressors.

In addition to these, other notable candidates emerged that included, regulators of mitochondrial metabolism (*parkin*), autophagy (*Atg13* and *Atg7*), and cytoskeletal remodeling (*Septin 4*), all of which functioned as positive growth modulators. Collectively, these findings underscore the significant role of immune cell intrinsic states in the systemic coordination of animal growth. Particularly, the identification of genes involved in immune cell lipid metabolism and mitochondrial function pointed to a broader theme of immune-metabolic regulation in maintaining growth homeostasis under dietary stress. This central observation guided our subsequent efforts to more deeply characterize the immune metabolic states governing growth on a high sugar diet (HSD).

## Transcriptional profiling highlights immune cell-specific metabolic rewiring induced by HSD

To gain a comprehensive understanding of the metabolic state changes induced by the high sugar diet (HSD), we performed genome-wide transcriptomic profiling via RNA sequencing of

immune cells (Appendix Fig. S1A). In parallel, we analyzed the whole larval transcriptome to distinguish global effects of HSD from immune cell-specific responses. Immune cells were isolated from larvae subjected to 4 hr.HSD or control HSD (Ct.HSD) feeding, and total RNA was extracted for bulk sequencing. RF immune cells served as the reference control (Appendix Fig. S1A). Whole larval RNA sequencing under the same conditions was also performed to provide a comparative framework (Appendix Fig. S1A).

The biological processes influenced by 4 hr.HSD and long-term Ct.HSD in blood cells, using Gene Ontology (GO) analysis highlighted immediate transcriptional changes that remained persistent even in the constitutive HSD condition (Appendix Table S2 and Appendix Table S3). These included downregulation of genes encoding JAK STAT signaling pathway, Toll/Imd, ecdysone signaling, and Wnt signaling pathway. Along with these, genes involved in cell migration, cell matrix adhesion, and integrin signaling were also seen downregulated upon HSD treatment. The transcriptional changes converged with morphological analysis shown in Fig. 1 and implied that high sugar transcriptionally dampened their immune potential and cytoskeletal remodeling proteins. We also assessed the expression of some of the screen candidates, like upd2 (JAK/STAT Pathway), Fz, alpha-catenin (Wnt signaling pathway), but did not observe any transcriptional alteration. Nevertheless, the overall transcriptional downregulation of the aforementioned signaling pathways highlighted their sensitivity to excessive sugar and the associated implication on growth identified in the screen (Fig. 2C), corroborated with their functional requirement.

Analysis of upregulated pathways in blood cells revealed that metabolic processes were among the most overrepresented biological categories (Fig. 3; Appendix Table S2 and Appendix Table S3). Notably, lipid metabolism emerged as a key altered process, with several lipogenic genes showing significant upregulation (Fig. 3A). This included *Acetyl CoA carboxylase (ACC)*—the rate-limiting enzyme in de novo fatty acid synthesis (Parvy et al, 2012)—as well as all major enzymatic components of the triacylglycerol (TAG) synthesis pathway, namely *GPAT, AGPAT*, and *Lipin* (Heier and Kühnlein, 2018) (Fig. 3; Appendix Tables S2 and S3). ACC upregulation was specific to the 4 hr.HSD condition (Fig. 3B), while genes like *Gpdh1* and *Lipin* remained elevated even under chronic HSD exposure (Fig. 3B; Appendix Table S3).

Interestingly, the GO term "fatty acid biosynthetic process" was enriched only in the chronic HSD condition (Ct.HSD), driven by upregulation of elongase genes such as *CG8534, CG9459, CG30008*, and *CG33100* (Appendix Table S3). This suggests an adaptation in fatty acid chain length in immune cells following prolonged sugar exposure. Although immune cells also upregulated *Acyl CoA synthetase (Acsl)*, a key enzyme in β-oxidation, lipolytic genes were not broadly represented, supporting cell biological observations of increased lipid droplet accumulation under HSD. These findings point to a transcriptionally driven metabolic reprogramming favoring lipid biosynthesis.

Further, enzymes involved in pyruvate cycling—such as *Malic enzyme b (Men b)* and *Mitochondrial pyruvate carrier (Mpc1)*—were also upregulated (Appendix Table S2), indicating enhanced pyruvate flux. Consistently, genes associated with the TCA cycle were elevated in HSD immune cells (Fig. 3A,B), including *midline uncoordinated (muc)*, which has pyruvate dehydrogenase (PDH)-

like activity (FlyBase Reference Report: Marygold, 2024.1.15), *2 oxoacid dehydrogenase complexes, Citrate synthase 1 (Cs1)* and *Succinate dehydrogenase (Sdh)*. While *muc* expression was increased at 4 hr.HSD but not at Ct.HSD, *Cs1* and *Sdh* remained upregulated throughout, suggesting sustained TCA activity.

In addition, pathways linked to glutathione metabolism, cell division (including spindle assembly and cytokinesis) and actin cytoskeleton reorganization were significantly enriched (Appendix Tables S2 and S3), reinforcing the impact of sugar stress on immune cell proliferation and cytoskeletal dynamics (Fig. 1D). These data collectively emphasize that immune cells undergo transcriptional metabolic reprogramming in response to high dietary sugar, with early activation of TCA and lipogenic pathways evident as soon as 4 hr.HSD post exposure.

Importantly, this transcriptional shift was highly specific to immune cells (Appendix Table S4). Genes such as *ACC, GPAT1, AGPAT*, and *Lipin* were upregulated in blood tissue (Fig. 3B) but not in the whole animal (Fig. EV2A,B). In contrast, whole animal transcriptomics revealed an overall downregulation of metabolic genes, especially under chronic HSD (Fig. EV2B), along with an upregulation of developmental programs (Appendix Table S4). This contrast highlights a tissue-specific response where immune cells uniquely rewire their metabolism to handle dietary sugar stress —an adaptation not mirrored at the systemic level.

## Conflicting immune metabolic states in HSD: catabolic immune cell states- restricts, while lipogenic immune metabolic reprogramming, promotes growth

To further interrogate the immune cell intrinsic metabolic states relevant to systemic growth modulation under high sugar diet (HSD), we conducted a focused set of investigations based on the distinct immune metabolic signatures uncovered in our transcriptomic analysis. These signatures included elevated TCA cycle activity, enhanced pyruvate metabolism, and upregulation of lipid metabolic genes. While these pathways are likely engaged to mitigate excess dietary sugar, we next sought to determine whether their activity directly influenced systemic growth. To address this, we employed both metabolic and genetic approaches, systematically dissecting the contribution of each pathway to organismal growth regulation under HSD conditions.

As the first step, we addressed TCA and performed liquid chromatography tandem mass spectrometry (LC-MS/MS). Subsequent to this, it was followed by genetic approaches to modulate corresponding TCA genes and assess the impact of animal growth on HSD condition. At this stage of our analysis, we conducted a rather quantitative approach to score animal growth. We chose wing area and fly body length (Lee et al, 2004, 2008) as a proxy for estimating the extent of changes brought by corresponding genetic manipulations on animal growth.

Metabolic flux analysis with isotopic U13C pyruvate was performed (Buescher et al, 2015; Jang et al, 2018) to discern any changes in the rate of TCA activity. Specifically, immune cells from regular food (RF) and constitutive high sugar diet (Ct.HSD) conditions were incubated with U13C pyruvate, and the flow of C13 into TCA cycle intermediates was assessed (Fig. 4A). Pyruvate enters the TCA cycle via pyruvate dehydrogenase (PDH), where it is converted into acetyl CoA, and this contributes to two carbons into the TCA metabolites. Pyruvate incorporates three carbons in

**A**

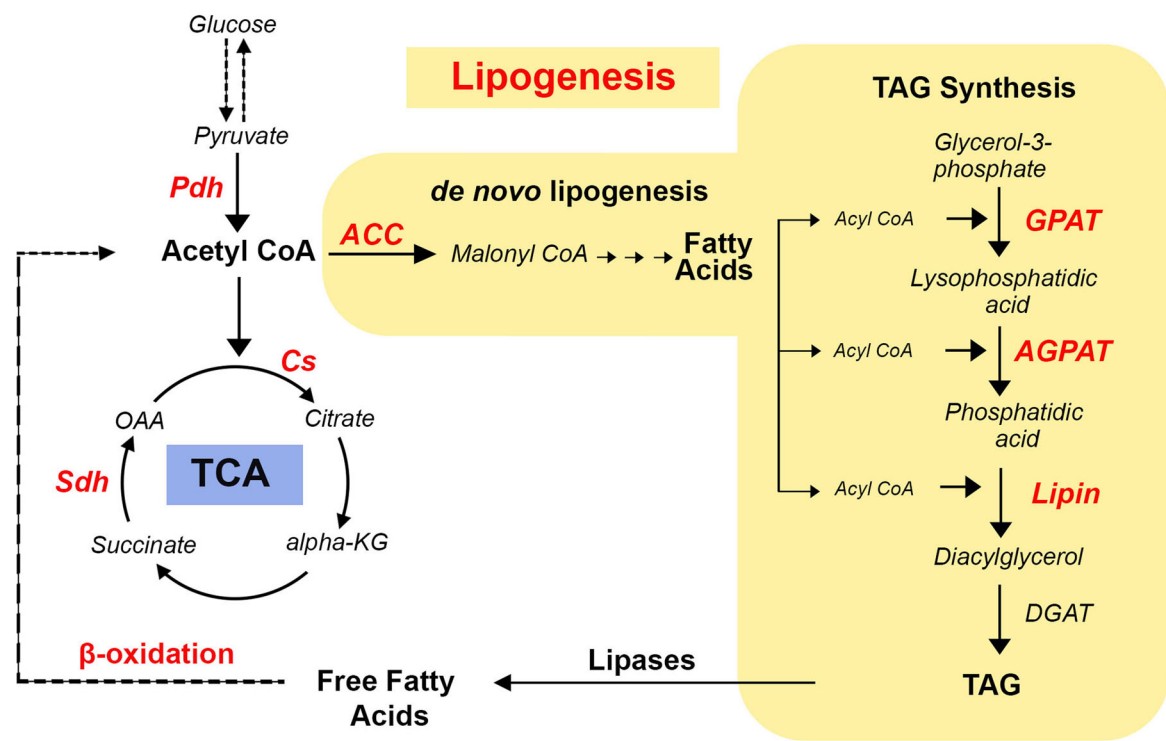

**B**

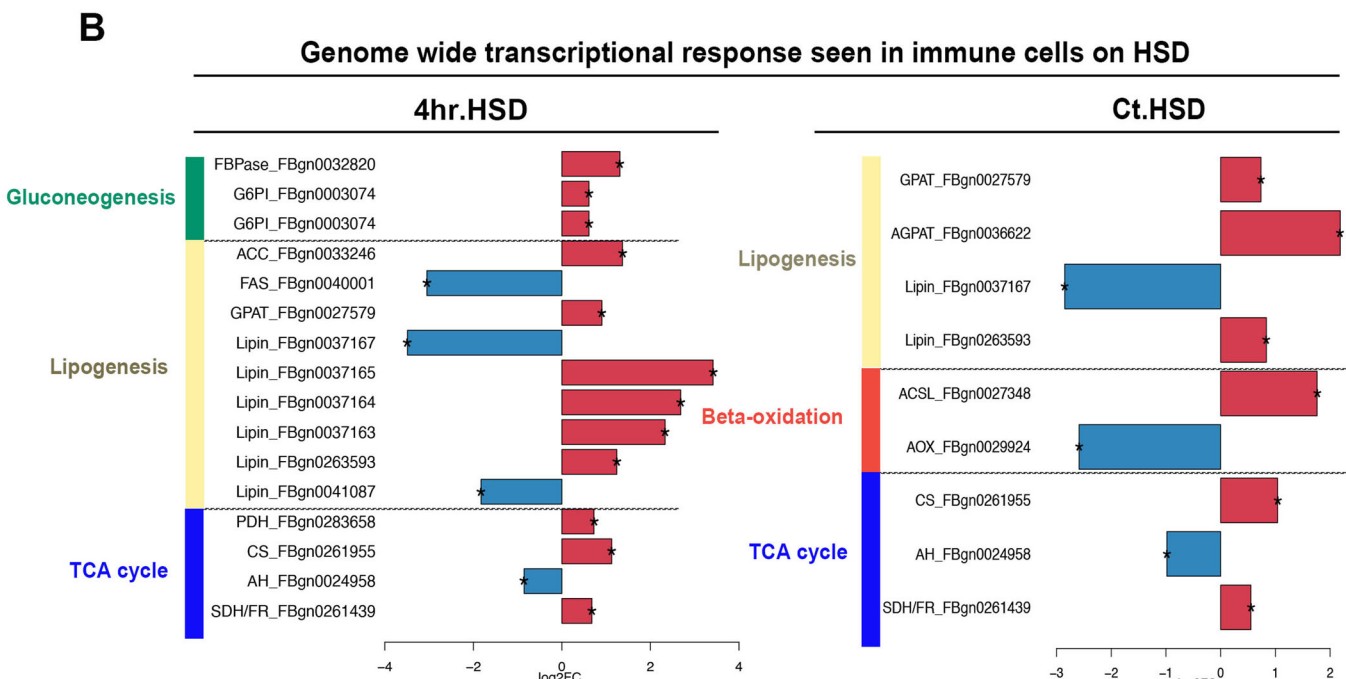

◄  **Figure 3.  Dietary sugar stress induces metabolic rewiring in immune cells.**

(A) Diagrammatic representation of overall transcriptional changes seen in metabolic genes in immune cells on HSD with short-term (4 hr.HSD) and long-term (Ct.HSD) exposure. All genes shown in red indicate their transcriptional upregulation, which includes TCA enzymes, de novo lipogenesis, TAG synthesis pathway, and beta oxidation enzymes. (B) Bar plots show temporal changes in respective metabolic genes and their paralogs. Red bars indicate upregulated genes, and blue bars indicate downregulated genes. ACC, which is a key de novo lipogenic enzyme, is seen upregulated immediately upon HSD exposure, but not with long-term exposure, when only TAG synthesis is seen upregulated. beta oxidation enzyme, acyl CoA synthetase long chain (Acsl) is however seen upregulated in constitutive HSD condition, but not in 4 hr.HSD. Source data are available online for this figure.

oxaloacetate (OAA) via pyruvate carboxylase (PC), which further adds on to citrate and thus contributes to the TCA cycle. Pyruvate is also converted into lactate via lactate dehydrogenase (LDH) and contributes to all three carbons of lactate (Fig. 4A). Thus, the differential labeling of carbons in TCA metabolites and lactate was considered as a measure of change in pyruvate flux under RF and Ct.HSD condition. Apart from metabolic flux analysis, we also conducted steady-state targeted comparative analysis of TCA cycle metabolites. For this, immune cells were isolated from animals raised on RF and Ct.HSD exposure and processed for steady-state metabolite analysis.

The levels of TCA cycle metabolites between RF and Ct.HSD overall failed to show any difference in steady state conditions (Appendix Fig. S2A). However, the isotopic metabolite measurements revealed increased flux of pyruvate into TCA metabolites and also into lactate under HSD conditions (Fig. 4A'; Appendix Fig. S2B). Specifically, our isotopic labeling data showed increased, higher 13 C label incorporation in citrate upon HSD, which indicates the increased pyruvate flux towards TCA cycle (Fig. 4A'; Appendix Fig. S2B). Moreover, malate also showed an increase in M + 2 label incorporation in the HSD condition which is donated by PDH-mediated entry of pyruvate into the TCA (Fig. 4A'; Appendix Fig. S2B). These data showed that immune cells upon high sugar exposure are more oxidative than on a regular diet. The rate of PC metabolism in HSD condition was however reduced as decrease in M + 3 labeling in OAA was seen and could be attributed to the corresponding rise in PDH activity-driven entry into the TCA cycle (Fig. 4A'; Appendix Fig. S2B). An increased flow of labeled C13 pyruvate into lactate in HSD condition was also apparent (Fig. 4A'; Appendix Fig. S2B). Even though Ldh transcript levels did not reveal any significant upregulation, the increased M + 3 labeling in lactate upon HSD exposure implied increased LDH activity and demonstrated elevated aerobic glycolytic activity in these immune cells (Vander Heiden et al, 2009). Thus, considering the biochemical data, increased sugar exposure in the immune cells, exaggerated the overall flow of pyruvate into the TCA via PDH and into lactate via LDH (Fig. 4A').

Next, we modulated these steps to comprehend any precise contribution of these metabolic states to the growth regulation of HSD. Pyruvate can undergo two primary metabolic fates: conversion to lactate via lactate dehydrogenase (LDH), or transformation into acetyl-CoA through the action of pyruvate dehydrogenase (PDH) (Fig. 4B). To gauge control exerted via PDH, we genetically downregulated PDH by expressing *RNAi* against the *Pyruvate dehydrogenase E1 alpha subunit* (*Pdha*) in the immune cells (*Hml^Δ>GFP/Pdha^RNAi*). All the genetic knockdowns conducted were compared with their respective genetic background controls. This genetic manipulation showed no change in animal size considering both wing areas and body size measurements, which remained

comparable to control sizes across both the genders (Figs. 4C–D',G,H and EV3A,B',E,F). Contrarily, when we modulated pyruvate conversion towards lactate and performed similar genetic knockdown of *Ldh* enzyme (*Hml^Δ>GFP/Ldh^RNAi*), we observed a significant increase in adult fly sizes both in terms of wing area and body length (Figs. 4E,E',G,H and EV3C,C',E,F). The recovery of growth seen on HSD was unlike *Pdha^RNAi*, and unveiled a unique influence of immune cell glycolytic state on animal size control. We also did the converse experiment to further elevate *Ldh* in the immune cells by overexpressing *Ldh* in them. This resulted in further reduction in animal sizes on HSD (Figs. 4F,F",G,H and EV3D,D',E,F), and supported a growth inhibitory role for lactate. The data suggested that the growth retardation seen in HSD could arise as a consequence of elevated Ldh activity in them. The data were unlike PDH genetic manipulations and unveiled a stronger influence of immune cell glycolytic state on animal size control. The recovery in adult fly sizes seen with downregulating *Ldh* in immune cell, indicated the sufficiency of immune cell lactate production on adult growth inhibition on HSD.

Following this, an in-depth investigation of lipid metabolism genes was undertaken. Both from the transcriptomic data and the genetic screen a substantial contribution of members from this pathway was highly evident. Genes included, the de novo lipogenic enzyme *ACC* (*Acetyl CoA carboxylase*), TAG synthesis enzymes, and the lipolytic gene *bmm* (*brummer*), which prompted us to undertake their systematic characterization. We first validated the RNA sequencing results using real-time quantitative PCR (qPCR) and analyzed immune cells isolated from larvae reared on regular food (RF), short-term high sugar diet (4 hr.HSD), and constitutive HSD (Ct.HSD) conditions. qPCR analysis confirmed a significant upregulation of *ACC* only in 4 hr.HSD immune cells and not under Ct.HSD conditions (Fig. EV4A1). We also examined expression of *GPAT1, Agpat4, Lpin*, and *midway* (encoding *DGAT1*), and observed that *GPAT1* and *midway* were significantly upregulated in Ct.HSD (Fig. EV4A2–A5). *Gpdh1*, which generates glycerol 3-phosphate for TAG synthesis, was also upregulated under Ct.HSD (Fig. EV4A6). Also, lipolytic enzyme, *bmm*, its transcript levels were upregulated on Ct.HSD as well (Fig. EV4A7). These data suggest a biphasic transcriptional response in immune cells: early induction of de novo fatty acid synthesis via *ACC* in 4 hr.HSD, and sustained induction of TAG synthesis genes under prolonged HSD, and in parallel breakdown of lipids.

Interestingly, when assessed for ACC protein (Fig. 5A) expression by immunostaining of immune cells from RF, 4 hr.HSD, and Ct.HSD larvae (Fig. 5B–D), ACC protein levels were contrasting to the qPCR data. ACC protein levels were not elevated in 4 hr.HSD (Fig. 5C,E) and were significantly reduced in Ct.HSD (Fig. 5D,E), suggesting a post transcriptional regulation of ACC that resulted in reduction in its protein levels in immune cells. To check the

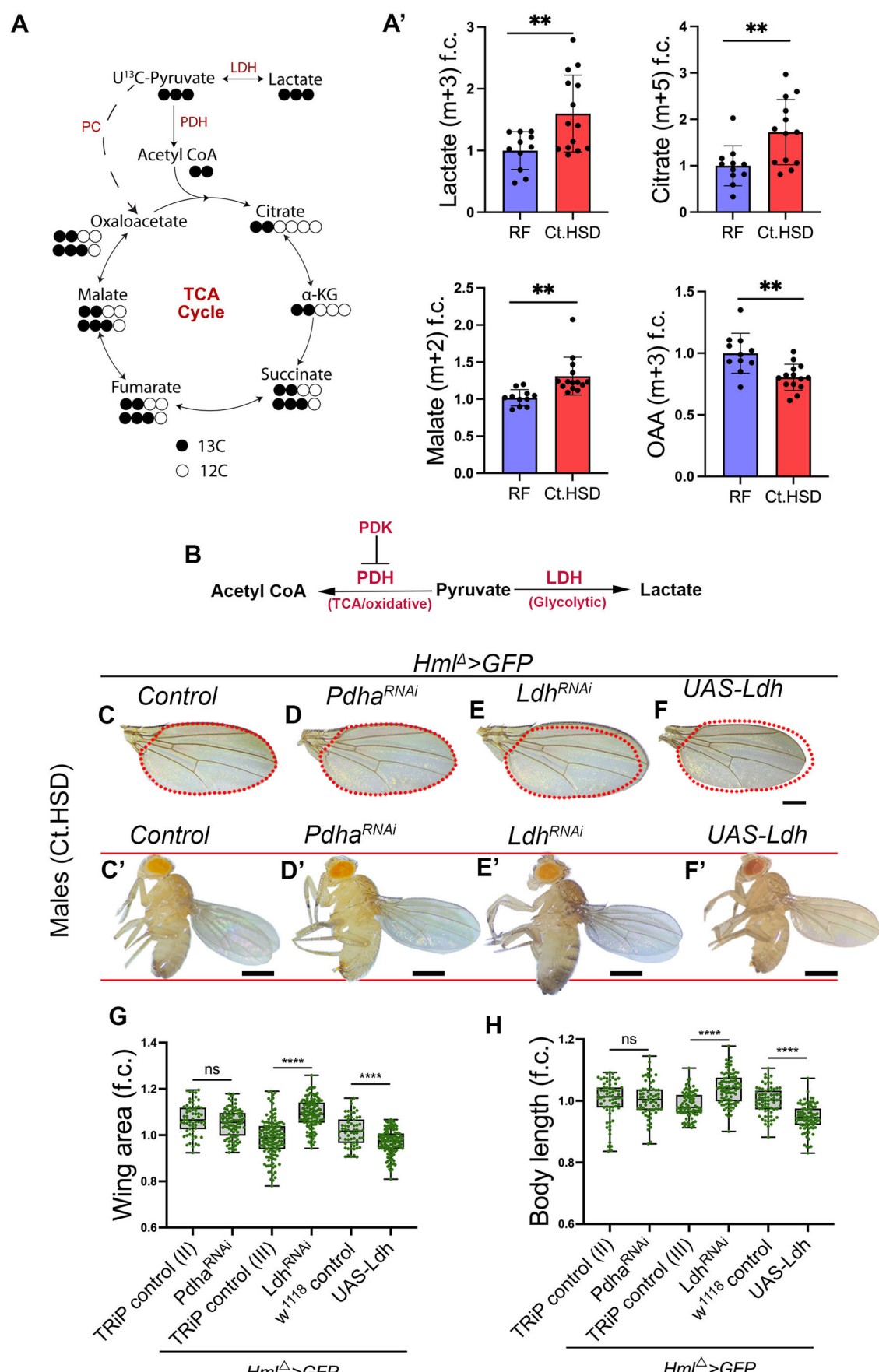

◄ **Figure 4. Glycolytic state in immune cells represses growth on HSD.**

(A) Schematic representation of U13C pyruvate label incorporation in TCA metabolites and lactate, where Pdh-derived pyruvate conversion labels two carbon in TCA metabolites via acetyl CoA and PC mediated pyruvate leads to three carbon incorporation in OAA. (A') Isotopic distribution of labeled U13C pyruvate in $Hml^{\Delta}>GFP/w^{1118}$ (Control, RF) and $Hml^{\Delta}>GFP/w^{1118}$ (Ct.HSD) conditions. Ct.HSD resulted in an increase in M + 5 label incorporation in citrate ($n = 13$, $P = 0.0057$), M + 2 label incorporation in malate ($n = 14$, $P = 0.0018$), a decrease in M + 3 label incorporation in OAA ($n = 14$, $P = 0.0013$) and an increase in M + 3 label incorporation in lactate ($n = 14$, $P = 0.0052$) in comparison to $Hml^{\Delta}>GFP/w^{1118}$ (Control, RF), citrate (m + 5, $n = 11$), malate (m + 2, $n = 12$), OAA (m + 3, $n = 11$), and lactate (m + 3, $n = 11$). (B) Pyruvate metabolism into acetyl CoA under the regulation of PDH enzyme fuels the TCA /oxidative metabolism. PDK inhibits PDH activity and regulates TCA. Pyruvate conversion to lactate is driven by Ldh enzymatic activity. (C–H) Modulating larval immune cell TCA and glycolytic activity affects adult growth. Representative images of wings and flies of adult males (C–F') showing size phenotype on Ct.HSD from respective genetic backgrounds. Compared to (C, C') Ct.HSD Control ($Hml^{\Delta}>GFP/w^{1118}$), moderating TCA activity by (D, D') expressing PdhaRNAi ($Hml^{\Delta}>GFP/Pdha^{RNAi}$) to reduce TCA resulted in no change in animal size. However, (E, E') downregulating glycolytic activity by expressing $Ldh^{RNAi}$ ($Hml^{\Delta}>GFP/Ldh^{RNAi}$) lead to size increase and increasing glycolytic activity via Ldh overexpression ($Hml^{\Delta}>GFP/UAS-Ldh$) (F,F') showed size reduction. (G) Quantification of wing area in $Hml^{\Delta}>GFP/Pdha^{RNAi}$ (Ct.HSD, $N = 3$, $n = 91$, $P = 0.0855$) in comparison to $Hml^{\Delta}>GFP/TRiP (II)$ control (Ct.HSD, $N = 3$, $n = 62$) and $Hml^{\Delta}>GFP/Ldh^{RNAi}$ (Ct.HSD, $N = 3$, $n = 125$, $P < 0.0001$) in comparison to $Hml^{\Delta}>GFP/TRiP (III)$ control (Ct.HSD, $N = 3$, $n = 131$). $Hml^{\Delta}>GFP/UAS-Ldh$ (Ct.HSD, $N = 3$, $n = 119$, $P < 0.0001$) in comparison to $Hml^{\Delta}>GFP/w^{1118}$ control (Ct.HSD, $N = 3$, $n = 61$). (H) Quantification of body length in $Hml^{\Delta}>GFP/Pdha^{RNAi}$ (Ct.HSD, $N = 3$, $n = 66$, $P = 0.6513$) in comparison to $Hml^{\Delta}>GFP/TRiP (II)$ control (Ct.HSD, $N = 3$, $n = 67$) and $Hml^{\Delta}>GFP/Ldh^{RNAi}$ (Ct.HSD, $N = 3$, $n = 87$, $P < 0.0001$) in comparison to $Hml^{\Delta}>GFP/TRiP (III)$ control (Ct.HSD, $N = 3$, $n = 82$). $Hml^{\Delta}>GFP/UAS-Ldh$ (Ct.HSD, $N = 3$, $n = 83$, $P < 0.0001$) in comparison to $Hml^{\Delta}>GFP/w^{1118}$ control (Ct.HSD, $N = 3$, $n = 71$). Data information: RF and Ct.HSD correspond to regular food and constitutive high sugar diet, respectively. Scale bar: 0.5 mm for flies and 0.25 mm for wings. In quantification graphs, shown in (A'), each dot represents an experimental repeat and in (G, H), each dot represents single animal. Except for (A'), where comparisons are with respect to control on RF, in all other panels comparison for significance is with respective background control on HSD. Asterisks mark statistically significant differences (*$P < 0.05$; **$P < 0.01$; ***$P < 0.001$; ****$P < 0.0001$). The statistical analysis applied for (A') is unpaired $t$ test, for other (G, H) Mann–Whitney test. N indicates the number of independent biological replicates, and n refers to the total number of animals analyzed. Only the right wing of each adult fly was selected for quantification. The differences in wing areas or fly body lengths in panels are indicated with a red dotted line or two horizontal red lines that highlight changes across genotypes. In bar graphs, data are presented as mean ± SD. Box plots show the median (center line), 25th–75th percentiles (bounds of box), and whiskers extending to the minimum and maximum values; all individual data points are shown. Source data are available online for this figure.

functional relevance of ACC in Ct.HSD, we used RNAi mediated knocked down of ACC in immune cells ($Hml^{\Delta}>GFP/ACC^{RNAi}$). In $ACC^{RNAi}$ condition, we observed a significant reduction in immune cell lipid content relative to Ct.HSD blood cells (Figs. 5F,G and EV4B,C). More importantly, ACC knockdown also resulted in reduced animal size, evident in both wing area and body length across sexes (Figs. 5K–L',P,Q and EV4G–H',L,M), indicating its role in growth promotion. Thus, even though immune cells showed limited ACC protein levels in Ct.HSD condition, the genetic data confirmed ACC's active state in HSD immune cells and its functional contribution to intracellular lipogenesis and consequences on systemic growth modulation. However, given the reduced ACC protein levels in immune cells, we tested whether ACC overexpression could lead to any further betterment and rescue HSD-induced growth defect. Indeed, ACC overexpression ($Hml^{\Delta}>GFP/UAS-ACC$) markedly increased immune lipid levels furthermore (Figs. 5H and EV4D) and resulted in significant enhancement of systemic growth (Figs. 5M,M' and EV4I,I') suggesting that dampened ACC protein expression in immune cells constrains growth potential under HSD and raising it beyond the threshold of what is seen in HSD had beneficial consequences on growth in this condition.

Following de novo lipogenesis, we examined the contribution of TAG synthesis pathway genes to growth. Here, we undertook immune-specific knockdown of Gpat4 (rate-limiting enzyme, $Hml^{\Delta}>GFP/Gpat4^{RNAi}$) and Agpat3 ($Hml^{\Delta}>GFP/Agpat3^{RNAi}$) which were also identified in the screen. Their genetic knockdowns resulted in reduced intracellular lipid levels in the immune cells on HSD (Figs. 5I,J and EV4E,F), and phenocopied the ACC loss-of-function effects. Both manipulations resulted in smaller animals, as quantified by wing area and body length in both sexes (Figs. 5N–O',P,Q and Fig. EV4J–K',L,M) and re-confirmed the screen findings.

We know that on HSD, circulating TAGs are elevated, which are likely derived from the fat body, which is also in a lipogenic mode

(Pasco and Léopold, 2012; Musselman et al, 2011). Hence, we also evaluated if circulating lipid uptake by immune cells and any contribution this trajectory had on intracellular lipid content and systemic growth. For this, we targeted croquemort (crq), a CD36 homolog involved in lipid scavenging (Franc et al, 1996; Guillou et al, 2016; Kiran et al, 2022). CRQ protein levels were elevated in Ct.HSD immune cells (Appendix Fig. S3A–C), and its knockdown resulted in reduced lipid accumulation (Appendix Fig. S3D–E') in immune cells. This genetic condition also demonstrated diminished animal growth (Appendix Fig. S3F–K') across males and females in wing span and body length, mirroring the effects of impaired lipogenesis described above.

Altogether, the results stated above, indicate that both lipid synthesis de novo and TAG synthesis along with lipid uptake, contributed to building intracellular pools of lipids in the immune cells in HSD condition. Surprisingly, loss of any of these steps was sufficient to deprive intracellular pools of lipid and cause further growth retardation. The data suggested a combinatorial mode where multiple arms of the lipid metabolic pathway cooperated to maintain elevated levels of intracellular TAG lipids in the immune cells and together supported animal growth under HSD. Surprisingly, this implied that lipogenesis favored growth, and in spite of its induction, the animals on HSD demonstrated a growth defect.

Immune cells are unlike fat body tissue, they lack any specialized storage structures, and therefore have limited capacity to retain excessive synthesized lipids. As they exhibited a pronounced lipogenic phenotype under HSD conditions, the parallel expression of lipolytic genes like brummer (bmm) lipase—responsible for triacylglycerol (TAG) hydrolysis (Figs. EV5A and EV4A7) was consistent with the notion of limited storage. Thus, the compensatory activation of the intracellular lipolytic pathway in HSD immune cells could be their intrinsic response. The impact of lipolysis on growth was therefore characterized, and we observed that immune cell-specific knockdown of bmm ($Hml^{\Delta}>GFP/bmm^{RNAi}$) resulted in significant accumulation of lipid droplets

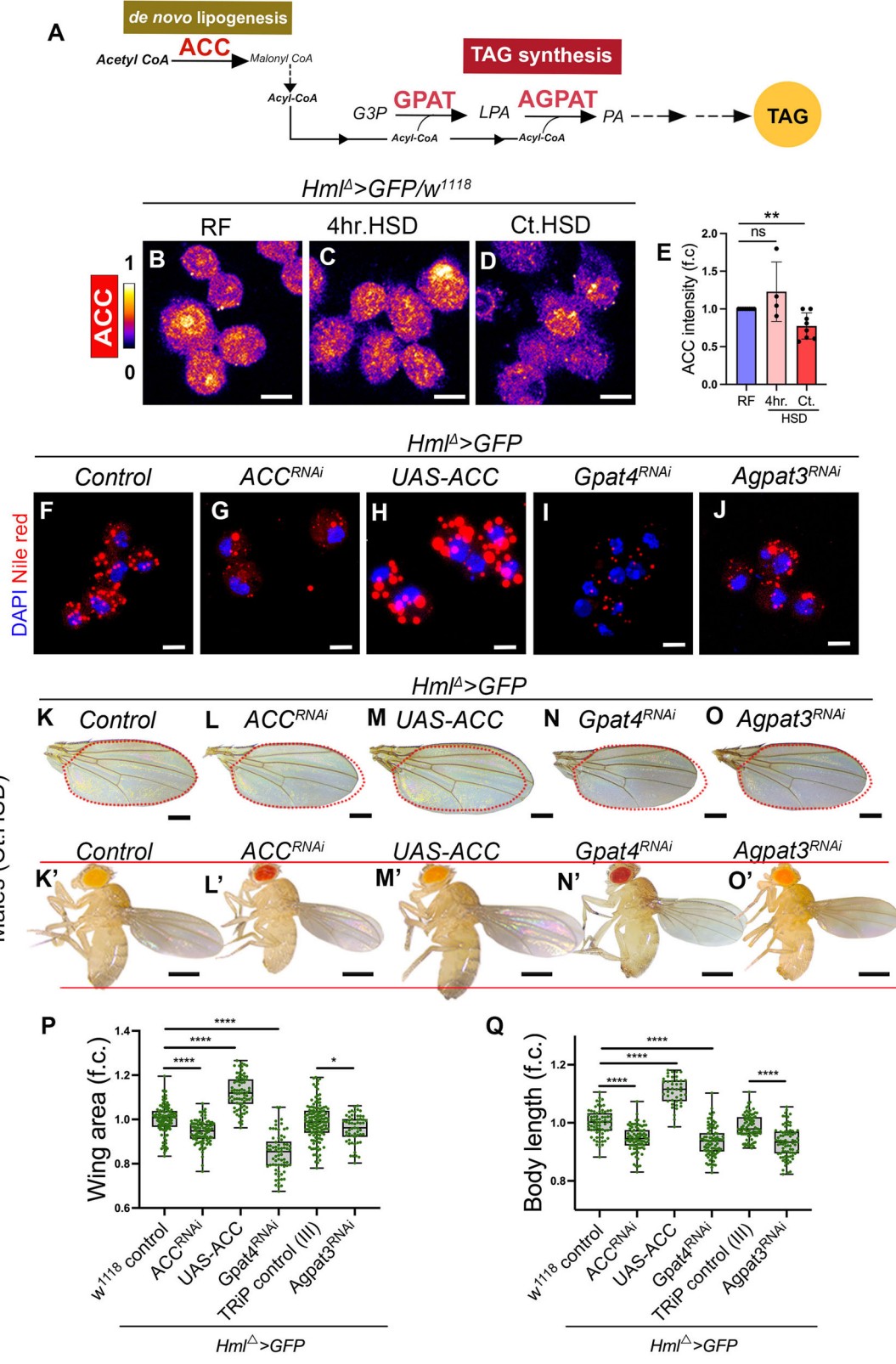

**Figure 5. De novo lipogenesis and TAG synthesis in immune cells acts as pro-growth on HSD.**

(A) Schematic representation of de novo lipogenesis and Triacylglycerol (TAG) synthesis pathway. (B–D) Representative images of immune cells stained for Acetyl CoA carboxylase (ACC) protein. Compared to RF control (*Hml^Δ^>GFP/w^1118^*) (B), ACC expression is unchanged under 4 hr.HSD (*Hml^Δ^>GFP/w^1118^*) (C) but markedly reduced under constitutive HSD (*Hml^Δ^>GFP/w^1118^*) (D). (E) Relative quantification of ACC protein expression, RF (N = 8, n = 80), 4 hr.HSD (N = 4, n = 40, P = 0.1150) and Ct.HSD (N = 8, n = 80, P = 0.0024). (F–J) Representative images of immune cells stained with Nile Red (red) to visualize lipid droplets. Compared to Ct.HSD control (*Hml^Δ^>GFP/w^1118^*) (F), loss of ACC function (*Hml^Δ^>GFP/ACC^RNAi^*) (G) or loss of TAG synthesis enzymes (*Hml^Δ^>GFP/Gpat4^RNAi^*) (I) and (*Hml^Δ^>GFP/Agpat3^RNAi^*) (J) reduces lipid droplet accumulation. In contrast, increased ACC expression (*Hml^Δ^>GFP/UAS-ACC*) (H) elevates lipid droplet levels. See Fig. EV4B–F for GFP-labeled immune cells co-stained with Nile Red. (K–O') Modulating larval immune cell lipid homeostasis alters adult male fly growth under Ct.HSD. Representative images of wings (K–O) and adult male flies (K'–O'). Compared to Ct.HSD control (*Hml^Δ^>GFP/w^1118^*) (K, K'), knockdown of ACC (*Hml^Δ^>GFP/ACC^RNAi^*) (L, L') reduces adult male size, while increased ACC expression (*Hml^Δ^>GFP/UAS-ACC*) (M, M') partially restores growth, producing larger adult males than control. Loss of TAG synthesis by knocking down Gpat4 (*Hml^Δ^>GFP/Gpat4^RNAi^*) (N, N') or Agpat3 (*Hml^Δ^>GFP/Agpat3^RNAi^*) (O, O') also reduces adult male size compared to control. (P) Quantification of wing area in *Hml^Δ^>GFP/ACC^RNAi^* (N = 3, n = 90, P < 0.0001), *Hml^Δ^>GFP/UAS- ACC* (N = 3, n = 75, P < 0.0001), *Hml^Δ^>GFP/Gpat4^RNAi^* (N = 3, n = 70, P < 0.0001) in comparison to Ct.HSD Control, *Hml^Δ^>GFP/w^1118^*, N = 3, n = 101). *Hml^Δ^>GFP/Agpat3^RNAi^* (N = 3, n = 55, P = 0.0158 in comparison to *Hml^Δ^>GFP/TRiP (III) control* (Ct.HSD, N = 3, n = 131). (Q) Quantification of body length in *Hml^Δ^>GFP/ACC^RNAi^* (N = 3, n = 83, P < 0.0001), *Hml^Δ^>GFP/UAS-ACC* (N = 3, n = 43, P < 0.0001), *Hml^Δ^>GFP/Gpat4^RNAi^* (N = 3, n = 88, P < 0.0001) in comparison to Ct.HSD Control, *Hml^Δ^>GFP/w^1118^*, N = 3, n = 71). *Hml^Δ^>GFP/Agpat3^RNAi^* (N = 3, n = 80, P < 0.0001 in comparison to *Hml^Δ^>GFP/TRiP (III) Control* (Ct.HSD, N = 3, n = 82). Data information: DNA is stained with DAPI (blue). RF and Ct.HSD corresponds to regular food and a constitutive high sugar diet, respectively. ACC staining is shown in spectral mode in (B–D). Nile red (red) mark lipids in (F–J). Scale bar: 5 μm for immune cells, 0.5 mm for flies and 0.25 mm for wings. (P, Q) Each dot represents an animal and in (E) each dot represents an experimental repeat. Except for (E), where comparisons are with respect to Control on RF, in all other panels comparison for significance is with respective background control on Ct.HSD. Asterisks mark statistically significant differences (*P < 0.05; **P < 0.01; ***P < 0.001; ****P < 0.0001). The statistical analysis applied for (E) is unpaired t test, for other panels (P, Q) Mann–Whitney test. N indicates the number of independent biological replicates, and n refers to the total number of larvae analyzed. Only right wing from each adult fly was selected for quantification. The differences in wing areas or fly body lengths in panels is indicated with a red dotted line or two horizontal red lines that highlight changes across genotypes. In bar graphs, data are presented as mean ± SD. Box plots show the median (center line), 25th–75th percentiles (bounds of box), and whiskers extending to the minimum and maximum values; all individual data points are shown. Source data are available online for this figure.

within these cells (Fig. EV5B–C'). This indicated its active state in lipid breakdown and in this context, loss of *bmm* also conferred a pronounced enhancement in systemic growth (Fig. EV5E–F',J–K',H,I,M,N). Contrarily, overexpression of *bmm* in immune cells (*Hml^Δ^>GFP/UAS-bmm*) which depleted cellular lipid content significantly (Fig. EV5D,D') further exacerbated HSD-induced growth defects and resulted in much smaller adults on HSD (Fig. EV5G,G',L,L',H,I,M,N).

Thus, the concurrent state of lipogenic and lipolytic programs are conflicted and our results reveal that it is perhaps the rates of these processes, lipid synthesis versus its breakdown that determine the total levels of intracellular TAGs seen in HSD immune cells. This context becomes central towards supporting growth, as the later step of lipolysis antagonizes systemic growth, while the former promotes it. The data importantly reveal that preserving a lipid-rich state within immune cells—either by promoting lipogenesis or by suppressing lipolysis—is sufficient to restore growth under dietary sugar stress comparable to sizes seen in homeostasis. We propose that the restrain on storage capacity within immune cells, is probably why immune cells cannot sustain continued lipogenesis. Even though lipogenesis favors growth, its induction with concurrent induction of lipolysis counters it, rendering an overall negative impact on growth.

In light of these findings, we sought to determine whether these immune metabolic states identified were general regulators of growth or whether they represented sugar-sensitive, adaptive programs specifically engaged to support growth under dietary stress. To address this, we examined the impact of modulating key metabolic pathways in immune cells under homeostatic conditions —i.e., a regular diet (RF)—by measuring adult body size as done for HSD.

We assessed glycolysis, TCA cycling, lipid synthesis, lipid uptake, and lipolysis by specifically downregulating *Ldh*, *Pdha*, *ACC*, *Gpat4*, *Agpat3*, *crq*, and *bmm* in immune cells. The resulting adult sizing as done for HSD was also conducted for RF condition.

Inhibition of TCA activity via *Pdha* knockdown did not affect adult size under regular diet conditions (Fig. EV6A,A',C,C'), suggesting that TCA in immune cells is not a limiting factor for systemic growth at baseline. However, *Ldh* knockdown had a discernible growth increase on body size and wing area, with the size increase being more prominent in females compared to males (Fig. EV6B,B',D,D'). This was surprising, as loss of *Ldh* from the blood cells could manifest a growth increase over and above the homeostatic size of the regular food control flies. These genetic data suggested that blood cells in RF are also glycolytic but considering the mass spectrometry-based metabolic flux analyses, it is significantly lower in regular diet compared to high sugar diet (HSD) conditions (Appendix Fig. S2B). Nevertheless, the negative influence of immune cell *Ldh* activity on systemic growth even in homeostatic conditions implied this immune metabolic state to be a growth suppressor regardless of the dietary state. To directly assess its sufficiency in this regard, we asked if enhanced immune glycolysis could retard growth. We overexpressed *Ldh* in immune cells under regular diet conditions, and strikingly, this manipulation recapitulated the HSD-induced growth suppression phenotype, resulting in significantly smaller adult flies (Fig. EV6B,B',D,D'). These data collectively demonstrated that immune cell glycolysis, mediated by *Ldh*, repressed organismal growth and alluded to an interesting plasticity in growth that can be moderated by this metabolic state. Importantly, the homeostatic fly size that is set to scale is following the repression brought about by the glycolytic input from the immune cells. If this repression is unveiled, the fly sizes can be much larger than seen in homeostasis. Contrarily, when further enforced, can lead to much smaller adults. We propose that the degree of *Ldh*-dependent glycolytic activation in immune cells governs the extent of its systemic growth outcomes, wherein excessive glycolysis, could possibly override any growth-promoting signal and enforce a growth-restrictive state.

We similarly assessed the contribution of lipid anabolic pathways to growth regulation under homeostatic (regular food;

RF) conditions. Targeted knockdown of key lipogenic genes—*ACC* ($Hml^\Delta > GFP/ACC^{RNAi}$), *Gpat4* ($Hml^\Delta > GFP/Gpat4^{RNAi}$), and *Agpat3* ($Hml^\Delta > GFP/Agpat3^{RNAi}$)—in immune cells did not impact adult body size in either sex (Fig. EV6E,E'), indicating that immune cell lipogenesis is not essential for maintaining systemic growth under nutrient replete conditions. This is also true for wing span areas for the majority of the lines across genders (Fig. EV6G,G'). These findings contrast with observations under high sugar diet (HSD), where lipogenesis is a limiting factor for growth, suggesting that the metabolic requirement for lipid biosynthesis in immune cells is condition-specific.

We then tested whether lipid uptake by immune cells plays a more general role in growth regulation. Knockdown of *crq* ($Hml^\Delta > GFP/crq^{RNAi}$) resulted in no change in adult size, even under a regular diet (Fig. EV6I–J'). Lastly, we assessed the impact of impairing lipid catabolism via *bmm* knockdown ($Hml^\Delta > GFP/bmm^{RNAi}$). In $bmm^{RNAi}$ animals displayed no change in body length (Fig. EV6I–J').

Strikingly, enforced activation of lipogenesis via overexpression of *ACC* in immune cells under RF condition resulted in a significant increase in body size compared to controls (Fig. EV6F,F',H,H'). This overgrowth phenotype parallels the effect of *Ldh* knockdown, indicating that immune cell-specific lipogenic activity while it is dispensable for growth in RF condition, it is sufficient to drive growth-promoting signals when upregulated, even in the absence of dietary stress. Also, forced expression of *bmm* to drive breakdown of intracellular lipids resulted in smaller adults, more prominent in the males (Fig. EV6I,I') as opposed to the females (Fig. EV6J,J') and indicated the sufficiency of immune-derived fatty acids in negatively affecting growth independent of the dietary state (Fig. EV6I–J').

Together, these results highlight that immune cell lipogenesis is not required for growth under normal dietary conditions; its elevation in HSD is likely an adaptive program initiated to potentiate systemic growth, underscoring a context-dependent utilization of immune lipogenic state in organismal size control. In this regard, the intracellular storage limitation of an immune cell invokes the generation of FFA (Free Fatty Acid) through lipolysis. The negative impact via immune-derived lipids on systemic growth, in addition to elevated Ldh activity, exerts an overall suppressive state leading to a smaller-sized adult on HSD.

Overall, the findings reveal immune metabolic insights underlying the growth restriction seen in HSD, and further highlights the sufficiency of moderating it to achieve optimal growth while thriving in stress scenarios. The prime function of immune cells and the importance of their intracellular metabolic homeostasis in systemic growth control and enabling developmental plasticity was an unexpected observation from these data.

## Immune metabolic changes influence imaginal disc development

The findings above highlighted the potential of immune cells to control systemic growth, both under homeostatic and dietary stress conditions. The emergence of a unique adaptive growth program by immune cells under HSD led us to investigate the link between these cells and adult size regulation.

To understand how these adult size differences manifested, we examined whether they originated from early developmental defects and therefore we assessed larval growth. Specifically, we measured third-instar larval body length across the various immune-specific genetic manipulations under HSD, but we did not observe any consistent correlation between changes in adult size and larval body lengths (Fig. 6A–G). In conditions like $ACC^{RNAi}$, $crq^{RNAi}$ where adult sizes are decreased and $bmm^{RNAi}$ where it is increased, respectively, we did not observe any change in larval sizing (Fig. 6A–D,V). However, notable exceptions that included $Ldh^{RNAi}$ and *ACC* gain-of-function (GOF), where an increase in adult size was observed, larval sizes were increased (Fig. 6E,F,W) and $Gpat4^{RNAi}$, which resulted in smaller adult flies, showed mildly reduced larval growth (Fig. 6G,W).

Given the lack of any consistent link to larval body size, we sought to assess imaginal discs, which are key determinants of adult structures (Cohen et al, 1991). We performed a comparative analysis of wandering third-instar larval wing imaginal discs under HSD and assessed the effects of immune-specific genetic perturbations on their growth. We also evaluated fat body cell areas to determine whether immune influences were organ-specific. Strikingly, immune cell manipulations consistently altered wing disc growth across all genetic conditions, irrespective of changes detected in larval sizing. $Ldh^{RNAi}$, $ACC^{RNAi}$, *UAS-ACC*, $Gpat4^{RNAi}$, $crq^{RNAi}$, $bmm^{RNAi}$, in all of these genetic conditions a corresponding change in wing disc areas was evident (Fig. 6H–N,V',W'). Conditions ($ACC^{RNAi}$, $Gpat4^{RNAi}$, $crq^{RNAi}$) that resulted in smaller adults showed reduction in the sizes of the wandering 3rd instar larval imaginal discs (Fig. 6I,J,N,V'), and vice versa, where $Ldh^{RNAi}$, *UAS-ACC* where larger adults were obtained, revealed a corresponding increase in larval wing imaginal disc sizes (Fig. 6L,M,W'). This confirmed a direct involvement of immune cells in regulating adult size via control of imaginal tissue development without moderating larval sizing. In addition, alterations in fat body development suggested a broader, multi-organ crosstalk initiated by immune cells. Conditions like $crq^{RNAi}$, $ACC^{RNAi}$, $Gpat4^{RNAi}$ showed a reduction fat body cell areas (Fig. 6O–Q,U,V''), while $Ldh^{RNAi}$ and *UAS-ACC*, that had larger larval and adult animal sizes, revealed increase in fat body cell areas (Fig. 6S,T,W''). However, no significant difference in fat body cell size observed in $bmm^{RNAi}$ condition (Fig. 6R). These observations in totality suggest that, overall, adult size changes mediated by immune cells may occur independent of any control of larval growth and instead was a consequence of scaling of imaginal tissues. The engagement with fat body and the dynamics of growth signaling in this tissue alluded to an organismal-level crosstalk initiated by immune cells in growth homeostasis.

We also characterized the phenomenon from the standpoint of changes in immune cells, and quantified immune cell (hemocyte) numbers across the genetic conditions (Fig. 6V''',W'''). Only in some manipulations, we could identify an effect on hemocyte counts albeit the changes in immune numbers were mild. For instance, conditions resulting in smaller adults—such as $ACC^{RNAi}$ and $Gpat4^{RNAi}$—showed reduction in hemocyte numbers (Fig. 6V''',W'''), but $crq^{RNAi}$ and $bmm^{RNAi}$ revealed no difference with respect to its corresponding controls (Fig. 6V'''). Interestingly, *ACC* overexpression, which produced larger adults, actually exhibited reduced hemocyte numbers with respect to its control, while $Ldh^{RNAi}$ which also yields larger adults, revealed no difference in blood cell numbers (Fig. 6W'''). This dissociation between changes in adult growth dynamics and immune cell numbers aligns

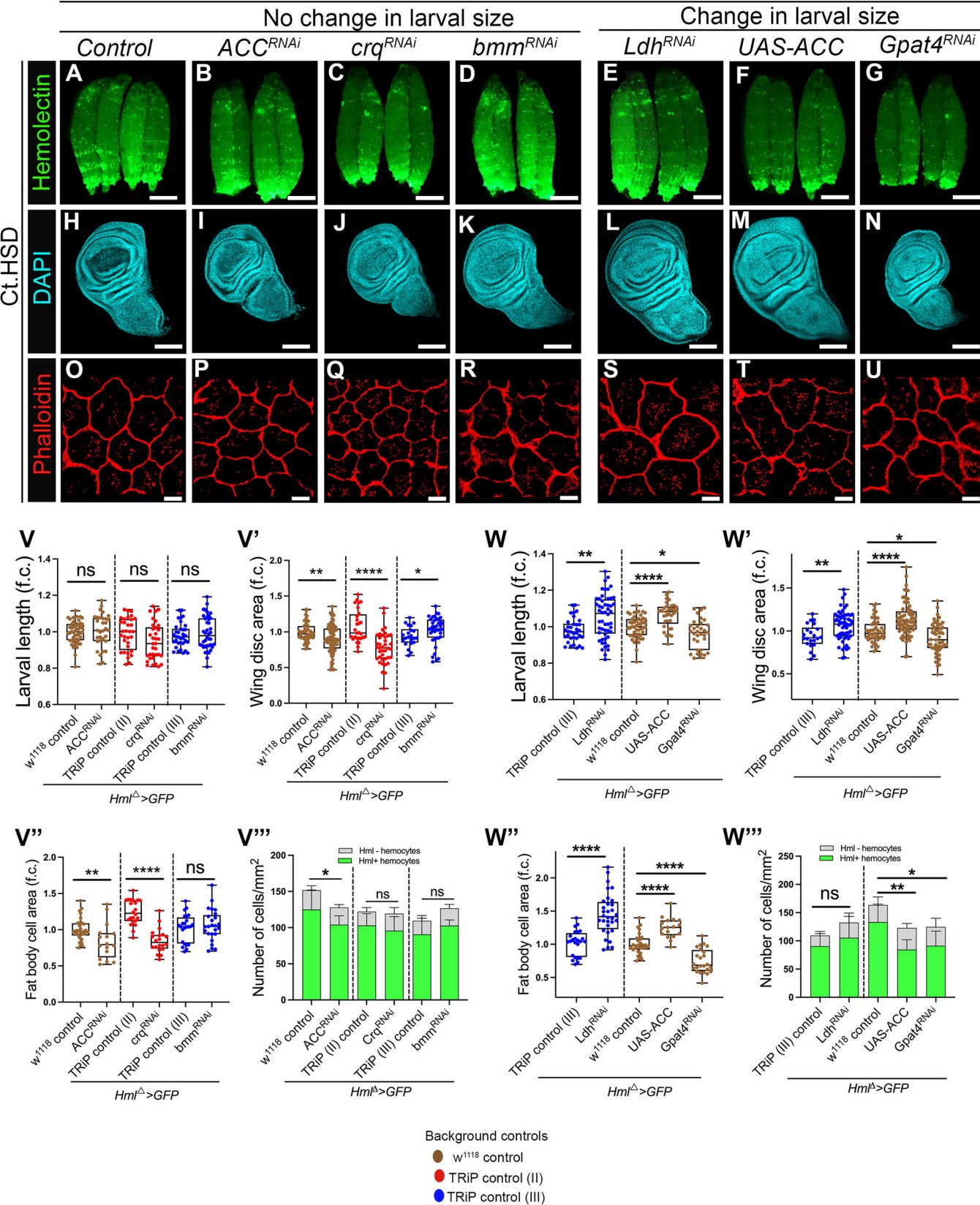

Figure 6. Immune metabolic state changes and impact on imaginal disc development.

(A–U) Representative images of wandering third-instar larvae, wing imaginal disc and fat body respectively on Ct.HSD to showcase the size change. (V) Quantification of larval body length (WL3, Ct.HSD) in $Hml^\Delta$>GFP/ACC$^{RNAi}$ (N = 3, n = 32, P = 0.5110) in comparison to Ct.HSD Control, $Hml^\Delta$>GFP/w$^{1118}$, N = 3, n = 50). $Hml^\Delta$>GFP/crq$^{RNAi}$ (N = 3, n = 34, P = 0.0543) in comparison to $Hml^\Delta$>GFP/TRiP (II) control (Ct.HSD, N = 3, n = 32). $Hml^\Delta$>GFP/bmm$^{RNAi}$ (N = 3, n = 40, P = 0.7099) in comparison to $Hml^\Delta$>GFP/TRiP (III) control (Ct.HSD, N = , n = 3, N = 34). (V') Quantification of wing disc area (WL3, Ct.HSD) in $Hml^\Delta$>GFP/ACC$^{RNAi}$ (N = 3, n = 79, P = 0.0030) in comparison to Ct.HSD Control, $Hml^\Delta$>GFP/w$^{1118}$, N = 3, n = 47). $Hml^\Delta$>GFP/crq$^{RNAi}$ (N = 3, n = 39, P < 0.0001) in comparison to $Hml^\Delta$>GFP/TRiP (II) control (Ct.HSD, N = 3, n = 28), $Hml^\Delta$>GFP/bmm$^{RNAi}$ (N = 3, n = 40, P = 0.0189) in comparison to $Hml^\Delta$>GFP/TRiP (III) control (Ct.HSD, N = 3, n = 22). (V'') Quantification of fat body cell area in $Hml^\Delta$>GFP/ACC$^{RNAi}$ (N = 3, n = 16, P = 0.0051) in comparison to Ct.HSD Control, $Hml^\Delta$>GFP/w$^{1118}$, N = 3, n = 31). $Hml^\Delta$>GFP/crq$^{RNAi}$ (N = 3, n = 20, P < 0.0001) in comparison to $Hml^\Delta$>GFP/TRiP (II) control (Ct.HSD, N = 3, n = 22), $Hml^\Delta$>GFP/bmm$^{RNAi}$ (N = 3, n = 25, P = 0.7738) in comparison to $Hml^\Delta$>GFP/TRiP (III) control (Ct.HSD, N = 3, n = 24). (V''') Quantification of total immune cell numbers Hml+ (green bar) and Hml- (space gray bar) in $Hml^\Delta$>GFP/ACC$^{RNAi}$ (N = 3, n = 30, P = 0.0140), in comparison to Ct.HSD Control, $Hml^\Delta$>GFP/w$^{1118}$, N = 3, n = 30). $Hml^\Delta$>GFP/crq$^{RNAi}$ (N = 3, n = 20, P = 0.9823) in comparison to $Hml^\Delta$>GFP/TRiP (III) control (Ct.HSD, N = 3, n = 13). $Hml^\Delta$>GFP/bmm$^{RNAi}$ (N = 3, n = 20, P = 0.1387) in comparison to $Hml^\Delta$>GFP/TRiP (III) control (Ct.HSD, N = , n = 30). (W) Quantification of larval body length (WL3, Ct.HSD) in $Hml^\Delta$>GFP/Ldh$^{RNAi}$ (Ct.HSD, N = 3, n = 52, P = 0.0025) in comparison to $Hml^\Delta$>GFP/TRiP (III) control (Ct.HSD, N = 3, n = 50). $Hml^\Delta$>GFP/UAS-ACC (N = 3, n = 33, P < 0.0001), $Hml^\Delta$>GFP/Gpat4$^{RNAi}$ (N = 3, n = 33, P = 0.0328) in comparison to Ct.HSD Control, $Hml^\Delta$>GFP/w$^{1118}$, N = 3, n = 38). (W') Quantification of wing disc area (WL3, Ct.HSD) in $Hml^\Delta$>GFP/Ldh$^{RNAi}$ (Ct.HSD, N = 3, n = 59, P = 0.0032) in comparison to $Hml^\Delta$>GFP/TRiP (III) control (Ct.HSD, N = 3, n = 22). $Hml^\Delta$>GFP/UAS-ACC (N = 3, n = 71, P < 0.0001), $Hml^\Delta$>GFP/Gpat4$^{RNAi}$ (N = 3, n = 56, P = 0.0156) in comparison to Ct.HSD Control, $Hml^\Delta$>GFP/w$^{1118}$, N = 3, n = 47). (W'') Quantification of fat body cell area in $Hml^\Delta$>GFP/Ldh$^{RNAi}$ (Ct.HSD, N = 3, n = 32, P < 0.0001) in comparison to $Hml^\Delta$>GFP/TRiP (III) control (Ct.HSD, N = 3, n = 24). $Hml^\Delta$>GFP/UAS-ACC (N = 3, n = 17, P < 0.0001), $Hml^\Delta$>GFP/Gpat4$^{RNAi}$ (N = 3, n = 24, P < 0.0001) in comparison to Ct.HSD Control, $Hml^\Delta$>GFP/w$^{1118}$, N = 3, n = 31). (W''') Quantification of total immune cell numbers in $Hml^\Delta$>GFP/Ldh$^{RNAi}$ (Ct.HSD, N = 3, n = 28, P = 0.2516) in comparison to $Hml^\Delta$>GFP/TRiP (III) control (Ct.HSD, N = 3, n = 40). $Hml^\Delta$>GFP/UAS-ACC (N = 3, n = 24, P = 0.0072) and $Hml^\Delta$>GFP/Gpat4$^{RNAi}$ (N = 3, n = 24, P = 0.0139) in comparison to Ct.HSD Control, $Hml^\Delta$>GFP/w$^{1118}$, N = 3, n = 18). Data information: Ct.HSD correspond to a constitutive high sugar diet. Immune cells are shown in green ($Hml^\Delta$>UAS-GFP), wing discs stained with DAPI (cyan), fat body stained with phalloidin (red). Scale bars: 1000 μm for larval images (A–G), 100 μm for wing imaginal discs (H–N), and 20 μm for fat body (O–U). In quantification graphs (V–V'') and (W–W'') each dot represents an animal. Comparison for significance is with respective background controls on Ct.HSD. Asterisks mark statistically significant differences (*P < 0.05; **P < 0.01; ***P < 0.001; ****P < 0.0001). The statistical analysis applied for (V''', W''') is Two-way ANOVA with Sidak's multiple comparison test, for other panels (V, V', V'', W, W', W'') Mann–Whitney test. N indicates the number of independent biological replicates, and n refers to the total number of animals analyzed. In bar graphs, data are presented as mean ± SD. Box plots show the median (center line), 25th–75th percentiles (bounds of box), and whiskers extending to the minimum and maximum values; all individual data points are shown. Source data are available online for this figure.

with our previous findings (P et al, 2020), reinforcing the notion that immune cell metabolic state, rather than their number, governs systemic growth.

## Discussion

### Dietary stress-induced macrophage metabolic reprogramming as a determinant of animal growth

This study highlights a pivotal link between macrophage metabolic reprogramming and animal growth under dietary stress. By integrating immune cell profiling, genetic screening, transcriptomics, and metabolomics, we reveal how macrophages adapt metabolically to high sugar diets and how these changes modulate systemic growth. Under a normal diet, larval immune cells are minimally glycolytic and maintain lipid scavenging. However, high sugar exposure induces enhanced glycolysis, TCA cycle activity, and triacylglycerol (TAG) synthesis. Although this lipogenic shift provides metabolic buffering, it disrupts growth regulation. Our data show that glycolytic and lipolytic states inhibit growth, while TAG synthesis supports it. Genetic interventions reducing glycolysis/lipolysis or enhancing lipogenesis restore normal growth, suggesting an imbalance in immune metabolic states drives growth impairment under high sugar. The limited lipogenic capacity of immune cells and their predominant catabolic nature prevent them from fully mitigating sugar-induced stress, contributing to reduced adult size (Appendix Fig. S4).

The negative impact of high sugar on childhood growth is well recognized, yet its mechanistic basis remains poorly understood. Beyond known insulin resistance, our study introduces immune lipid metabolism as a critical modulator. Immune-derived lactate significantly contributes to growth retardation even under a regular diet, with effects exacerbated on HSD. The lactate-associated growth impairment echoes findings from intrauterine growth restriction models (Marconi et al, 1990). Conversely, diversion of sugar metabolism into lipogenesis, as seen with ACC gain-of-function, supports growth, likely by reducing pyruvate flux toward LDH. The specific impact on imaginal disc growth retardation is intriguing, and thus, we hypothesize that the excessive sugar breakdown with intermediates like lactate shifts the imaginal disc homeostasis to growth impairment. We hypothesize that the diversion of sugar breakdown on HSD into lipids however favors growth and limits the negative context imposed by sugar catabolism. The induction of de novo lipogenesis while allowing an alternate route to metabolize sugars, most likely also restricts pyruvate availability for LDH function, and is perhaps how the gain of ACC enables growth recovery. This is a lucrative hypothesis that however remains to be tested. While lipogenesis offers a protective alternative pathway, its extent in immune cells is restricted. The growth benefit seen with ACC overexpression implies that repression of lipogenesis may be lifted by enhancing ACC activity. This may reflect limitations in metabolic substrates or the high energetic cost of lipogenesis compared to direct lipid uptake.

### Immune cells as lipogenic organs under stress

Although not conventional storage organs, immune cells show a lipogenic shift under HSD, similar to fat bodies (Musselman et al, 2013). Transcriptomic data reveal upregulation of lipogenic genes and TAG synthesis enzymes, alongside increased intracellular TAG and lipid scavenging receptors like crq. This adaptation suggests immune cells contribute to systemic lipid buffering. Unlike inflammatory lipid accumulation, this sugar-induced lipogenesis correlates with downregulated immune responses, reflecting a metabolic, not inflammatory, adaptation. While limited expression

of ACC restricts lipogenesis, forced ACC overexpression enhances growth, indicating a growth-promoting lipogenic axis in immune cells. The energetic efficiency of lipid uptake and storage limitations may explain the restrained lipogenesis. The identification of oxysterol-binding proteins promoting lipogenesis and growth in our screen reinforces the immune lipogenic role under dietary stress.

The limited extent of lipogenic induction in immune cells may be because these cells unlike the fat body are never designed for storage functions. Lipid breakdown is therefore facilitated by bmm and most likely adds to raise the pool of free fatty acids (FFA) and negatively influences growth. FFA and their link with the development of inflammation and insulin insensitivity in peripheral tissues is well established (Johnson and Olefsky, 2013). High levels of circulating FFA and their uptake by non-adipose organs that cannot store fatty acids or their derivatives develop lipotoxicity leading to systemic insulin resistance (Postic and Girard, 2008; Unger, 2003). Thus, it is possible that catabolic activities of immune cells leading to lactate together with the elevated levels of FFA facilitates growth retardation through invoking insulin resistance, which is not unexpected as we do find changes in fat body cell areas when immune metabolic changes are conducted. Fat body cell areas showed recovery in conditions with reduced *Ldh* activity and gain of *ACC*, while conditions like loss of lipid synthesis that retarded growth also revealed a corresponding reduction in fat body cell areas. Collectively, our data opens a new paradigm to look at these cells in the face of dietary stresses, functioning much like fat body or adipose tissues and operating beyond their role in defensive functions.

## Immune metabolic heterogeneity at the interface of growth coordination

Our study is the first to systematically explore immune-metabolic contributions to growth using genetic and multi-omic approaches. Findings indicate contrasting immune metabolic states—glycolytic/lipolytic vs. lipogenic—with opposing effects on growth. This dichotomy suggests heterogeneity within the Hml$^+$ macrophage population, possibly influenced by ontogeny or induced by genetic manipulation. Previous single-cell studies have identified immune subpopulations with distinct metabolic profiles, supporting our hypothesis (Cattenoz et al, 2020). The HSD-induced metabolic reprogramming may mirror both embryonic (lipogenic/glycolytic) and larval (oxidative/lipolytic) states. Future single-cell studies are needed to validate the presence and function of these subpopulations in growth regulation.

Macrophages are increasingly recognized as systemic regulators, integrating nutrient cues and modulating organismal growth. Our previous work and others have demonstrated their influence on insulin signaling and developmental timing(P et al, 2020; Odegaard and Chawla, 2008). Here, we expand this framework to include additional immune-derived signals like adenosine, Imaginal morphogenesis protein-Late 2 (Impl2), unpaired 3 (upd3), Drosophila insulin-like peptide 8 (dilp8), hedgehog (hh), and PDGF- and VEGF-related factors (Pvfs), which collectively coordinate growth and metabolic balance. Immune metabolites and cytokines, including lactate, acetyl-CoA, unpaired-2 (Upd-2), Wingless (Wg), and neuropeptides like CCHamide2 (CCHa-2) and short Neuropeptide F (sNPF), further illustrate their multi-organ crosstalk capacity (Appendix Fig. S4). These findings underscore macrophages as active players in defining growth potential beyond traditional immune roles. The interplay of immune heterogeneity with coordination of growth however remains to be discerned and is a topic for our future explorations.

## Conclusions

This study uncovers a novel role for immune metabolic reprogramming in growth control, revealing macrophages as key regulators of systemic physiology in response to dietary challenges. The conserved role of macrophages in growth regulation across species, and their metabolic plasticity, positions them as central players in maintaining organismal homeostasis. While the stunted growth is often viewed as a pathological outcome, it may also reflect adaptive responses to environmental stress, including dietary excess. Our preliminary findings suggest enhanced TCA activity in immune cells improves survival on HSD, supporting a model where limited lipogenesis and reduced growth serve as developmental adaptations. Further studies are also needed to assess how these changes induced by immune cells influence long-term fitness, including fecundity and lifespan. Overall, this study offers a tip of the iceberg understanding of the immune-growth axis and lends a new avenue for investigations, addressing metabolic disorders linked to diet and development.

## Methods

**Reagents and tools table**

| Reagent/resource | Reference or source | Identifier or catalog number |
|---|---|---|
| **Experimental models** | | |
| *Drosophila melanogaster* | Bloomington Drosophila Stock Center (BDSC); Vienna Drosophila Resource Center (VDRC); FlyORF (University of Zurich) | N/A |
| *Pdha*$^{RNAi}$ | BDSC | #55345 |
| *Ldh*$^{RNAi}$ | BDSC | #33640 |
| UAS-Ldh | FLYORF | #F002924 |
| *ACC*$^{RNAi}$ | VDRC | #108631 |
| UAS-ACC | BDSC | #63225 |
| *Gpat4*$^{RNAi}$ | VDRC | #100728 |
| *Agpat3*$^{RNAi}$ | BDSC | #50568 |
| *bmm*$^{RNAi}$ | BDSC | #25926 |
| UAS-bmm | BDSC | #76600 |
| *crq*$^{RNAi}$ | BDSC | #40831 |
| UAS-LSD2-GFP | Michael Welte | N/A |
| TRiP RNAi control attp2 | BDSC | #36303 |
| TRiP RNAi control attp40 | BDSC | #36304 |
| **Antibodies** | | |
| Rabbit anti-ACC | Gift from Jacques Montagne, I2BC (CEA/CNRS/Université Paris-Saclay), France | NA (gift antibody) |
| Rabbit anti-Crq | Gift from Nathalie Franc, The Scripps Research Institute, La Jolla, USA | NA (gift antibody) |

| Reagent/resource | Reference or source | Identifier or catalog number |
|---|---|---|
| Goat anti-rabbit Alexa 546 | Invitrogen | A-11035 |
| **Oligonucleotides and other sequence-based reagents** | | |
| qRT-PCR Primers | Sequence | |
| Rp49 | F-5'-CGGATCGATATGCTAAGCTGT-3' R-5'-GCGCTTGTTCGATCCGTA-3' | This study |
| crq | F-5'-CACCTGCGCCAGTTATCGACAGCTGAG-3' R-5'-AATAGCCAAGGTGGGAATAATCCAG-3' | This study |
| ACC | F-5'-GCTGAATGAGGAGACCTCTAAC-3' R-5'-GGGAACGGGAAGGATGAAATA-3' | This study |
| Fasn1 | F-5'-CACCCAATACTCGGGTTCTATG-3' R-5'-CTTGCAGCTCAACAACGTAAAT-3' | This study |
| Gpdh1 | F-5'-GCACCACCAACACAAACATAC-3' R-5'-CTGCGGCCAACAACAAATC-3' | This study |
| Gpat1 | F-5'-CTTTAACGAGCCCTACTCCATAC-3' R-5'-GCAGATGGCTTGTAGACCTT-3' | This study |
| Agpat1 | F-5'-TTCCACCCATAGAGGGAAATAAC-3' R-5'-GCAGGCCAGTCCAATAACT-3' | This study |
| Lipin | F-5'-CAGTGGAGTGGTGACAGATAAA-3' R-5'-CTCCTCCTTGGAGAAGTCAATG-3' | This study |
| Midway | F-5'-ACGAAGGCGAAGGACATAAC-3' R-5'-CCCAAGCCCTCTGCAATTA-3' | This study |
| bmm | F-5'-GCAACACGAACAAGGTGAAAG-3' R-5'-TCGACAGAGCCTTCGTAGAT-3' | This study |
| **Chemicals, enzymes, and other reagents** | | |
| DAPI | Sigma | Cat# D9542 |
| Nile red | Sigma | Cat# N3013 |
| Phalloidin-Red | Sigma | Cat# 94072 |
| Dihydroethidium | Invitrogen | Cat# D11347 |
| Vectashield | Vector Laboratories | Cat# H-1000-10 |
| Latex beads | Thermo Fisher Scientific | Cat# F8801 |
| BCA protein assay kit | Thermo Fisher Scientific | Cat# 23225 |
| Glucose assay kit | Sigma | Cat# GAGO20 |
| Triglyceride assay kit | Sigma | Cat# T2449 |
| Sucrose | Qualigens | Cat# Q15925 |
| NGS | Jackson Immuno Research | Cat# 005 000-121 |
| Trizol | Life Technologies | Cat# 15596018 |
| SuperScript II Reverse Transcriptase kit | Invitrogen | Cat# 18064014 |
| SYBR Green Master Mix | Applied Biosystems | Cat# A5741 |
| LCMS water | Fisher scientific | Cat# W6-4 |
| OBHA | Sigma | Cat# B22984 |
| EDC | Sigma | Cat# 03450 |
| Pyridine buffer | Sigma | Cat# 270407 |
| U13C Pyruvate | Cambridge Isotope Laboratories | Cat# CLM 2440-0.5 |
| Methanol | Fisher scientific | Cat# A456-4 |
| **Software** | | |
| Fiji/ImageJ | National Institutes of Health | https://ImageJ.nih.gov/ij/ |
| Office Excel Power Point 2016 | Microsoft | N/A |

| Reagent/resource | Reference or source | Identifier or catalog number |
|---|---|---|
| Graphpad Prism 10 | GraphPad Software Inc. | https://www.graphpad.com |
| Adobe Photoshop 2025 | Adobe Systems, San Jose, CA | https://www.adobe.com |
| Flybase | Thurmond et al, 2019 | https://flybase.org |
| BDSC database | Bloomington Drosophila Stock Center | https://bdsc.indiana.edu |
| VDRC database | Vienna Drosophila Resource Center | https://stockcenter.vdrc.at |
| **Other** | | |
| Olympus FV3000 5-Laser confocal microscope | Olympus Corporation | https://www.olympus-lifescience.com |
| Leica MZ10 F modular stereo microscope | Leica Microsystems | https://www.leica-microsystems.com |
| Sciex QTRAP 5500 LC–MS/MS system | Sciex | https://sciex.com |
| MultiQuant Software | Sciex | https://sciex.com |

## Drosophila genetics

Flies were raised on standard cornmeal medium (5% sucrose) at 25 °C. For high sugar diet, the sugar content was increased fivefold to 25% sucrose. The *RNAi* lines were obtained either from Bloomington Drosophila Stock Center (BDSC, Bloomington, IN) or Vienna Drosophila Research Centre (VDRC). The Gal4 line used was $Hml^\Delta>UAS\ GFP$ (Sinenko and Mathey-Prevot, 2004) and $w^{1118}$ flies were used as controls. All genetic crosses were set up at 25 °C and then transferred to 29 °C where they were grown until analysis either as larvae or as adults. See reagents and tools for a complete list of genes and their BDSC or VDRC stock numbers.

## High sugar diet exposure and genetic screen

We utilized two different dietary regimes of high sugar diet (HSD). For the short-term 4 hr.HSD regime, $Hml^\Delta>GFP/w^{1118}/RNAi$ feeding third-instar larvae (72 hr. AEL) reared on regular food (RF, containing 5% sucrose) were transferred to HSD (containing 25% sucrose) where they were allowed to feed for a brief period of four hours only. For Ct.HSD regime, $Hml^\Delta>GFP/w^{1118}/RNAi$ embryos were collected on RF and transferred to HSD (containing 25% sucrose). The larvae were reared at 29 °C until feeding the 3rd instar stage, following which they were processed for experiments related to immune cells and until eclosion for experiments related to adult body size.

Setting up of the *RNAi* screen was conducted with a total of 1052 *RNAi* strains which were specifically expressed in Hml+ differentiating immune cells of the *Drosophila* larvae using the $Hml^\Delta$ GAL4,UAS-2xEGFP (BDSC#30140) as the driver line (Sinenko and Mathey-Prevot, 2004). These *RNAi* strains were VDRC and BDSC TRiP lines maintained at National Centre for Biological Sciences (NCBS), Bangalore fly facility. Importantly, the majority of these lines from the facility have been used in multiple studies at NCBS and thus validated (Agrawal et al, 2013; Mishra et al, 2024; Janardan et al, 2020).

## Immune cell counts

For quantification of sessile and circulating immune cells, protocol described by (Petraki et al, 2015) was used to isolate the two immune cell populations. Briefly, three feeding third-instar larvae were allowed to bleed for a few seconds in PBS following an incision at both the larval posterior and anterior ends. After the release of the circulating immune cells, the same larvae were transferred to another well and sessile immune cells attached to the larval cuticle released by a process of scraping and/or jabbing. For quantifying total immune cells, there was no separation of circulating and sessile immune cells. Images were acquired with five fields per sample at ×20 magnification. For cell counting, a particle analyzer in ImageJ was used with size range of 2 infinity. For cell clusters typically counted as one by the software, the number of cells in those clusters was estimated by manual counting. The counting was done for DAPI-positive (representing total blood cells). These were then classified for expression of GFP in them as a readout of $Hml^\Delta GAL4, UAS-2xEGFP$ transgene expression in them. Thus, in the field of view, the total number of blood cells were distributed into HmlGFP+DAPI+ (Hml +), or HmlGFP-DAPI+ (Hml-) cells. Counting assays were performed in at least two wells per experiment and independently repeated at least three times. The cell numbers obtained were quantified per larva and represented as the number of immune cells per square millimeter $(mm^2)$.

## Immunohistochemistry and staining

For all other experiments except the cell count assay, total immune cells comprising of circulating and sessile pool were analyzed. Immune cells were bled and allowed to settle for 20 min in a humid chamber. Cells were then fixed with 4% formaldehyde in PBS for 10 min and washed twice (10 min each wash) with 0.3%PBT (0.3% Triton X in 1 × PBS) for permeabilization and were further blocked in 5% normal goat serum (NGS, Jackson ImmunoResearch, 005 000-121), for 40 min at room temperature. Cells were next incubated in the respective primary antibodies with appropriate dilution in 5% NGS overnight at 4 °C. After primary antibody incubation, Cells were washed twice in 0.3% PBT for 10 min each. This was followed by incubation of cells in respective secondary antibodies for 2 h. at room temperature. After secondary antibody incubation, cells were washed in 0.3% PBT for 10 min following a DAPI + 0.3% PBT wash for 10 min. Excess DAPI was washed off by a wash of 0.3% PBT for 10 min. Cells were mounted in Vectashield (Vector Laboratories) and then imaged using confocal microscopy (Olympus FV3000). Primary antibodies used were rabbit αACC (1:1000, Jacques Montagne, I2BC, France), rabbit αcrq (1:100, The Scripps Research Institute, La Jolla, USA). The secondary antibody Alexa Fluor 546 (Invitrogen) was used at 1:500 dilution. Nuclei were visualized using DAPI (Sigma).

For Nile Red staining, formaldehyde fixed cells were incubated in 1:1000 solution of 0.02% Nile Red (Sigma-Cat. No. N3013) for 20 min, washed and mounted similarly. Images were acquired on Olympus FV3000 confocal microscope with a step size of 1 μm at ×40 or ×60 magnification.

For phalloidin staining, cells were first permeabilized with 0.1% Triton X 100 in PBS (PBST) for 5 min and then incubated for 2 h with Atto 565 Phalloidin (Sigma-Aldrich # 94072) diluted 1:100 in 1 × PBS.

Phalloidin staining was used to assess cell morphology and filopodia length and number. Specifically, for measuring filopodia length, it was done as described in (Hao et al, 2018). Briefly, the line tool on ImageJ was used to draw a line over a filopodium from its tip to cell body with extensions greater than 0.5 μm being classified as filopodia.

For tissue staining, whole larvae were inverted and fixed at 4% formaldehyde for 40 min and then three washes with 1×PBS and processed subsequently for specific staining.

ROS staining were done as described in (Owusu-Ansah and Banerjee 2009). Larval immune cells were stained with 1:1000 DHE (Dihydroethidium) (Invitrogen, Molecular Probes, D11347) dissolved in 1 × PBS for 15 min in the dark. Immune cells were washed in 1 × PBS twice and fixed with 4% formaldehyde for 5 min at room temperature in the dark. After this, 1 × PBS wash was given to the immune cells, and this step was repeated twice and then Vectashield (Vector Laboratories) was added. The immune cells were imaged immediately.

## Immune cell phagocytosis assay

Immune cells bled in PBS were treated with 0.1 μm latex beads (ThermoFischer Scientific #F8801) for 15 min and washed three times with PBS to remove the excess free beads. Cells were then fixed with 4% formaldehyde in PBS for 10 min, washed with PBS and mounted in Vectashield with DAPI (Vector Laboratories) for imaging. For measuring phagocytic capacity, phagocytic index was measured as the number of engulfed latex beads per immune cell (Hao et al, 2018).

## Image analysis and quantification of expression intensities

ImageJ software was used for analysis. For all images, across all experiments, with staining in circulating immune cells, the quantification of the expression pattern or intensities was done in the following manner. At least two wells per experiment was analyzed. Each well had immune cells obtained from a maximum of five larvae. Five to six images were captured for each well at ×60 magnification, and the staining was assessed for 5–6 cells/field. The analyses were carried out for at least 60–70 cells per experiment, and this was repeated independently at least three to five times. The quantifications shown in the graphs represent the average expression from these cells across batches. Images were assembled in Adobe Photoshop 2025.

## Immune cell biochemical assays: triacylglycerol and glucose measurements

TAG and glucose measurements were done as shown in (P et al, 2020). Briefly, immune cells bled from at least fifteen larvae per experiment were collected in PBS, followed by centrifugation at 1000 rpm to pellet the cells. To the pellet, 0.05% 1 × PBST (Tween 20) was added and vortexed intermittently by keeping it on ice. Protein levels of immune cells were estimated using BCA protein assay kit (ThermoFischer Scientific #23225). For measuring glucose and TAG levels, immune cell samples were first heat-inactivated at 70 °C for 10 min and then subjected to metabolite analysis using GOD POD kit (Sigma#GAGO20) and Triglyceride assay kit (Sigma#T2449), respectively. Assays were performed on Varioskan LUX Multimode Microplate Reader and metabolite levels in each

sample were normalized to total protein levels. At least two to three biological replicates were used, and the assays were performed in at least five independent experiments (see Legends for "n", total number of larvae for the assays).

## RNA isolation, bulk RNA sequencing, and real-time PCR

Immune cells from thirty to forty feeding 3rd instar larvae fed on RF, 4 hr.HSD and Ct.HSD were collected in PBS on ice and stored at −80 °C. Total RNA was extracted using Trizol (Life Technologies) followed by assessment of RNA integrity (>7) and purity using an Agilent 2100 Bioanalyzer. Illumina Hi seq kits were used to construct sequencing libraries following standard protocol, and 100 bp single-end reads were generated at Sequencing facility, NCBS (Bengaluru, India).

For Real-Time PCR, RNA was first converted to first-strand cDNA using the SuperScript II Reverse Transcriptase kit (Invitrogen#18064014) following the manufacturer's instructions. Real-Time PCR was performed in QuantStudio 5 Real Time PCR System (Applied Biosystems) using SYBR Green Master Mix (Applied Biosystems#A5741) and gene-specific primers. The primers designed using IDT's Primer Quest Tool are listed in the Reagents and Tools Table. Relative quantification of transcript levels was achieved using the Comparative Ct method (delta delta Ct) using Rp49 as endogenous control. At least three biological replicates were used and repeated three times (see Legends for "n", total number of larvae for qPCR).

## RNA seq data analysis

Post sequencing, 30–40 million single-end reads were obtained. FastQC v0.11.5 was used to perform the initial quality check. Adapters were trimmed from the reads using cutadapt v1.8.3 (-a AGATCGGAAGAGCACACGTCTGAACTCCAGTCA). The trimmed reads were mapped to the Drosophila genome (*Drosophila melanogaster*. BDGP6.22) using Hisat2 v2.1.0. Read counting was done using featureCounts v2.0.0. DESeq2 v1.40.1 was used to perform the read count normalization and differential expression analysis (Ge et al, 2020; Kanehisa and Goto, 2000; Kim et al, 2015; Liao et al, 2014; Love et al, 2014; Martin, 2011). Genes that showed a fold change of at least 2 (up or down), with an adjusted *P* value of less than 0.05, were considered as differentially expressed for further analysis. Gene ontology and KEGG pathway enrichment analysis of the differentially expressed genes was done on the ShinyGO v0.60 webserver. Genes that are associated with each metabolic pathway considered here, were retrieved from the KEGG database (http://www.genome.jp/).

## Adult fly size and wing analysis

The adult fly progeny of the tested crosses viz *RNAi* lines with the immune cell-specific driver, *Hml*^Δ^>GFP were collected after the eclosion on high sugar diet and kept on 29 °C for 1–2 days in normal food vials for acclimatization. Both females and males were kept together. After 2 days, the male and female flies were separated, and flies were grouped for imaging for the body size. Each fly was kept in a lateral position and fly wings were moved backward to expose the body. The length from the anterior end of a head to the posterior end of the abdomen in the flies was measured (Lee et al, 2004, 2008). For wing area quantification, the right wing

of each individual fly separately for male and female was plucked with forceps and mounted on a glass slide. The distilled water was used to mount the wing on a glass slide for proper orientation. The slides were covered with a coverslip and sealed with nail paint. The images of the wings were captured with the Leica MZ 10 F modular stereo microscope and LASX software. Fiji ImageJ was used to quantify the wing phenotype for wing area using the polygon section tool in the software. The scale was calibrated by converting pixel dimensions to millimeters (mm). The hinge region of the wing was excluded during boundary marking. Wing span and adult fly length were measured in more than 50 animals.

## Metabolite extraction and derivatization

For metabolite extraction, blood cells from five feeding 3rd instar larvae per replicate were extracted and 200 μl of 80% ice-cold Methanol was added. After this, 100 μl of LC/MS grade water was added and the samples were incubated on ice for 30 min. Then 200 μl chloroform was added and samples were vortexed for 30 s and centrifuged at 13,000 RPM for 10 min at 4 °C. The upper phase was transferred into a fresh tube, dried down in a Vacufuge plus speed vac at room temperature, and derivatized further with OBHA/EDC for metabolite analysis. The interphase was taken for protein estimation for normalization purpose. Proteins were resuspended in 5% SDS and heated at 37 °C for 30 min. The protein concentration was determined using the Pierce BCA Protein Assay Kit Assay (ThermoFisher). For steady state analysis, the metabolite levels were normalized by per sample per total protein amount in microgram (μg).

For derivatization of metabolites (Walvekar et al, 2018; Tan et al, 2014), the dried samples were dissolved in 50 μl of LC/MS grade water, and 50 μl of 1 M EDC (in Pyridine buffer) was added. Samples were kept on a thermomixer for 10 min. at room temperature and 100 μl of 0.5 M OBHA (in Pyridine buffer) was added. The samples were incubated again for 1.5 h on the thermomixer at 25 °C, and metabolites were extracted by adding 300 μl of ethyl acetate, and this step was repeated three times. Samples were dried down in a Vacufuge plus speed vac at room temperature and stored at −80 °C until run for LC/MS analysis. A minimum of three biological replicates were used per condition.

## 13 C labelling and stable isotope tracer analysis

For isotopomer tracer analysis, five feeding 3rd instar (WI) larvae were washed twice in PBS and immune cells were extracted. Blood cells were incubated in 10 mM of U13C Pyruvate in 1 × PBS (Cambridge Isotope Laboratories, CLM 2440-0.5) for 30 min. Blood cells were centrifuged down at 13,000 RPM for 10 min. and 200 μl of 80% ice-cold methanol was added to each sample and stored at −80 °C. Samples were further processed for metabolite extraction as done for steady-state analysis.

## Liquid chromatography mass spectrometry (LC/MS) analysis

The metabolite extract was separated using a Waters XBridge C18 Column (2.1 mm, 100 mm, 3.5 mm) coupresulted in an Agilent QQQ 6470 system. The autosampler and column oven were held at 4 °C and 25 °C, respectively. The column was used with buffer A (Water and 0.1% Formic Acid) and buffer B (100% acetonitrile and

0.1% Formic Acid). The chromatographic gradient was run at a flow rate of 0.300 ml/min as follows: 0 min: gradient 10% B; 0.50 min: gradient 10% B; 8 min: gradient 100% B; 10 min: gradient 10% B; 11 min: gradient at 10% B. and 16 min: gradient held at 10% B. The mass spectrometer was operated in MRM, positive ion mode. Mass spectrometry detection was carried out on a QQQ Agilent 6470 system with ESI source. For metabolite quantification, Peak areas were processed using MassHunter workstation (Agilent). Microsoft Excel 2016 and GraphPad Prism 9 software were used for statistical analysis. Q1/Q3 transitions and retention times (RT) for all analyzed metabolites are listed in Appendix Table S5.

### Graphics

Figure panels were prepared using Adobe Photoshop 2025, and schematics were created using BioRender.com and Microsoft PowerPoint 2016.

### Comparisons, sample size, and statistical analyses

For all measurement and quantifications shown in the graphs, that include adult growth measurement, larval sizes, immune cell numbers, wing disc areas and fat body mass, across all genetic combinations, the comparisons were made against their respective background *RNAi* control line (Krejčová et al, 2019, 2023). The respective comparisons made are shown in each representative graphs across all figure panels.

In our analysis, *n* denotes the total number of samples analyzed, and *N* refers to the number of independent biological replicates. For all fly growth plots depicting wing area and body length, each data point corresponds to an individual animal. Data from all three experimental batches were combined and presented as fold change plots, with statistical significance assessed using the Mann–Whitney test. Sample sizes were selected based on the assay's sensitivity to detect relevant differences. All statistical analyses were conducted using GraphPad Prism 10, with data calculations performed in Microsoft Excel 2016. Comparisons between groups with normal distributions were analyzed using unpaired, two-tailed Student's *t* test with Welch's correction for unequal variances, while Mann–Whitney tests were used for comparisons of medians. Two-way ANOVA with main effects analysis, followed by either Dunnett's or Sidak's multiple comparisons tests, was applied where appropriate. For image presentation, confocal images were uniformly adjusted for levels and channels in Adobe Photoshop, solely for visualization; all quantitative image analyses were based on unprocessed raw data or maximum intensity projections. Asterisks in figures indicate statistical significance as follows: ($*P < 0.05$; $**P < 0.01$; $***P < 0.001$; $****P < 0.0001$). Exact $P$ values are provided in the figure legends.

## Data availability

All raw RNA sequencing (RNA seq) reads associated with the study are available from the NCBI SRA (Accession PRJNA1090274) and are available at the following link: https://dataview.ncbi.nlm.nih.gov/object/PRJNA1090274?reviewer=tumbh80rnl0p2hcum3ivfspvts.

The source data of this paper are collected in the following database record: biostudies:S-SCDT-10_1038-S44319-025-00574-7.

## Peer review information

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

## Acknowledgements

We thank Prof. Jacques Montagne for the Acetyl CoA-Carboxylase (ACC) antibody, Prof. Nathalie Franc for the Croquemort (Crq) antibody, the Bloomington Drosophila Stock Center (BDSC) and Vienna Drosophila Resource Center (VDRC) for fly stocks, the FlyBase for the literature. We acknowledge the National Centre for Biological Sciences (NCBS), Centre for Cellular and Molecular Platforms (C CAMP) for Central Imaging & Flow Cytometry Facility (CIFF) and the fly facility. We thank the metabolomics facility at MPF, Bindley Bioscience Center, and Dr. Vikki Weake, Purdue University for help with metabolomic experiments. Owing to space limitations, we apologize to our colleagues whose work is not cited. This study was supported by Department of Science and Technology—CRG (Grant number CRG/2021/002815), USIAS Indo French grant awarded to TM and AG, CNRS International Research Project entitled "MACHUB" jointly awarded to AG and TM, Human Frontier Science Program (HFSP, grant number RGP018/2023) grant awarded to TM, and DBT/Wellcome Trust Senior Research Fellowship (Grant number IA/S/22/1/506259) awarded to TM. This work was also supported by the DBT/Wellcome Trust India Alliance Fellowship [Grant number IA/E/18/1/504327] awarded to AM. SN is a Graduate Student at inStem, in the Mukherjee lab, and is supported by Council of Scientific & Industrial Research (CSIR)-Fellowship (1121732485). MG is a Graduate Student at inStem, in the Mukherjee lab, and is supported by DST INSPIRE (DST/INSPIRE/.03/2018/000052) and, SERB OVDF fellowship (SB/S9/Z-03/2017-XVI(2020-21)) facilitated the work done at Purdue University.

## Author contributions

**Anusree Mahanta**: Conceptualization; Data curation; Formal analysis; Validation; Methodology; Writing—original draft. **Sajad Ahmad Najar**: Conceptualization; Data curation; Formal analysis; Supervision; Validation; Investigation; Visualization; Methodology; Writing—original draft; Project administration; Writing—review and editing. **Nivedita Hariharan**: Data curation; Formal analysis; Validation; Investigation. **Ajit Bhowmick**: Data curation; Formal analysis; Validation; Investigation; Methodology. **Syed Iqra Rizvi**: Data curation; Formal analysis; Validation; Investigation. **Manisha Goyal**: Data curation; Formal analysis; Validation; Investigation. **Preethi Parupalli**: Data curation; Formal analysis; Validation; Investigation. **Ramaswamy Subramanian**: Data curation; Supervision; Project administration. **Angela Giangrande**: Conceptualization; Funding acquisition; Project administration. **Dasaradhi Palakodeti**: Conceptualization; Supervision; Project administration. **Tina Mukherjee**: Conceptualization; Resources; Data curation; Software; Formal analysis; Supervision; Funding acquisition; Validation; Investigation; Visualization; Methodology; Writing—original draft; Project administration; Writing—review and editing.

Source data underlying figure panels in this paper may have individual authorship assigned. Where available, figure panel/source data authorship is listed in the following database record: biostudies:S-SCDT-10_1038-S44319-025-00574-7.

## Disclosure and competing interests statement

The authors declare no competing interests.

# Expanded View Figures

**Figure EV1. Dietary sugar stress affects larval macrophage physiology.**

(A–B''') HSD alters Hml+ immune cell numbers. Representative images of sessile (A–A''') and circulatory immune cells (B–B'''), on RF (A, B), 4 hr.HSD (A', B') and Ct.HSD (A'', B''). Compared to sessile immune cells in (A) $Hml^\Delta$>GFP/$w^{1118}$ (Control (RF), (A') $Hml^\Delta$>GFP/$w^{1118}$ (4 hr.HSD) did not show any dramatic change in their numbers, but (A'') $Hml^\Delta$>GFP/$w^{1118}$ (Ct.HSD) larvae show significant reduction in Hml+ sessile population. See quantifications in (A'''). Circulating cell numbers in (B') $Hml^\Delta$>GFP/$w^{1118}$ (4 hr.HSD) and (B'') $Hml^\Delta$>GFP/$w^{1118}$ (Ct.HSD) showed no striking difference compared to the Control (B). See quantifications in (B'''). (A''') Quantification of sessile Hml+ immune cell numbers in $Hml^\Delta$>GFP/$w^{1118}$ (Control, RF, $N = 3$, $n = 18$), $Hml^\Delta$>GFP/$w^{1118}$ (4 hr.HSD, $N = 3$, $n = 18$, $P = 0.5504$) and Hml$\Delta$>GFP/w1118 (Ct.HSD, $N = 3$, $n = 18$, $P < 0.0001$). (B''') Quantification of circulatory Hml+ immune cell numbers in $Hml^\Delta$>GFP/$w^{1118}$ (Control, RF, $N = 3$, $n = 18$), $Hml^\Delta$>GFP/$w^{1118}$ (4 hr.HSD, $N = 3$, $n = 18$, $P = 0.1771$) and $Hml^\Delta$>GFP/$w^{1118}$ (Ct.HSD, $N = 3$, $n = 18$, $P = 0.9668$). (C–C'') Representative images of immune cells to assess ROS levels. (C) ROS level in immune cell of $Hml^\Delta$>GFP/$w^{1118}$ (Control, RF). (C') $Hml^\Delta$>GFP/$w^{1118}$ (4 hr.HSD) and (C'') $Hml^\Delta$>GFP/$w^{1118}$ (Ct.HSD) show no change in ROS levels as compared to Control (C). (D–D'') Representative images of immune cells with Nile red staining to assess for lipid droplet accumulation. (D) $Hml^\Delta$>GFP/$w^{1118}$ (Control, RF). (D') $Hml^\Delta$>GFP/$w^{1118}$ (4 hr.HSD) and (D'') $Hml^\Delta$>GFP/$w^{1118}$ (Ct.HSD) show gradual increase in immune cell lipid content compared to Control (D). (E–E'') Representative images of immune cells to assess phagocytosis through bead uptake assay 15 min post incubation. (E) $Hml^\Delta$>GFP/$w^{1118}$ (Control, RF). (E') $Hml^\Delta$>GFP/$w^{1118}$ (4 hr.HSD) and (E'') $Hml^\Delta$>GFP/$w^{1118}$ (Ct.HSD) show reduction in number of internalized beads when compared to Control (E). (F–F'') Representative confocal images of immune cells assessed for cellular morphology. (F) $Hml^\Delta$>GFP/$w^{1118}$ (Control, RF). (F') $Hml^\Delta$>GFP/$w^{1118}$ (4 hr.HSD) and (F'') $Hml^\Delta$>GFP/$w^{1118}$ (Ct.HSD) show reduction both in number as well as in length of filopodia compared to Control (F). (G–G'') Representative images of immune cells assessed for lipid droplets with UAS-LSD2-GFP reporter line. (G) actin-GAL4/UAS-LSD2-GFP (Control, RF). (G') actin GAL4/UAS-LSD2-GFP (4 hr.HSD) and (G'') actin-GAL4/UAS-LSD2-GFP (Ct.HSD) show gradual increase in immune cell lipid droplets (green) compared to Control (G). Data information: DNA is stained with DAPI (blue), immune cells are marked in green (Hml>UAS-GFP). (A–B'') scale bar is 100 μm and (C–G'') scale bar is 5 μm. Comparisons for significance are with regular food (RF) conditions and asterisks mark statistically significant differences (*$P < 0.05$; **$P < 0.01$; ***$P < 0.001$; ****$P < 0.0001$). The statistical analysis applied for (A''', B''') is two-way ANOVA with Dunnett's multiple comparison test. RF, 4 hr.HSD and Ct.HSD indicate conditions of larvae fed on regular food (RF), 4 hour high sugar diet (4 hr.HSD) and constitutive high sugar diet (Ct.HSD), respectively. $N$ indicates the number of independent biological replicates, and n refers to the total number of animals analyzed. See methods for further details on larval numbers and sample analysis for each of the experiments. In bar graphs, data are presented as mean ± SD. Source data are available online for this figure.

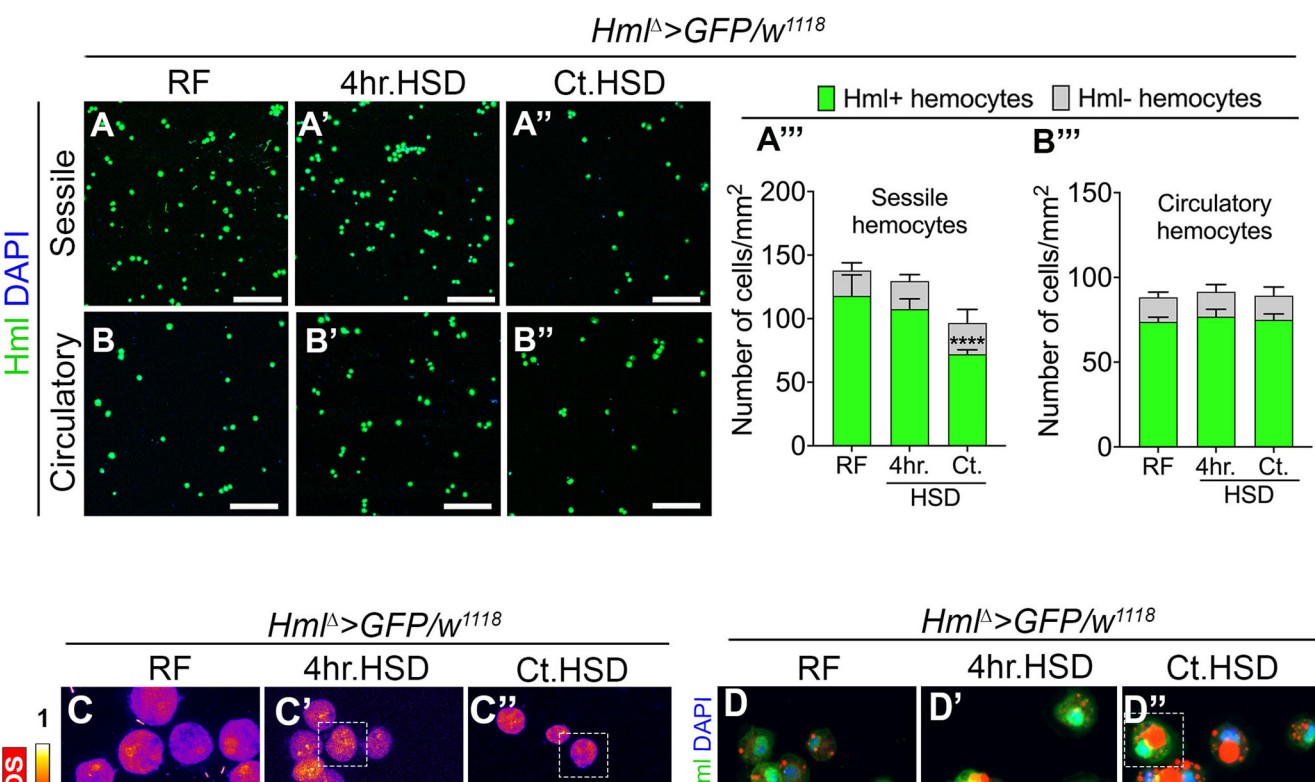

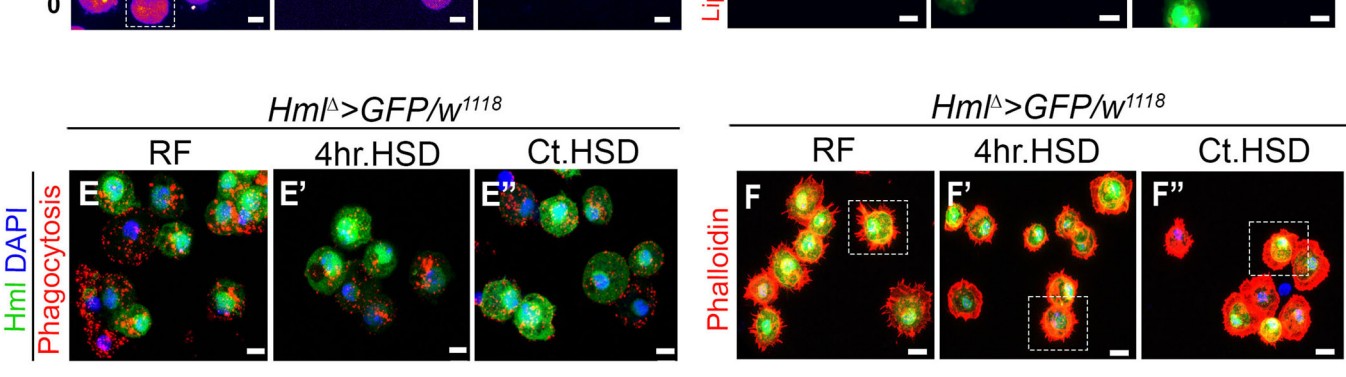

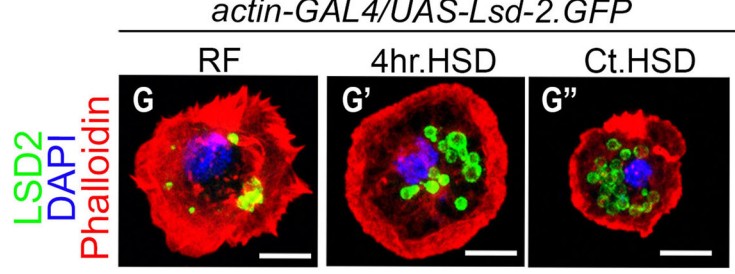

## Genome wide transcriptome of the whole animal raised on HSD

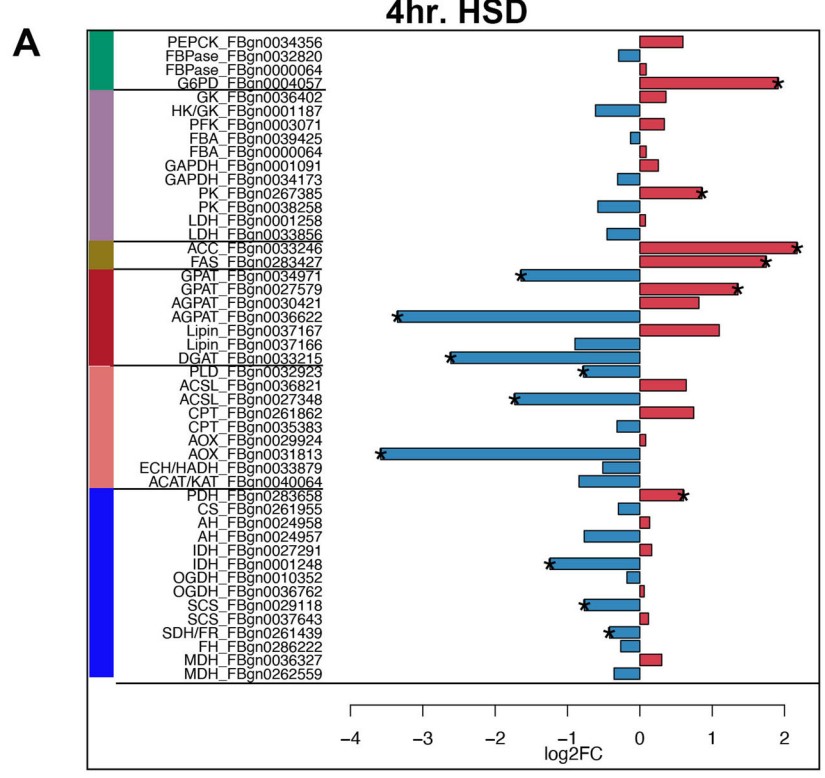

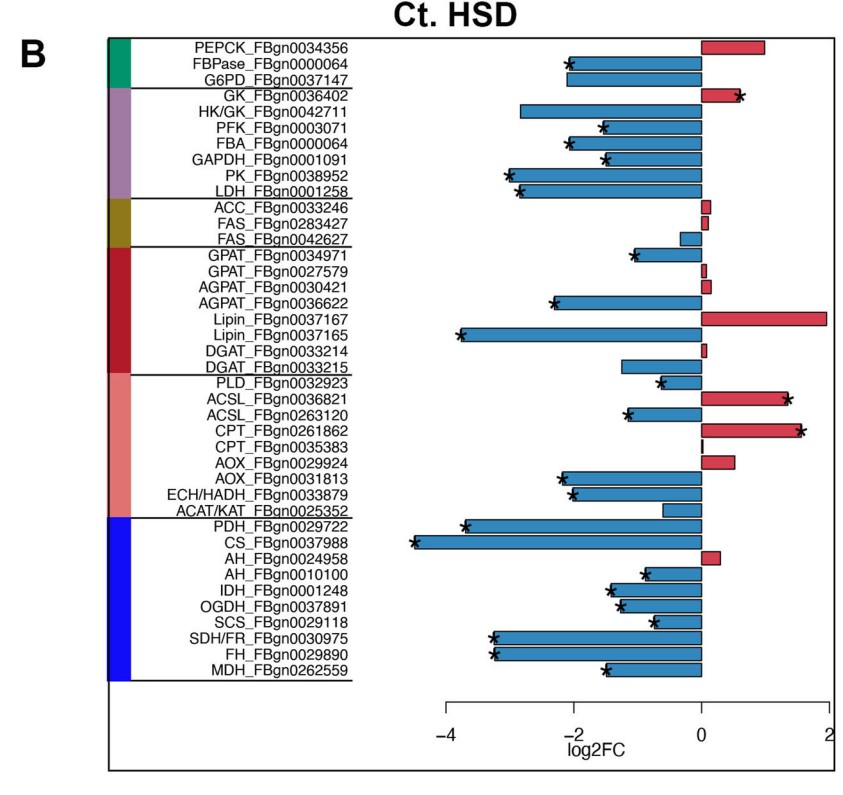

Gluconeogenesis     Glycolysis     Denovo lipogenesis     TAG synthesis     beta-oxidation     TCA

◀ **Figure EV2. High sugar diet dampens metabolic events in whole larvae.**

(A, B) Bar plots of upregulated (in red) and downregulated genes (in blue) of different metabolic pathways in whole animal (larvae) raised on 4 hr.HSD and Ct.HSD, respectively. Metabolic genes are downregulated in 4 hr.HSD whole larvae and this is sustained in long-term Ct.HSD animals. Source data are available online for this figure.

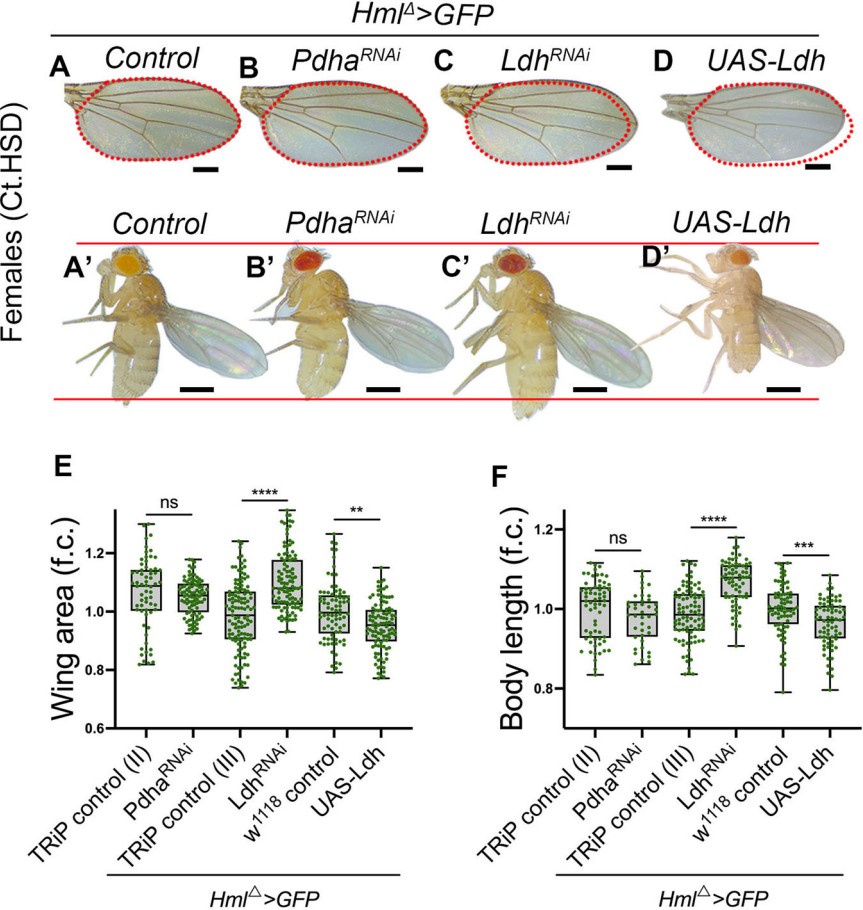

**Figure EV3. Glycolytic state in immune cells represses growth on HSD.**

(A–D′) Modulating larval immune cell TCA and glycolytic activity to show adult size change. Representative images of wings (A–D) and adult females (A′–D′) showing size phenotype on Ct.HSD from respective genetic backgrounds. Compared to (A, A′) Ct.HSD Control (*Hml^Δ>GFP/w^1118*), (B, B′) expressing *Pdha^RNAi* (*Hml^Δ>GFP/Pdha^RNAi*) in immune cells to reduce TCA activity did not show any size change. (C, C′) Downregulating immune cell glycolytic activity by expressing *Ldh^RNAi* (*Hml^Δ>GFP/Ldh^RNAi*) causes increase in size. (D, D′) Overexpression of Ldh (*Hml^Δ>GFP/UAS-Ldh*) showed size reduction. (E) Quantification of wing area in *Hml^Δ>GFP/Pdha^RNAi* (Ct.HSD, N = 3, n = 91, P = 0.1102) in comparison to *Hml^Δ>GFP/TRiP (II) control* (Ct.HSD, N = 3, n = 64), *Hml^Δ>GFP/Ldh^RNAi* (Ct.HSD, N = 3, n = 106, P < 0.0001) in comparison to *Hml^Δ>GFP/TRiP (III) control* (Ct.HSD, N = 3, n = 121) and *Hml^Δ>GFP/UAS-Ldh* (Ct.HSD, N = 3, n = 104, P = 0.0015) in comparison to *Hml^Δ>GFP/w^1118* control (Ct.HSD, N = 3, n = 81). (F) Quantification of body length in *Hml^Δ>GFP/Pdha^RNAi* (Ct.HSD, N = 3, n = 40, P = 0.0565) in comparison to *Hml^Δ>GFP/TRiP (II) control* (Ct.HSD, N = 3, n = 65) and *Hml^Δ>GFP/Ldh^RNAi* (Ct.HSD, N = 3, n = 70, P < 0.0001) in comparison to *Hml^Δ>GFP/TRiP (III) control* (Ct.HSD, N = 3, n = 89) and *Hml^Δ>GFP/UAS-Ldh* (Ct.HSD, N = 3, n = 73, P = 0.0003) in comparison to *Hml^Δ>GFP/w^1118* control (Ct.HSD, N = 3, n = 75). Data information: Scale bar: 0.5 mm for flies and 0.25 mm for wings. In quantification graphs, shown in panel (E, F) each dot represents an animal. Comparison for significance is with respect to respective background control on Ct.HSD. Asterisks mark statistically significant differences (*P < 0.05; **P < 0.01; ***P < 0.001; ****P < 0.0001). The statistical analysis applied for (E, F) is Mann–Whitney test. N indicates the number of independent biological replicates, and n refers to the total number of animals analyzed. Only right wing from each adult fly was selected for quantification. The differences in wing areas or fly body lengths in panels is indicated with a red dotted line or two horizontal red lines that highlight changes across genotypes. RF and Ct.HSD correspond to regular food and constitutive high sugar diet respectively. Box plots show the median (center line), 25th–75th percentiles (bounds of box), and whiskers extending to the minimum and maximum values; all individual data points are shown. Source data are available online for this figure.

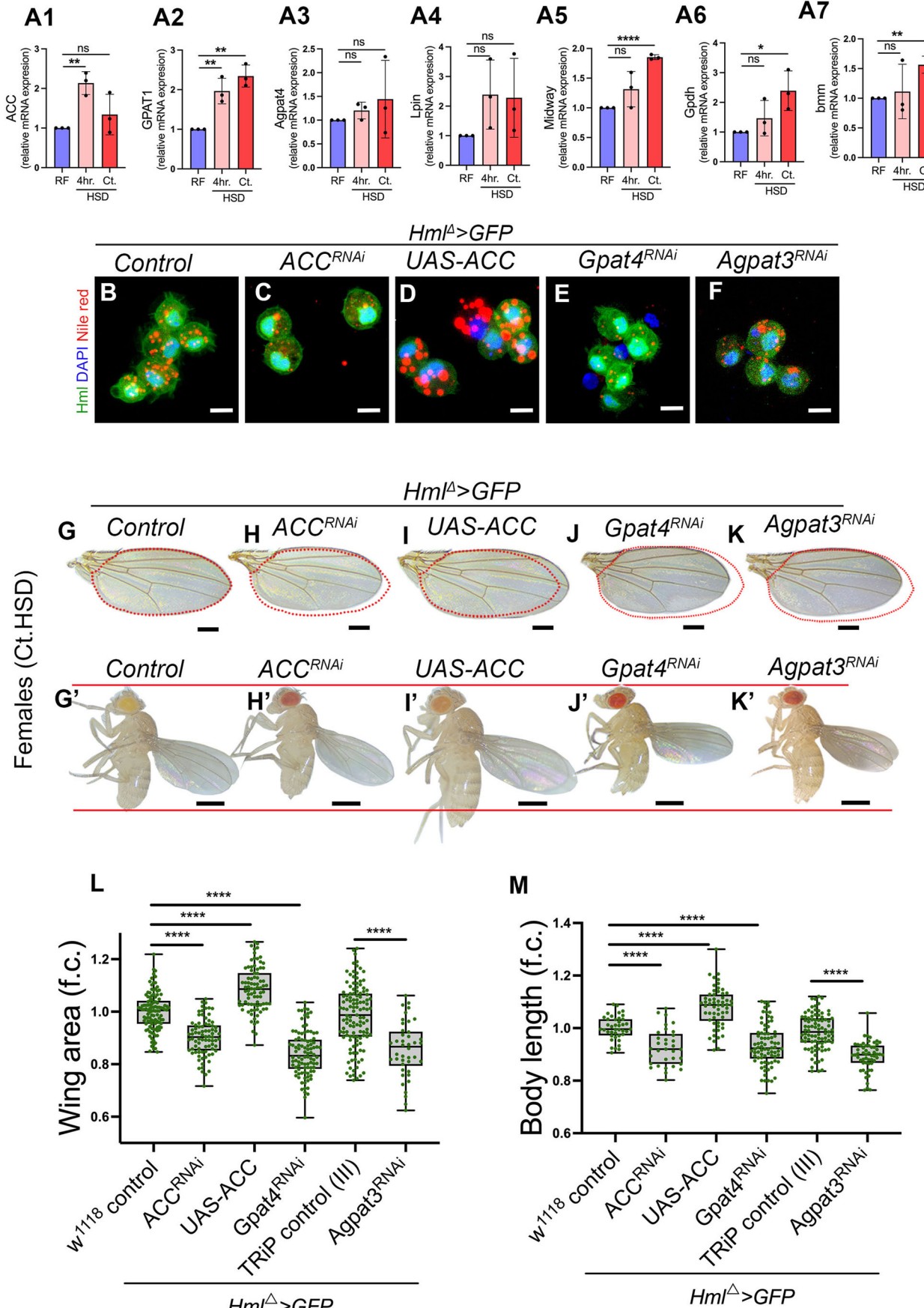

◀ **Figure EV4. Immune cell lipid homeostasis and systemic growth regulation on HSD.**

(A1–A7) Relative expression of immune-specific lipid metabolism genes in *Hml^Δ^>GFP/w^1118^* larvae exposed to 4 hr.HSD and Ct.HSD, measured by RT-PCR. (A1) ACC show a significant upregulation at 4 hr.HSD ($N = 3$, $n = 105$, $P = 0.0025$) but not at Ct.HSD ($N = 3$, $n = 120$, $P = 0.3125$). (A2) GPAT1 was upregulated at 4 hr.HSD ($N = 3$, $n = 120$, $P = 0.0066$) and Ct.HSD ($N = 3$, $n = 120$, $P = 0.0011$) as well. (A3) Agpat4 and (A4) Lpin did not show any change at 4 hr.HSD and Ct.HSD. (A5) Midway showed no change at 4 hr.HSD ($N = 3$, $n = 120$, $P = 0.1424$) however showed upregulation at Ct.HSD ($N = 3$, $n = 120$, $P < 0.0001$). (A6) Gpdh1 showed no change at 4 hr.HSD ($N = 3$, $n = 120$, $P = 0.2473$) however it was upregulated in Ct.HSD ($N = 3$, $n = 120$, $P = 0.0223$). (A7) bmm show upregulation at Ct.HSD ($N = 3$, $n = 120$, $P = 0.0025$) and no change at 4 hr.HSD ($N = 3$, $n = 120$, $P = 0.6920$). All comparisons are with control *Hml^Δ^GFP > /w^1118^* on RF ($N = 3$, $n = 105$). (B–F) Representative images of immune cells stained with Nile Red (red) to visualize lipid droplets in respective genotypes under Ct.HSD. (B) Ct.HSD Control. (C) Immune-specific knockdown of ACC (*Hml^Δ^>GFP/ACC^RNAi^*) reduces de novo lipid synthesis and results in fewer lipid droplets compared to control. (D) Immune-specific overexpression of ACC (*Hml^Δ^>GFP/UAS-ACC*) increases de novo lipid synthesis and shows more lipid droplets. (E, F) Knockdown of Gpat4 (*Hml^Δ^>GFP/Gpat4^RNAi^*) or Agpat3 (*Hml^Δ^>GFP/Agpat3^RNAi^*) reduces TAG synthesis and leads to fewer lipid droplets compared to control. (G–K') Modulating larval immune cell lipid homeostasis affects adult growth. Representative images of wings (G–K) and adult females (G'–K') showing size phenotype on Ct.HSD from respective genetic backgrounds. Compared to (G, G') Ct.HSD Control (*Hml^Δ^>GFP/w^1118^*), (H, H') loss of ACC function (*Hml^Δ^>GFP/ACC^RNAi^*) leads to growth retardation while (I, I') ACC gain of function (*Hml^Δ^>GFP/UAS-ACC*) shows growth recovery and the flies are much larger than Ct.HSD Control adults (G, G'). Similarly, loss of TAG synthesis, by blocking (I, I') Gpat4 (*Hml^Δ^>GFP/Gpat4^RNAi^*) or (K, K') Agpat3 (*Hml^Δ^>GFP/Agpat3^RNAi^*) shows reduction in animal size. (L) Quantification of wing area in *Hml^Δ^>GFP/ACC^RNAi^* ($N = 3$, $n = 81$, $P < 0.0001$), *Hml^Δ^>GFP/UAS-ACC* ($N = 3$, $n = 83$, $P < 0.0001$), *Hml^Δ^>GFP/Gpat4^RNAi^* ($N = 3$, $n = 96$, $P < 0.0001$) in comparison to Ct.HSD *control*, *Hml^Δ^>GFP/w^1118^*, $N = 3$, $n = 115$). *Hml^Δ^>GFP/Agpat3^RNAi^* ($N = 3$, $n = 42$, $P < 0.0001$ in comparison to *Hml^Δ^>GFP/TRiP (III) control* (Ct.HSD, $N = 3$, $n = 121$). (M) Quantification of body length in *Hml^Δ^>GFP/ACC^RNAi^* ($N = 3$, $n = 33$, $P < 0.0001$), *Hml^Δ^>GFP/UAS-ACC* ($N = 3$, $n = 58$, $P < 0.0001$), *Hml^Δ^>GFP/Gpat4^RNAi^* ($N = 3$, $n = 78$, $P < 0.0001$) in comparison to Ct.HSD Control, *Hml^Δ^>GFP/w^1118^*, $N = 3$, $n = 40$). *Hml^Δ^>GFP/Agpat3^RNAi^* ($N = 3$, $n = 56$, $P < 0.0001$ in comparison to *Hml^Δ^>GFP/TRiP (III) control* (Ct.HSD, $N = 3$, $n = 89$). Data information: DNA is stained with DAPI (blue), immune cells are shown in green (*Hml^Δ^>UAS-GFP*). Nile red (red) staining to mark lipids in (B–F). Scale bar: 5 µm for immune cells, 0.5 mm for flies and 0.25 mm for wings. In quantification graphs (A1–A7) each dot represents an experimental repeat and in graphs (L, M) each dot represents an animal. Except for panel (A1–A7) where comparisons are with respect to Control on RF, in all other panels comparison for significance is with respect to respective background control on Ct.HSD. Asterisks mark statistically significant differences (*$P < 0.05$; **$P < 0.01$; ***$P < 0.001$; ****$P < 0.0001$). The statistical analysis applied for (A1–A7) is unpaired *t* test, for other panels (L, M) Mann– Whitney test. *N* indicates the number of independent biological replicates, and *n* refers to the total number of animals analyzed. Only right wing from each adult fly was selected for quantification. The differences in wing areas or fly body lengths in panels is indicated with a red dotted line or two horizontal red lines that highlight changes across genotypes. RF, 4 hr.HSD and Ct.HSD correspond to regular food, 4 h high sugar diet and constitutive high sugar diet respectively. See methods for further details on larval numbers and sample analysis for each of the experiments. In bar graphs data are presented as mean ± SD. Box plots show the median (center line), 25th–75th percentiles (bounds of box), and whiskers extending to the minimum and maximum values; all individual data points are shown. Source data are available online for this figure.

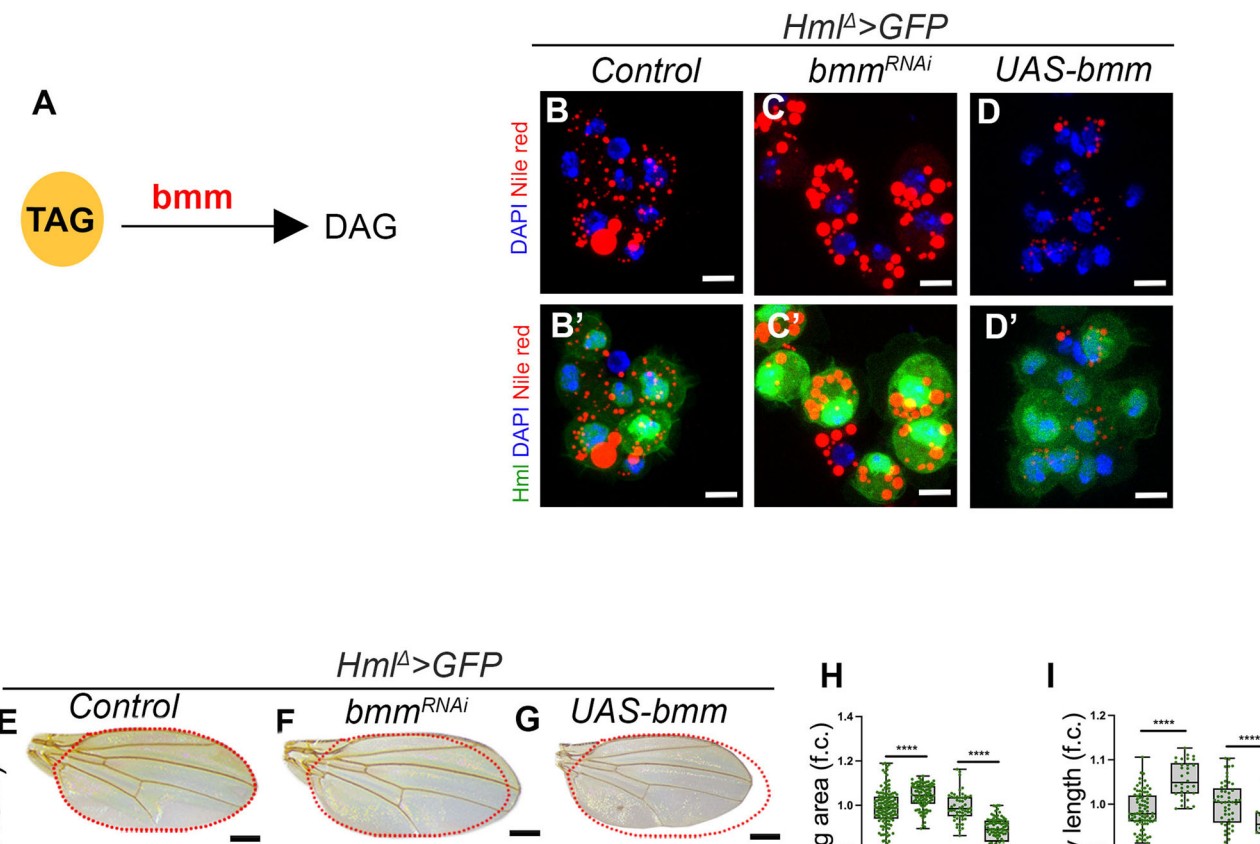

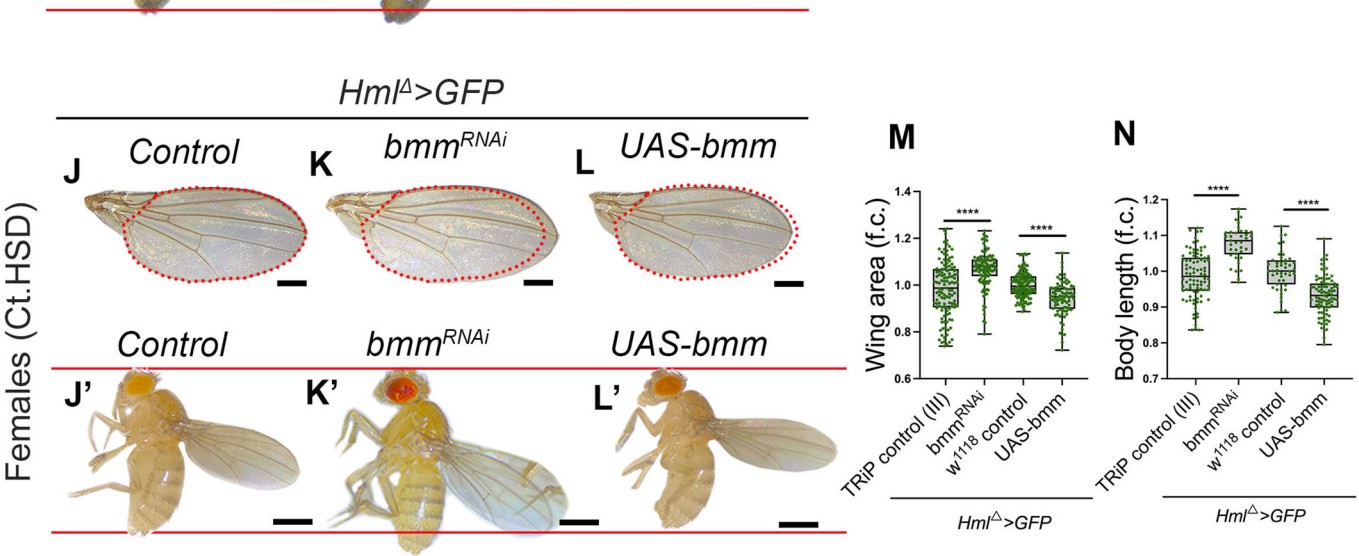

◄ **Figure EV5. Immune cell lipolytic state as inhibitor of systemic growth on HSD.**

(A) Schematic shows bmm is a lipase enzyme which breaks down triacylglycerol (TAG) into diacylglycerol (DAG). (B–D') Representative images of immune cells from Ct.HSD larvae stained with Nile Red (red) to visualize lipid droplets. (B, B') Ct.HSD Control ($Hml^\Delta$>GFP/$w^{1118}$). (C, C') Immune-specific knockdown of bmm ($Hml^\Delta$>GFP/$bmm^{RNAi}$) increases lipid droplet accumulation compared to control. (D, D') Immune-specific overexpression of bmm ($Hml^\Delta$>GFP/UAS-bmm) reduces lipid droplet levels compared to control. (E–G') Modulating larval immune cell lipolysis affects adult growth. Representative images of wings and flies of adult males (E–G') showing size phenotype on Ct.HSD from respective genetic perturbations. Compared to (E, E') Ct.HSD Control ($Hml^\Delta$>GFP/$w^{1118}$), (F, F') loss of bmm ($Hml^\Delta$>GFP/$bmm^{RNAi}$) resulted in recovery in adult fly size while (G, G') increase in bmm expression ($Hml^\Delta$>GFP/UAS-bmm) in immune cells caused a further reduction in size. (H) Quantification of wing area in (H) $Hml^\Delta$>GFP/$bmm^{RNAi}$ ($N = 3$, $n = 95$, $P < 0.0001$) in comparison to $Hml^\Delta$>GFP/TRiP (III) control (Ct.HSD, $N = 3$, $n = 131$). $Hml^\Delta$>GFP/UAS- bmm ($N = 3$, $n = 89$, $P < 0.0001$ in comparison to Ct.HSD Control, $Hml^\Delta$>GFP/$w^{1118}$, $N = 3$, $n = 54$). (I) Quantification of body length in (H) $Hml^\Delta$>GFP/$bmm^{RNAi}$ ($N = 3$, $n = 32$, $P < 0.0001$) in comparison to $Hml^\Delta$>GFP/TRiP (III) control (Ct.HSD, $N = 3$, $n = 82$). $Hml^\Delta$>GFP/UAS-bmm ($N = 3$, $n = 67$, $P < 0.0001$ in comparison to Ct.HSD Control, $Hml^\Delta$>GFP/$w^{1118}$, $N = 3$, $n = 52$). (J–L) Representative images of fly wings and (J'–L') adult females showing size phenotype on Ct.HSD from respective genetic backgrounds. Compared to (J, J') Ct.HSD Control ($Hml^\Delta$>GFP/$w^{1118}$), (K, K') knockdown of bmm ($Hml^\Delta$>GFP/$bmm^{RNAi}$) or (L, L') increase in its expression ($Hml^\Delta$>GFP/UAS-bmm) in immune cells causes either a recovery in adult fly size or a further reduction in size, respectively. (M) Quantification of female wing area in $Hml^\Delta$>GFP/$bmm^{RNAi}$ ($N = 3$, $n = 108$, $P < 0.0001$) in comparison to $Hml^\Delta$>GFP/TRiP (II) control (Ct.HSD, $N = 3$, $n = 121$), and $Hml^\Delta$>GFP/UAS-bmm ($N = 3$, $n = 85$, $P < 0.0001$) in comparison to HSD Control, ($Hml^\Delta$>GFP/$w^{1118}$, $N = 3$, $n = 132$). (N) Quantification of female body length in $Hml^\Delta$>GFP/$bmm^{RNAi}$ ($N = 3$, $n = 37$, $P < 0.0001$) in comparison to $Hml^\Delta$>GFP/TRiP (III) control (Ct.HSD, $N = 3$, $n = 89$), and $Hml^\Delta$>GFP/UAS-bmm ($N = 3$, $n = 93$, $P < 0.0001$) in comparison to HSD Control, ($Hml^\Delta$>GFP/$w^{1118}$, $N = 3$, $n = 47$). Data information: DNA is stained with DAPI (blue), immune cells are shown in green ($Hml^\Delta$>UAS-GFP). Nile red (red) staining to mark lipids in (B–D'). Scale bar: 5 µm for immune cells, 0.5 mm for flies and 0.25 mm for wings. In quantification (H, I, M, N), each dot represents an animal. Comparison for significance is with respective background control on HSD. Asterisks mark statistically significant differences (*$P < 0.05$; **$P < 0.01$; ***$P < 0.001$; ****$P < 0.0001$). The statistical analysis applied for (H, I, M, N) is Mann–Whitney test. $N$ indicates the number of independent biological replicates, and n refers to the total number of animals analyzed. Only right wing from each adult fly was selected for quantification. The differences in wing areas or fly body lengths in panels is indicated with a red dotted line or two horizontal red lines that highlight changes across genotypes. RF and Ct.HSD correspond to regular food and constitutive high sugar diet, respectively. Box plots show the median (center line), 25th–75th percentiles (bounds of box), and whiskers extending to the minimum and maximum values; all individual data points are shown. Source data are available online for this figure.

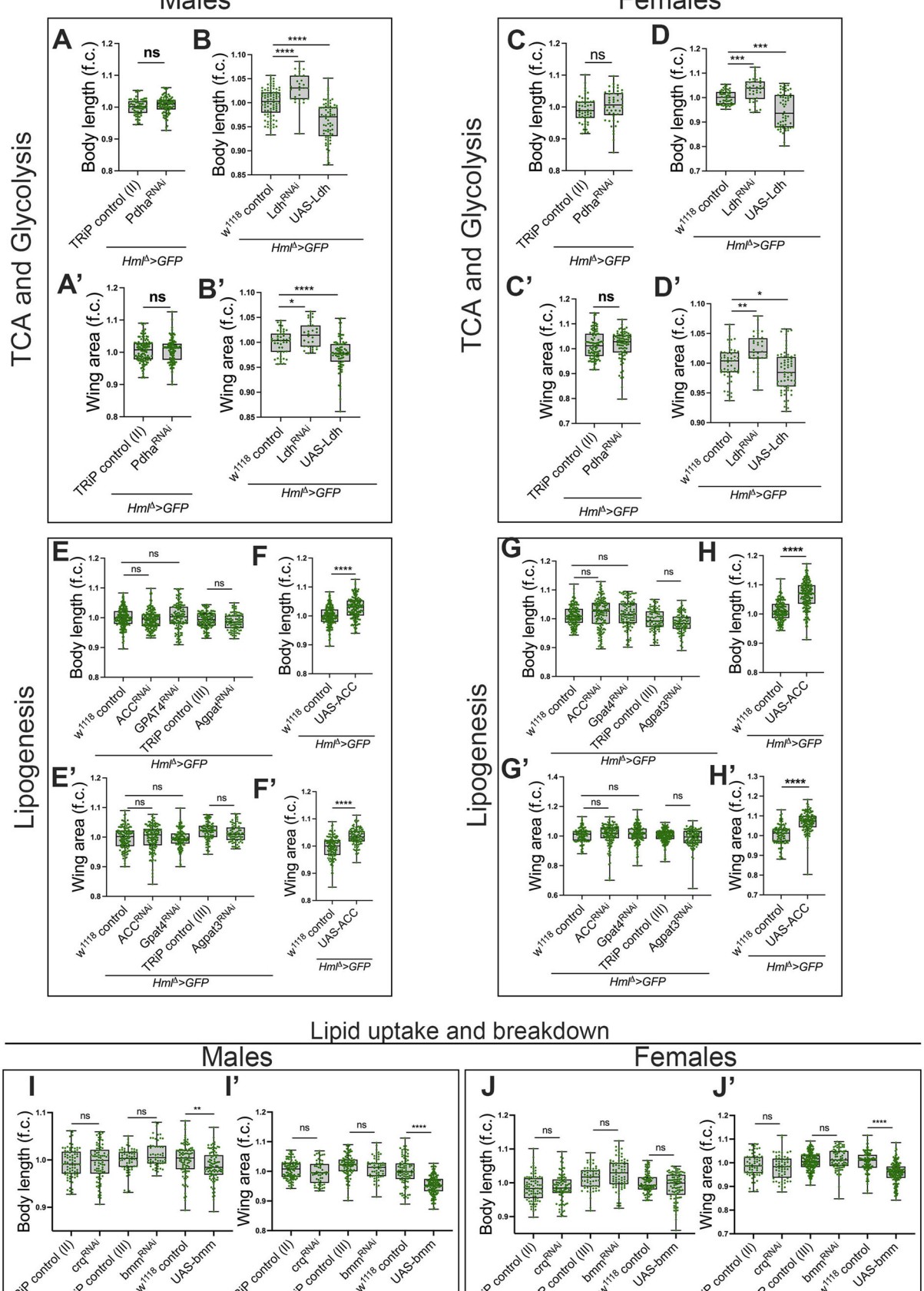

◀ **Figure EV6. Control of homeostatic growth by macrophage metabolic state changes.**

(A) Quantification of body length (RF, males) in $Hml^\Delta$>GFP/Pdha$^{RNAi}$ ($N = 3$, $n = 87$, $P = 0.0957$) in comparison to $Hml^\Delta$>GFP/TRiP (II) control ($N = 3$, $n = 62$). (A') Quantification of wing area (RF, males) in $Hml^\Delta$>GFP/Pdha$^{RNAi}$ ($N = 3$, $n = 108$, $P = 0.8986$) in comparison to $Hml^\Delta$>GFP/TRiP (II) control ($N = 3$, $n = 108$). (B) Quantification of body length (RF, males) in $Hml^\Delta$>GFP/Ldh$^{RNAi}$ ($N = 3$, $n = 29$, $P < 0.0001$) and $Hml^\Delta$>GFP/UAS-Ldh ($N = 3$, $n = 73$, $P < 0.0001$) in comparison to $Hml^\Delta$>GFP/$w^{1118}$ control ($N = 3$, $n = 81$). (B') Quantification of wing area (RF, males) in $Hml^\Delta$>GFP/Ldh$^{RNAi}$ ($N = 3$, $n = 28$, $P = 0.0413$) and $Hml^\Delta$>GFP/UAS-Ldh ($N = 3$, $n = 81$, $P < 0.0001$) in comparison to $Hml^\Delta$>GFP/$w^{1118}$ control ($N = 3$, $n = 47$). (C) Quantification of body length (RF, females) in $Hml^\Delta$>GFP/Pdha$^{RNAi}$ ($N = 3$, $n = 45$, $P = 0.0895$) in comparison to $Hml^\Delta$>GFP/TRiP (II) control ($N = 3$, $n = 52$). (C') Quantification of wing area (RF, females) in $Hml^\Delta$>GFP/Pdha$^{RNAi}$ ($N = 3$, $n = 82$, $P = 0.4170$) in comparison to $Hml^\Delta$>GFP/TRiP (II) control ($N = 3$, $n = 92$). (D) Quantification of body length (RF, females) in $Hml^\Delta$>GFP/Ldh$^{RNAi}$ ($N = 3$, $n = 38$, $P = 0.0004$) and $Hml^\Delta$>GFP/UAS-Ldh ($N = 3$, $n = 58$, $P = 0.0005$) in comparison to $Hml^\Delta$>GFP/$w^{1118}$ control ($N = 3$, $n = 51$). (D') Quantification of wing area (RF, females) in $Hml^\Delta$>GFP/Ldh$^{RNAi}$ ($N = 3$, $n = 34$, $P = 0.0014$) $Hml^\Delta$>GFP/UAS-Ldh ($N = 3$, $n = 64$, $P = 0.0188$) in comparison to $Hml^\Delta$>GFP/$w^{1118}$ control ($N = 3$, $n = 52$). (E) Quantification of body length (RF, males) in $Hml^\Delta$>GFP/ACC$^{RNAi}$ ($N = 3$, $n = 165$, $P = 0.0681$), $Hml^\Delta$>GFP/Gpat4$^{RNAi}$ ($N = 3$, $n = 119$, $P = 0.4515$) in comparison to RF control, $Hml^\Delta$>GFP/$w^{1118}$ ($N = 3$, $n = 188$). $Hml^\Delta$>GFP/Agpat3$^{RNAi}$ ($N = 3$, $n = 88$, $P = 0.0535$ in comparison to $Hml^\Delta$>GFP/TRiP (III) control ($N = 3$, $n = 116$). (E') Quantification of wing area (RF, males) in $Hml^\Delta$>GFP/ACC$^{RNAi}$ ($N = 3$, $n = 131$, $P = 0.4355$), $Hml^\Delta$>GFP/Gpat4$^{RNAi}$ ($N = 3$, $n = 114$, $P = 0.4514$) in comparison to RF control, $Hml^\Delta$>GFP/$w^{1118}$ ($N = 3$, $n = 126$). $Hml^\Delta$>GFP/Agpat3$^{RNAi}$ ($N = 3$, $n = 73$, $P = 0.053$) in comparison to $Hml^\Delta$>GFP/TRiP (III) control ($N = 3$, $n = 91$). (F) Quantification of body length (RF, males) in $Hml^\Delta$>GFP/UAS-ACC ($N = 3$, $n = 176$, $P < 0.0001$) in comparison to RF control, $Hml^\Delta$>GFP/$w^{1118}$ ($N = 3$, $n = 189$). (F') Quantification of wing area (RF, males) in $Hml^\Delta$>GFP/UAS-ACC ($N = 3$, $n = 109$, $P < 0.0001$) in comparison to RF control, $Hml^\Delta$>GFP/$w^{1118}$ ($N = 3$, $n = 127$). (G) Quantification of body length (RF, females) in $Hml^\Delta$>GFP/ACC$^{RNAi}$ ($N = 3$, $n = 141$, $P = 0.0549$) $Hml^\Delta$>GFP/Gpat4$^{RNAi}$ ($N = 3$, $n = 108$, $P = 0.1340$) in comparison to RF control, $Hml^\Delta$>GFP/$w^{1118}$ ($N = 3$, $n = 152$). $Hml^\Delta$>GFP/Agpat3$^{RNAi}$ ($N = 3$, $n = 90$, $P = 0.0555$ in comparison to $Hml^\Delta$>GFP/TRiP (III) control ($N = 3$, $n = 88$). (G') Quantification of wing area (RF, females) in $Hml^\Delta$>GFP/ACC$^{RNAi}$ ($N = 3$, $n = 119$, $P = 0.0536$), $Hml^\Delta$>GFP/Gpat4$^{RNAi}$ ($N = 3$, $n = 132$, $P = 0.2863$) in comparison to RF control, $Hml^\Delta$>GFP/$w^{1118}$ ($N = 3$, $n = 98$). $Hml^\Delta$>GFP/Agpat3$^{RNAi}$ ($N = 3$, $n = 102$, $P = 0.1034$) in comparison to $Hml^\Delta$>GFP/TRiP (III) control ($N = 3$, $n = 174$). (H) Quantification of body length (RF, females) in $Hml^\Delta$>GFP/UAS-ACC ($N = 3$, $n = 189$, $P < 0.0001$) in comparison to RF control, $Hml^\Delta$>GFP/$w^{1118}$ ($N = 3$, $n = 152$). (H') Quantification of wing area (RF, females) in $Hml^\Delta$>GFP/UAS-ACC ($N = 3$, $n = 126$, $P < 0.0001$) in comparison to RF control, $Hml^\Delta$>GFP/$w^{1118}$ ($N = 3$, $n = 98$). (I) Quantification of body length (RF, males) in $Hml^\Delta$>GFP/crq$^{RNAi}$ ($N = 3$, $n = 79$, $P = 0.6827$) in comparison to $Hml^\Delta$>GFP/TRiP (II) control (Ct.HSD, $N = 3$, $n = 78$). $Hml^\Delta$>GFP/bmm$^{RNAi}$ ($N = 3$, $n = 54$, $P = 0.1471$) in comparison to $Hml^\Delta$>GFP/TRiP (III) control (Ct.HSD, $N = 3$, $n = 60$). $Hml^\Delta$>GFP/UAS-bmm ($N = 3$, $n = 90$, $P = 0.0039$) in comparison to $Hml^\Delta$>GFP/$w^{1118}$ control (Ct.HSD, $N = 3$, $n = 90$). (I') Quantification of wing area (RF, males) in $Hml^\Delta$>GFP/crq$^{RNAi}$ ($N = 3$, $n = 56$, $P = 0.0852$) in comparison to $Hml^\Delta$>GFP/TRiP (II) control (Ct.HSD, $N = 3$, $n = 101$). $Hml^\Delta$>GFP/bmm$^{RNAi}$ ($N = 3$, $n = 58$, $P = 0.092$) in comparison to $Hml^\Delta$>GFP/TRiP (III) control (Ct.HSD, $N = 3$, $n = 97$). $Hml^\Delta$>GFP/UAS-bmm ($N = 3$, $n = 144$, $P < 0.0001$) in comparison to $Hml^\Delta$>GFP/$w^{1118}$ control (Ct.HSD, $N = 3$, $n = 90$). (J) Quantification of body length (RF, females) in $Hml^\Delta$>GFP/crq$^{RNAi}$ ($N = 3$, $n = 87$, $P = 0.6118$) in comparison to $Hml^\Delta$>GFP/TRiP (II) control (Ct.HSD, N $= 3$, $n = 82$). $Hml^\Delta$>GFP/bmm$^{RNAi}$ ($N = 3$, $n = 74$, $P = 0.0723$) in comparison to $Hml^\Delta$>GFP/TRiP (III) control (Ct.HSD, $N = 3$, $n = 59$). $Hml^\Delta$>GFP/UAS-bmm ($N = 3$, $n = 90$, $P = 0.2505$) in comparison to $Hml^\Delta$>GFP/$w^{1118}$ control (Ct.HSD, $N = 3$, $n = 90$). (J') Quantification of wing area (RF, females) in $Hml^\Delta$>GFP/crq$^{RNAi}$ ($N = 3$, $n = 60$, $P = 0.3552$) in comparison to $Hml^\Delta$>GFP/TRiP (II) control (Ct.HSD, $N = 3$, $n = 66$). $Hml^\Delta$>GFP/bmm$^{RNAi}$ ($N = 3$, $n = 77$, $P = 0.0938$) in comparison to $Hml^\Delta$>GFP/TRiP (III) control (Ct.HSD, $N = 3$, $n = 169$). $Hml^\Delta$>GFP/UAS-bmm ($N = 3$, $n = 155$, $P < 0.0001$) in comparison to $Hml^\Delta$>GFP/$w^{1118}$ control (Ct.HSD, $N = 3$, $n = 112$). Data information: In quantification graphs each dot represents an animal. Comparison for significance is with respective background controls. Asterisks mark statistically significant differences (*$P < 0.05$; **$P < 0.01$; ***$P < 0.001$; ****$P < 0.0001$). The statistical analysis applied for all panels (**A-J'**) is Mann–Whitney test. N indicates the number of independent biological replicates, and n refers to the total number of animals analyzed. RF is regular food. Box plots show the median (center line), 25th–75th percentiles (bounds of box), and whiskers extending to the minimum and maximum values; all individual data points are shown. Source data are available online for this figure.

