## [Peer Review File · EMBO Reports]

Macrophage metabolic reprogramming during dietary stress influences adult body size in *Drosophila*

Tina Mukherjee, Anusree Mahanta, Sajad Najar, Nivedita Hariharan, Ajit Bhowmick, Syed Rizvi, Manisha Goyal, Preethi Parupalli, Ramaswamy Subramanian, Angela Giangrande, and Dasaradhi Palakodeti

Corresponding author(s): Tina Mukherjee (tinam@instem.res.in), Anusree Mahanta (anusreem@instem.res.in)

Review Timeline:

Submission Date:	23rd Apr 24
Editorial Decision:	24th May 24
Appeal Received:	12th May 25
Editorial Decision:	2nd Jun 25
Revision Received:	26th Jul 25
Editorial Decision:	13th Aug 25
Revision Received:	21st Aug 25
Accepted:	25th Aug 25

Editor: Achim Breiling

Transaction Report:

Dear Dr. Mukherjee,

Thank you for the transfer of your manuscript to EMBO reports. I have now received the reports from the referees that were asked to evaluate your study, which can be found at the end of this email.

As you will see, although referee #1 is positive about the study, referee #2 and in particular referee #3 have major concerns, indicating that the functional characterization of the hemocytes remains superficial, and that conclusions drawn are not sufficiently supported by the data. Moreover, the referees note several technical and experimental shortcomings.

Given the comments of the referees, the amount of work required to address them, the fact that EMBO reports can only invite revision of papers that receive overall positive support from all the referees upon initial assessment, I cannot offer to publish your manuscript.

I am sorry to have to disappoint you this time. I nevertheless hope that the referee comments will be helpful in your continued work in this area, and I thank you once more for your interest in our journal.

Yours sincerely

Referee #1:

The manuscript presents a very interesting and innovative view of the function of the immune system in growth regulation. It is a very original and well-done work, which certainly deserves great attention. The data presented suggest a crucial but overlooked role of the immune system in the coordination of growth with metabolic stress conditions. Authors found that Macrophages in flies held on a high-sugar diet undergo a metabolic switch manifested mainly by enhanced lipogenesis and distinct metabolism of pyruvates and triacylglycerols. Enhanced lipogenesis in macrophages subsequently results in affected growth of the individual. The metabolic regime of macrophages themselves is under the control of NOTH and hedgehog signaling in this situation.

The authors present novel perspectives on the regulation of growth in metabolic stress. The quality of the whole work is in general on an exceptional level. It is experimentally well demonstrated that the effect of nutritional stress caused by a high-sugar diet on the growth of an individual is influenced by the metabolic state of macrophages. The work offers a broad-scale approach combining genetic screens, omics analyses, and classical tools of *Drosophila* experimental biology.

This work is the first to show a relationship between macrophage cellular metabolism and growth regulation, suggesting a generally valid mechanism by which metabolic stress or chronic inflammation can affect growth. The work highlights the important regulatory role of macrophages, which serve as metabolic sensors in the body.

I am confident that this work will find a wide scientific readership. This manuscript brings novel insights that may be further inspiring for research of infection or malnutrition impact on growth in insects, but also for broadening general conclusions of the work to mammalian experimental systems.

The whole work is written in a very understandable language, the text guides the reader through the individual experiments and the logic of the whole project is comprehensible. The conclusions are drawn according to the obtained data. Despite the detailed revision of the work, I have not identified any significant problems, major typos, or stylistic deficiencies.

I have just a few minor concerns that can be addressed in revision:

1) Data documents that HSD causes inhibition of plasmatocyte function and a decrease in their numbers. Are authors confident that cell numbers decline? In many cases, plasmatocytes stop displaying a GFP signal if they are overwhelmed with lipids, for instance in flies held on a high-fat diet.

Following this line, can authors comment on whether this treatment leads to enhanced differentiation of cells (transdifferentiation) towards lamellocytes? Have authors observed an enhanced number of these cells?

2) That is not completely clarified, where the lipids are coming from in flies with HSD? Data indicate that sugars are converted into lipids directly in macrophages, but also partially taken up via Crq.

Can some changes in lipid content be observed also in the fat body of flies held on HSD, is in FB enhanced lipolysis and mobilization of lipids?

I guess that neither of these possibilities is reflected in Fig. 6g which is a shame (the scheme, should be listed as 6g??? maybe

rather H).

Can authors from some of their observations indicate whether also other tissues in HSD enhanced lipid accumulation and potentially enhanced lipotoxicity?

Can there be some factors promoting the growth of individuals suggested in the scheme?

I found it a little bit confusing that "lipogenesis" (in green) is on the side of the control fly, however, lipogenesis itself is enhanced in HSD.

3) I found certain limits in the cited literature documenting M1/M2 polarization in *Drosophila* macrophages. I believe that there might be other publications better suited to this issue (for instance.... <https://doi.org/10.7554/eLife.50414>, <https://doi.org/10.1186/s12915-024-01858-5>).

Recently published work (<https://doi.org/10.1242/dev.202492>) indicates that macrophages play an essential role in lipid metabolism during metamorphosis. Some of the conclusions within this publication might be found interesting in the context of this work.

Summary of the review:

1. Does this manuscript report a single key finding? YES/NO

If YES, please describe it in one sentence.

YES - the manuscript documents the role of macrophages in sensing the metabolic status of the individuals influencing growth under metabolic stress conditions.

2. Is the reported work of significance (YES), or does it describe a confirmatory finding or one that has already been documented using other methods or in other organisms, etc (NO)?

YES

3. Is it of general interest to the molecular biology community

If YES, please say why, in a single sentence. If NO, please state which more specialized community you feel it is aimed at (or none), in a single word or phrase.

YES - although the work has been carried out on *Drosophila* as a model organism, the suggested mechanism may be generally valid and may be a significant inspiration for the whole immunometabolic community.

4. Is the single major finding robustly documented using independent lines of experimental evidence (YES), or is it really just a preliminary report requiring significant further data to become convincing, and thus more suited to a longer –format article (NO)?

YES

Referee #2:

In this study, Mahanta et al. describe the effect of dietary stress during development and its impact on organismal growth. They show that hemocytes act as metabolic sensors and participate in regulating body growth. The authors use a several -omics methods and genetics to characterize the transcriptional and metabolic state of hemocytes upon dietary stress.

While these approaches give a general overview of the status of hemocytes, the functional characterization remains superficial. In particular, it is hard to get a clear picture of what hemocytes are doing at the organismal level and how they are integrated in the known regulatory networks controlling organismal growth upon dietary stress. It is quite spectacular to observe that a 4hours dietary stress has such an impact on development. It would have been interesting to further explore this aspect and characterize the role of hemocytes here.

There are a lot of data presented in this study and it might worth considering splitting in different papers to further explore and clarify the physiological roles of hemocytes in this context. The focus on a specific regulatory pathway would be more appropriate.

Referee #3:

This manuscript describes an analysis of the role of macrophages in the control of animal growth in response to high sugar diet (HSD), using *Drosophila* as a model. Previous work from the lab (P et al, 2020) indicated that macrophage-like cells in the fly are involved in normal growth regulation and in controlling growth of flies under nutrient stress, namely high sugar diet (HSD).

The current study seeks to define the metabolic and other pathways involved in controlling the fly's macrophage response to HSD. In the interim, other groups have published additional studies suggesting a role for macrophages in growth control, involving insulin signalling (Juarez-Carreño S, Geissmann F., 2023) and mitochondrial metabolism (Sriskanthadevan-Pirahas S, et al., 2023), while Shin M, et al., 2020 argued a role for Hml- hemocytes. Mahanta et al. argue that macrophages respond to HSD by increasing glycolysis and TCA cycle activity (potentially growth inhibitory) and also increasing lipogenesis (potentially growth stimulatory), and the net result of this is reduced adult growth. To provide evidence for this, they (i) screen by knockdown for growth regulators expressed in macrophages, some of which have a metabolic role; (ii) undertake a transcriptomics analysis of macrophages under HSD versus controls and assess the effects of modulating some of the differentially expressed genes on adult growth under HSD conditions, leading to their final conclusions.

There is clearly a significant body of data in the manuscript. However, I do have major concerns about the quality and interpretation of some of those data, listed below.

Major concerns:

1. 'Genome-wide RNAi screen' for immune cell regulators (line 210): This is neither a genome-wide nor an unbiased screen. The authors have selected 1052 RNAi lines to screen without any clear or justified selection criteria. There are perhaps 13,500 genes in the genome, and although some of these may be non-macrophage-specific, I am sure there will be significantly more than 1000 genes expressed in these cells. Even if many of these genes are expressed at low levels, they may have a key function. The authors need to better explain in the main text how they selected these lines and remove the 'genome-wide' and 'unbiased' terminology.
2. The effects of RNAis, even the ones selected for detailed analysis, were only assessed under the HSD condition, not normal diet. There is therefore no test of whether these RNAis when expressed in macrophages have an effect on adult growth under all dietary conditions. My understanding is that the authors are screening for metabolic changes in macrophages that affect adult growth under HSD, but they do not check whether modulating the same genes under normal diet might have the same effects, so there is no selectivity to their assay - just screening for a change in mRNA levels does not inform us about the activities of the different enzymes they encode under HSD conditions. These control experiments need to be performed for each RNAi that is followed up, so that the results can be interpreted.
3. In this regard, and in light of their previous published data, one explanation for any changes could be that the knockdowns primarily alter the number of macrophages, rather than their activity and that this influences adult growth. The authors test four 'randomly selected' knockdowns (Fig. S2) to look at this, but they should test all the knockdowns with phenotypes, especially the ones that they follow up in the manuscript, to ascertain whether in some cases the effects could be explained by a simple change in cell number. Furthermore, knockdowns that reduce adult size via their effects on macrophages could do so for trivial reasons by making the macrophages non-viable. It doesn't appear that this is considered in the manuscript (although there are a couple of examples where knockdown and overexpression produce the opposite effects, which indicates there is some specificity to these specific changes; though again there is no analysis of Hml+ macrophage number).
4. One important point raised in Fig. S2 that makes the data very difficult to interpret is that at least in the case of the four knockdowns in Fig. S2, larval size during the entire period of HSD appears to be unchanged relative to HSD controls. It is not clear whether the larvae analysed in this figure are at early third instar or late stages, but if the former, it is also important to know what size the larvae have reached as they pupate. Is it third instar larval growth that is altered if the larvae are permitted to continue growing to adults (presumably on HSD), or probably less likely, is there a defect just in the organs, such as imaginal discs, that will form the adult structures? These seem critical questions, particularly since the timings would suggest that the knockdown effects on HSD-induced growth may be primarily happening after the transcriptional analysis of macrophages is performed in this study, despite the fact that the RNAi treatments are taking place through all larval instars when the authors suggest there is no obvious effect on larval growth. There needs to be much more clarity about when growth changes occur following knockdown, and both late third instar body and imaginal disc sizing to work out where the key problem lies. I found it somewhat difficult to work out from the text when the growth reductions induced by HSD in the absence of knockdown are taking place, but looking at Fig 1c, there seems to be a significant growth reduction by early third instar (is this about 4 hrs after moulting?) following Ct-HSD, which is during the time when the RNAi knockdowns presented in Fig. S2 did not have an effect. So how can this be consistent with these genes playing a role in those early HSD-induced growth changes?
5. I believe the control for RNAi knockdown experiments is a cross to a non-transgenic line. The authors should use another RNAi line, eg. luciferin, to at least activate the RNAi pathway in macrophages as a proper control.
6. I was concerned about how the data were being analysed and what some of the terms actually meant in the figure legends. As examples of the graphs in question (but there are several others; Fig S6f' reveals a reduction and an increase in different experiments, but the change is marked as significant), Fig. 1f, i and particularly l, where there is an N of three independent experiments (N = 3), each presumably involving multiple animals or groups of animals (n = between 30 and 105 total?). The individual mean values for each experiment (data points plotted) often look very different, with some of the means overlapping with controls. I think this might be expected because it is difficult to reproduce all the conditions for each HSD experiment consistently. But this means that the N used for the statistics calculation must be 3, because any individual values for specific flies are not independent of the experiment in which they were performed; if this N value is used, I cannot see how many of the results highlighted as significant can be significant with such a low N. Furthermore, given the variation in means, I doubt that an ANOVA is an appropriate test for the very low N. In some cases, the N is reduced to 2, eg. Fig. 1k, Fig. 4a (some overlaps with an N of 2 are still marked as significant) - it does not seem to be valid to undertake statistics with an N of 2, and this brings into question the basic metabolic data presented in the manuscript.

7. There are also some surprising results in the analysis. Both macrophage-specific hh and Smo knockdown induce recovery of growth defects in the HSD condition. Does this mean that these cells are being regulated by autocrine Hh signalling, which seems rather surprising? No comment is provided concerning this.

Minor comments:

1. I think it would be helpful for a native English speaker to go through the text, if submitting a revised version in the future. I found some of the wording difficult to interpret in places and the manuscript seemed over-long in terms of setting the scene for each dataset and explaining the data. Regarding clarity, for example, the abstract didn't really explain the experiments that had been undertaken in the manuscript, eg. did 'sugar-induced' mean 'induced by a high sugar diet'?, Did 'Stimulating pro-lipogenic immune state' refer to knockdown/overexpression experiments under conditions of HSD?
2. When the authors assess GO terms enriched in Ct.HSD and 4hr.HSD in macrophages and whole larvae, they do not indicate in the whole larvae whether the altered macrophage-specific GO terms flagged in Tables S2 and S3 are also altered in whole larvae; particularly for Ct.HSD, they just list GO terms that are altered, which have much lower P values. The effect (down-regulated or up-regulated) and P values for the equivalent GO terms identified in macrophages should be provided for whole larvae, so that the authors can demonstrate the specific rewiring described in the manuscript (line 396).

** As a service to authors, EMBO Press provides authors with the ability to transfer a manuscript that one journal cannot offer to publish to another journal, without the author having to upload the manuscript data again. To transfer your manuscript to another EMBO Press journal using this service, please click on Link Not Available

Institute for Stem Cell Science and Regenerative Medicine

Tina Mukherjee
Associate Investigator

inStem

12th May, 2025

Dear Dr. Breiling,

We have now revised our manuscript titled “**Dietary stress induced macrophage metabolic reprogramming, a determinant of animal growth**” originally submitted to The EMBO Reports with the previous manuscript number **EMBOR-2024-59471-T**.

It is substantially revised and we are indeed grateful to you for the opportunity to resubmit the revised version. It has taken us considerable amount of time, because it was a lot of work that needed to be done and in the revised version, we made a strong effort to address all major concerns raised.

We thank the reviewers for their constructive critique of our work and are also very grateful for the appreciation received mentioning this body of work as the first to show a specific correlation between macrophage metabolic states and adult growth regulation. Their feedback has significantly contributed to strengthening our manuscript. In this revised submission, we have incorporated substantial new datasets and analyses:

Major changes

- We have included **new data on homeostatic growth** (Fig. S9), revealing that even under a regular diet, immune glycolysis suppresses adult growth, and that its downregulation leads to increased organismal size.
- We have conducted an **extensive analysis of immune cell numbers** across all genotypes on HSD (Fig. S10), demonstrating that changes in immune cell state, rather than cell number, critically drive growth outcomes.
- We have **characterized developmental growth dynamics**, showing that growth regulation correlates specifically with larval imaginal disc size, and not overall larval body size (Fig. S10) uncovering a novel immune–imaginal disc cross-talk.
- We have **incorporated appropriate RNAi controls** (Fig. 4-6 and S5-10) and have expanded sample sizes in key experiments (e.g., Fig. 1f, i, k–n; Fig. 4a) to further strengthen statistical robustness.
- **Improved our statistical analysis** and with a greater number of data sets, at least 70-100 animals analyzed for the growth phenotypes, across all genetics conditions.
- We present a **revised model** (Fig. 6i) that integrates the new experimental findings that provides a more comprehensive understanding of the phenomenon.

Minor changes

- We have **updated the manuscript with additional references** and included Supplementary Table 4 comparing GO term enrichments between macrophage and whole-larva datasets.
- **Removed the Notch and hedgehog data**, as suggested by reviewer 2

- Extensively **re-written the manuscript** to shorten, and for improved clarity

From all the work done, we have the following key conceptual advances that further emerge and were previously missing from our study, these include:

- Identification of the immune glycolytic state as a central suppressor of organismal growth, operating under homeostatic condition. High-sugar stress conditions, further elevates this state, leading to a more stronger growth oppressive condition.
- HSD, also invokes lipolysis, together with glycolysis, this leads to an overall growth-retarding condition.
- Discovery that a lipogenic metabolic switch in immune cells functions as a growth-supportive adaptation, not operational in homeostatic growth. However, its induction remains insufficient to fully counteract the dominant growth-inhibitory effects of glycolysis and lipolysis under HSD.
- A metabolic imbalance between growth promoting (lipogenic) and growth inhibiting (lipolysis, glycolysis) on HSD, leads to early growth impairment of imaginal disc sizing that skews adult growth.
- Unveiling of a previously unrecognized developmental cross-talk between immune metabolic states and imaginal disc growth, providing a mechanistic insight into immune cells influence on organ-specific development and control of their developmental plasticity.
- Finally, the sufficiency of immune-metabolic changes, as opposed to changes in immune cell numbers to manifest a control in governing animal size is extraordinary.

In summary, these new results have strengthened the core findings of the manuscript significantly. From highlighting general immune metabolic states (such as the glycolytic index) controlling homeostatic growth to specific immune metabolic transitions that accompany as a consequence of exposure to dietary high sugar induced stress that in a context specific manner control growth emerge as the foremost understanding from these data. This has allowed us to systematically delineate the growth paradigm in HSD and unveil a completely novel insight into growth homeostasis through the lenses of immune-metabolic regulation. The extent to which this axis can moderate growth plasticity has been an explicitly intriguing finding from our revisions.

The data also points to a conjecture where crosstalk between immune metabolic state changes and larval imaginal disc sizing is evident. This specific immune-imaginal disc cross-talk is indeed unexpected. While the larger multi-organ cross-talk may also engage the fat body as we do see changes in fat body cell areas, the directness of the cross talk remains to be determined and will be taken on board as our future endeavors. Overall, the findings present a comprehensive immune-metabolic centric view of animal growth control, which to the best of our understanding is the first of its kind. The deterministic effect of immune metabolic state changes on growth modulation and its plasticity is indeed striking. The findings open up a new modality of growth governance via immune cells that has been overlooked.

Finally, the revised manuscript that has now work done over 9 months which involves a graduate student, a post doctorate fellow and project interns, with 28 new panels added with extensively revised data (Fig. S9 a, a', b, b', c, c', d, d', e, e', f, f', g, g', h, h', i, i', j, j', S10 v, v', v'', v''', w, w', w'', w''', Fig. 1f, i, k, l, m, n, Fig. 4a', g, h, Fig. 5e, p, q, Fig.6h, h', Fig.S5e, f, Fig.S6a1, a2, a3, a4, a5, a6, a7, l, m, Fig.S7c, h, h', k, k', Fig. S8g, g'), now offers a significantly enhanced and comprehensive view of immune control of developmental outcomes. We are confident that EMBO Reports will find these findings of broad interest, and given the emerging relevance of immunometabolism across diverse biological contexts will review the revised work favorably.

We respectfully submit the revised manuscript for your consideration, and we remain fully available to address any further queries or revisions if needed.

Thank you very much for your time, effort, and cooperation.

Dr. Tina Mukherjee
Associate Investigator,
Senior Research Fellow, DBT-Wellcome India Alliance
Institute for Stem Cell Science and Regenerative Medicine,
GKVK Post, Bellary Road,
Bangalore - 560065
Ph: 91-80-23666016 (o), 91-9620789846 (c)
email: tinam@instem.res.in
<https://www.instem.res.in/tina-mukherjee>

POINT BY POINT RESPONSE TO REVIEWER

We sincerely thank all the reviewers for their detailed and constructive critique of our work. In response, we have substantially revised the draft to address all concerns raised. The revised manuscript is now much more cohesive in its overall concept, with enhanced clarity regarding growth control, and includes additional new data on homeostasis. We have also expanded our investigation of larval imaginal discs to better understand growth dynamics, incorporated RNAi controls and blood cell counts, and improved the overall rigor of the study. As a result, the quality of the work is significantly superior to the previous version. We are deeply grateful to the reviewers for their insightful comments, which have greatly strengthened our manuscript.

Below, we provide a summary of all the changes made in the revised draft, followed by our detailed, point-by-point response to each comment. We hope that the improvements made will render the revised manuscript acceptable for publication.

Referee1:

The manuscript presents a very interesting and innovative view of the function of the immune system in growth regulation. It is a very original and well-done work, which certainly deserves great attention. The data presented suggest a crucial but overlooked role of the immune system in the coordination of growth with metabolic stress conditions. Authors found that Macrophages in flies held on a high-sugar diet undergo a metabolic switch manifested mainly by enhanced lipogenesis and distinct metabolism of pyruvates and triacylglycerols. Enhanced lipogenesis in macrophages subsequently results in affected growth of the individual. The metabolic regime of macrophages themselves is under the control of NOTH and hedgehog signaling in this situation.

The authors present novel perspectives on the regulation of growth in metabolic stress. The quality of the whole work is in general on an exceptional level. It is experimentally well demonstrated that the effect of nutritional stress caused by a high-sugar diet on the growth of an individual is influenced by the metabolic state of macrophages. The work offers a broad-scale approach combining genetic screens, omics analyses, and classical tools of *Drosophila* experimental biology.

This work is the first to show a relationship between macrophage cellular metabolism and growth regulation, suggesting a generally valid mechanism by which metabolic stress or chronic inflammation can affect growth. The work highlights the important regulatory role of macrophages, which serve as metabolic sensors in the body.

I am confident that this work will find a wide scientific readership. This manuscript brings novel insights that may be further inspiring for research of infection or malnutrition impact on growth in insects, but also for broadening general conclusions of the work to mammalian experimental systems.

The whole work is written in a very understandable language, the text guides the reader through the individual experiments and the logic of the whole project is comprehensible. The conclusions are drawn according to the obtained data. Despite the detailed revision of the work, I have not identified any significant problems, major typos, or stylistic deficiencies.

Author response: We sincerely thank reviewer for the encouraging remarks and appreciation of our work. We also thank the reviewer for the helpful insights and supportive critique which have been very helpful in revising our manuscript. Please find our response following the concern raised and the necessary change made to address each query made.

I have just a few minor concerns that can be addressed in revision:

1) Data documents that HSD causes inhibition of plasmatocyte function and a decrease in their numbers. Are authors confident that cell numbers decline? In many cases, plasmatocytes stop displaying a GFP signal if they are overwhelmed with lipids, for instance in flies held on a high-fat diet.

Author response: We acknowledge this point and exactly for the reasons stated with respect to GFP signal and its dampening, we have quantified DAPI positive cells to estimate total blood cell counts. The total counts represented in our plots include: HmlGFP⁺DAPI⁺ and HmlGFP⁻DAPI⁺ cells. Using this approach, we have counted all the blood cells and quantified total blood cell counts per mm².

The counting analyses have been done carefully making sure that Hml cells with low, moderate to high GFP signals typically observed in both regular food and high sugar animals were accounted for as HmlGFP⁺DAPI⁺ cells while absence to any GFP was counted as Hml-GFP negative cells. The images for cell counting experiment have been obtained at a 20X magnification to cover maximum fields. We have also imaged immune cells at 60X magnification with additional optical zoom (Fig 1g-j'' and Fig S1d-f'') which further enabled us to clearly observe cells with similar intensities of GFP signals in both the dietary regimes that are always counterstained with DAPI. Specifically, as seen in Figure 1g'' and Fig S1d'', even though immune cells from high sugar fed animals were loaded with excessive lipids, there was no effect on GFP signals in these cells. Rather, cells with varying GFP intensities irrespective of lipid content were observed across regular food and high sugar treated immune cells. Thus, we have used DAPI positive counter-stained cells to estimate total cells and then interpreted this data with respect to presence of GFP expression. This approach has been used in our past publications (P et al, 2020; Madhwal et al, 2020), and the methods section mentions this in detail now (See methods Pg 42).

Following this line, can authors comment on whether this treatment leads to enhanced differentiation of cells (transdifferentiation) towards lamellocytes? Have authors observed an enhanced number of these cells?

Author response: This is really an interesting comment, we do not observe lamellocytes in HSD. We have performed phalloidin stainings in immune cells to visualize these cells, considering that lamellocytes have a distinct large morphology. However, we did not observe any differentiation of cells towards lamellocytes. Our total blood cell counts also do not reveal any enhancement in their numbers with high sugar treatment (Fig. S1f-f'). We have added a comment in the results section referring to this point (Pg 10).

2) That is not completely clarified, where the lipids are coming from in flies with HSD? Data indicate that sugars are converted into lipids directly in macrophages, but also partially taken up via Crq.

Can some changes in lipid content be observed also in the fat body of flies held on HSD, is in FB enhanced lipolysis and mobilization of lipids?

Author response: We thank the reviewer for bringing this point. It is indeed correctly stated by the reviewer that the intracellular lipid content in the blood cells is derived from two pathways. One is induction of lipogenesis, both de novo and TAG-synthesis, which may be driven by sugar induced pathways. The other arm is clearly the systemic lipid uptake via lipid scavenging receptor, croquemort(crq). For the later, the source is systemic circulating lipids. In this regard, the hemolymph lipids is what the blood cells are internalizing, which we speculate is majorly derived from the fat body.

Our previous work from the lab (P et al, 2020) has demonstrated that HSD fed larvae have increased circulating TAGs and also elevated TAGs in the fat body. The accumulation of lipids in the fat body on HSD has been shown to be a consequence of a lipogenic induction enabled to provide a protective effect to the animal against the deleterious effects of high sugar diet (Musselman et al, 2011).

We therefore hypothesize that the immune cells are responding to the increased lipids released from the fat body as circulating free fatty acids. We have now included this possibility in the model (Fig 6i).

I guess that neither of these possibilities is reflected in Fig. 6g which is a shame (the scheme, should be listed as 6g??? maybe rather H).

Author response: We apologize for this error in labelling the model and have now correctly labelled it as Fig 6i.

Can authors from some of their observations indicate whether also other tissues in HSD enhanced lipid accumulation and potentially enhanced lipotoxicity?

Author response: Yes, we do observe increased lipid accumulation in adipose (P et al, 2020) and non-adipose tissues like larval imaginal discs and brain in HSD condition. We are currently following these observations, however at the current moment we are unable to comment on enhanced lipotoxicity.

Nevertheless, we would like to state the negative influence of immune-derived free fatty acids on growth. We show that raising lipolysis, via over-expression of bmm in blood cells, this leads to growth retarded flies regardless of dietary state (Fig. S9j-k and Fig. 6g, g'). Contrarily, blocking immune cell lipid breakdown leads to growth increase on HSD condition (Fig 6 g, g'). This alludes to immune catabolic event of lipolysis as a negative regulator of growth and places the immune-derived free fatty acids as drivers of lipotoxicity. This is a speculation that we provide as of now and in the revised draft we have added these new data on bmm (Fig. S9j-k) and also discussed to highlight the lipotoxic effects of FFA on systemic growth (Pg 33).

Can there be some factors promoting the growth of individuals suggested in the scheme?

Author response: We have revised the model to add this element. Our genetic screen has alluded to secreted ligands like upd2 cytokine and CCHamide hormone being involved as positive regulators of adult growth on HSD. This is based on the observation that knock-down of these ligands specifically in the immune cells resulted in further adult growth retardation on HSD indicative of their growth-promoting roles. As suggested, we have now included the signaling role of these factors in the scheme/model (Fig 6i). We thank the reviewer for suggesting this addition as it justifies a lot of our screen findings. This is discussed in Pgs 38, 39.

I found it a little bit confusing that "lipogenesis" (in green) is on the side of the control fly, however, lipogenesis itself is enhanced in HSD.

Author response: We have now restructured the model substantially to address concerns with respect to clarity (Fig. 6i). We thank the reviewer for raising this point.

3) I found certain limits in the cited literature documenting M1/M2 polarization in *Drosophila* macrophages. I believe that there might be other publications better suited to this issue (for instance.. <https://doi.org/10.7554/eLife.50414>, <https://doi.org/10.1186/s12915-024-01858-5>).

Recently published work (<https://doi.org/10.1242/dev.202492>) indicates that macrophages play an essential role in lipid metabolism during metamorphosis. Some of the conclusions within this publication might be found interesting in the context of this work.

Author response: We thank the reviewer for suggesting the relevant literature. We have now included the references related to M1/M2 macrophage polarization in *Drosophila* in the introduction section (Pg 5). We have now included the conclusions from the mentioned manuscript in our discussion.

Summary of the review:

1. Does this manuscript report a single key finding? YES/NO

If YES, please describe it in one sentence.

YES - the manuscript documents the role of macrophages in sensing the metabolic status of the individual, influencing growth under metabolic stress conditions.

2. Is the reported work of significance (YES), or does it describe a confirmatory finding or one that has already been documented using other methods or in other organisms, etc (NO)?

YES

3. Is it of general interest to the molecular biology community

If YES, please say why, in a single sentence. If NO, please state which more specialized community you feel it is aimed at (or none), in a single word or phrase.

YES - although the work has been carried out on *Drosophila* as a model organism, the suggested mechanism may be generally valid and may be a significant inspiration for the whole immuno-metabolic community.

4. Is the single major finding robustly documented using independent lines of experimental evidence (YES), or is it really just a preliminary report requiring significant further data to become convincing, and thus more suited to a longer format article (NO)?

YES

Referee #2:

In this study, Mahanta et al. describe the effect of dietary stress during development and its impact on organismal growth. They show that hemocytes act as metabolic sensors and participate in regulating body growth. The authors use a several -omics methods and genetics to characterize the transcriptional and metabolic state of hemocytes upon dietary stress.

While these approaches give a general overview of the status of hemocytes, the functional characterization remains superficial. In particular, it is hard to get a clear picture of what hemocytes are doing at the organismal level and how they are integrated in the known regulatory networks controlling organismal growth upon dietary stress. It is quite spectacular to observe that a 4hours dietary stress has such an impact on development. It would have been interesting to further explore this aspect and characterize the role of hemocytes here.

There are a lot of data presented in this study and it might worth considering splitting in different papers to further explore and clarify the physiological roles of hemocytes in this context. The focus on a specific regulatory pathway would be more appropriate.

Author response: *We appreciate the reviewer's insightful comments and fully acknowledge the concern that the manuscript currently offers a broad overview of hemocyte-mediated control of systemic growth. However, we respectfully note that this study represents the first comprehensive investigation into the role of immune cells—specifically hemocytes—as regulators of organismal growth through metabolic modulation.*

As suggested by the reviewer about splitting in different papers, we have now completely removed the hedgehog and Notch data in the revised manuscript and shortened it significantly.

In the revised manuscript, we have significantly expanded our metabolism dataset and refined our core observations by incorporating new experimental evidence. This includes detailed metabolic profiling under homeostatic (regular food; RF) conditions (Fig. S9), addition of appropriate background controls, quantitative analysis of larval (Fig. S10 a-g) and imaginal disc (Fig. S10 h-n) growth, and a broader exploration of metabolic transitions within immune cells. These enhancements have allowed us to build a more rigorous and mechanistic framework linking immune cell metabolism to growth outcomes.

A key insight from our study is the unexpected observation that even under homeostatic conditions, immune cells exert a suppressive influence on systemic growth via their metabolic state. Specifically, hemocyte-specific knockdown of Ldh under RF conditions results in significantly larger adults (Fig. S9 b,b' and d,d'). This demonstrates that glycolytic metabolism within hemocytes restricts organismal growth even in the absence of dietary stress, positioning immune cells as active modulators of baseline growth.

Furthermore, our data show that dietary stress induced by high-sugar diet (HSD) robustly reprograms immune metabolism, enhancing glycolysis, oxidative phosphorylation, and lipogenesis (Fig. 3,4). While these changes likely reflect adaptive responses to metabolic stress, our results indicate that not all metabolic transitions impact systemic growth equally. Notably, increased glycolysis via Ldh in immune cells is a dominant suppressor of growth under HSD (Fig. 4e, e'), while lipogenesis appears to play a compensatory (Fig. 5m, m')—yet insufficient—growth-promoting role. Thus, the growth restriction observed under HSD reflects a net outcome of these opposing metabolic influences.

Importantly, we now provide evidence that immune cell metabolism also influences the size of imaginal discs independently of larval size (Fig. S10 b-d and v). For instance, bmm knockdown results in larger imaginal discs without altering overall larval dimensions, suggesting that immune-derived signals directly or indirectly modulate disc

growth (Fig. S10 d, k and v, v'). Additionally, we observe associated changes in fat body cell size (Fig. S10 o-u and v'', w''), suggesting a broader systemic crosstalk, possibly involving endocrine or paracrine mediators such as hh, upd2, Wg, as evident from our screen. While we cannot conclusively determine whether there is a direct interaction between immune cells and imaginal discs, this remains an exciting avenue for future research.

In summary, the revised manuscript presents new data and a strengthened conceptual model illustrating how immune cells, via metabolic plasticity, influence systemic growth trajectories. We believe this work represents a foundational step in defining immune-metabolic integration during development. The antagonistic effect of immune metabolic states on growth is confounding which unveils the unexpected plasticity during growth enabled through changes in immune-cell metabolism. This study to the best of our knowledge and understanding the first to position immune cells in a comprehensive manner in the growth context. Although, we understand the enthusiasm for the cross-talk, the precise nature of immune-tissue interactions warrants further exploration, which is best suited as an independent study and is currently ongoing in the lab. We believe the current study on its own lays the groundwork for such investigations and should be viewed as the first in a series aimed at unravelling this emerging paradigm. I hope that in our attempts to substantially revise the manuscript, the revised draft will be of significant interest. We thank the reviewer for the enthusiasm of our work.

Referee #3:

This manuscript describes an analysis of the role of macrophages in the control of animal growth in response to high sugar diet (HSD), using *Drosophila* as a model. Previous work from the lab (P et al, 2020) indicated that macrophage-like cells in the fly are involved in normal growth regulation and in controlling growth of flies under nutrient stress, namely high sugar diet (HSD).

The current study seeks to define the metabolic and other pathways involved in controlling the fly's macrophage response to HSD. In the interim, other groups have published additional studies suggesting a role for macrophages in growth control, involving insulin signalling (Juarez-Carreño S, Geissmann F., 2023) and mitochondrial metabolism (Sriskanthadevan-Pirahas S, et al., 2023), while Shin M, et al., 2020 argued a role for Hml- hemocytes. Mahanta et al. argue that macrophages respond to HSD by increasing glycolysis and TCA cycle activity (potentially growth inhibitory) and also increasing lipogenesis (potentially growth stimulatory), and the net result of this is reduced adult growth. To provide evidence for this, they (i) screen by knockdown for growth regulators expressed in macrophages, some of which have a metabolic role; (ii) undertake a transcriptomics analysis of macrophages under HSD versus controls and assess the effects of modulating some of the differentially expressed genes on adult growth under HSD conditions, leading to their final conclusions.

There is clearly a significant body of data in the manuscript. However, I do have major concerns about the quality and interpretation of some of those data, listed below.

Author response: We thank reviewer 3 for the detailed review and constructive feedback on our manuscript which have helped identify areas that needed improvement. The overall comments have been incredibly valuable in strengthening our work and has indeed conceptualized our core findings comprehensively. We have addressed each concern raised to

the best of our understanding and made a sincere attempt to thoroughly revise the manuscript. Our point-by-point response lists the changes made against each comment here with.

Major concerns:

1. 'Genome-wide RNAi screen' for immune cell regulators (line 210): This is neither a genome-wide nor an unbiased screen. The authors have selected 1052 RNAi lines to screen without any clear or justified selection criteria. There are perhaps 13,500 genes in the genome, and although some of these may be non-macrophage-specific, I am sure there will be significantly more than 1000 genes expressed in these cells. Even if many of these genes are expressed at low levels, they may have a key function. The authors need to better explain in the main text how they selected these lines and remove the 'genome-wide' and 'unbiased' terminology.

Author Response: We agree with the reviewers' suggestion and we have now removed the terms "genome-wide" and "unbiased" from the text. It was an oversight on our behalf considering the number of RNAi lines tested.

The basis for selecting the 1052 RNAi lines for our genetic screen was solely based on the availability of RNAi lines housed at the NCBS fly facility. The NCBS stock center at Bangalore, is India's largest national fly stock center and houses more than 7000 fly lines. Importantly, majority of these RNAi lines from the facility have already been used in multiple studies at NCBS and thus validated (Agrawal et al, 2013; Mishra et al, 2024; Janardan et al, 2020) prior to our work. The RNAi collection houses VDRC lines obtained from laboratories outside our campus. Hence the collection is diverse, which is evident in the nature of the candidates identified in our screen, ranging from transcription factors, cell signaling receptors, metabolism, cytoskeletal proteins and secreted ligands. We have now clarified the collection which is mentioned in the results (Pg 10) and in the methods section of the manuscript (Pgs 42,43).

2. The effects of RNAis, even the ones selected for detailed analysis, were only assessed under the HSD condition, not normal diet. There is therefore no test of whether these RNAis when expressed in macrophages have an effect on adult growth under all dietary conditions. My understanding is that the authors are screening for metabolic changes in macrophages that affect adult growth under HSD, but they do not check whether modulating the same genes under normal diet might have the same effects, so there is no selectivity to their assay - just screening for a change in mRNA levels does not inform us about the activities of the different enzymes they encode under HSD conditions. These control experiments need to be performed for each RNAi that is followed up, so that the results can be interpreted.

Author response: We thank the reviewer for this very insightful suggestion. The effects of all the genotypes on adult growth in the context of homeostasis which is regular food diet has now been done and added in the revised manuscript (Fig. S9). This has made a significant difference to the way we now interpret our data. We observe that immune glycolytic state is a repressor of homeostatic growth as well (Fig. S9b, b' and S9d, d'). Contrastingly, the lipid pathway genes, their down-regulation failed to affect growth in homeostatic condition (Fig. S9e, e' and S9g, g'). Indicating that the lipogenic changes seen in HSD are adaptive in nature, corresponding to high sugar induced dietary stress.

Specifically, macrophage-specific down-regulation of de-novo lipogenesis (Hml^Δ>GFP/ACC^{RNAi}), TAG synthesis (Hml^Δ>GFP/GPAT4^{RNAi} and Hml^Δ>GFP/Agpat3^{RNAi}), lipolysis (Hml^Δ>GFP/bmm^{RNAi}) affected growth in the context of high-sugar diet (Fig. 5) and not in RF (Fig. S9e, e', i, i' and S9g, g', j, j'). Thus, indicating that these metabolic pathways are specific modulators of growth in this form of over-nutrition. However, what we found intriguing was that the forced induction of lipogenesis in regular diet via ACC gain-of-function, was sufficient to increase animal size over and above the homeostatic state (Fig. S9 f,f' and h,h'). This indicated that promoting a lipogenic state within the immune cells irrespective of the dietary stress, provided the animal with an overall growth promoting outcome. However, in the conditions of HSD, the gain of lipogenesis even though it is induced it is insufficient to exert a strong growth advantage.

In this regard, the sufficiency of gain of Ldh (Fig. S9 b,b' and d,d') or lipolysis via bmm over-expression (Fig. S9 i,i' and j,j') on growth retardation is evident in RF conditions, where these genetic states have led to smaller sized animals. Thus, highlighting the sufficiency of elevated immune- glycolytic activity and increased lipid breakdown in immune cells, independent of the dietary input as a prominent driver for animal growth retardation.

Taking into consideration both the data, finally we conclude that compared to immune cells on RF, HSD immune cells have a further heightened glycolytic state (Fig. 4a') and demonstrate a specific induction of a lipogenic state. In addition to this immune cells also maintain an active lipolytic state (Fig. S6 a7). While the induction of immune lipogenic state is an adaptive change induced to support growth in dietary stress however, the elevated immune glycolytic state and the maintenance of an active lipolytic state operate dominantly. We speculate that the immune lipogenic induction in these cells is insufficient to counter the negative effects of glycolytic and lipolytic activity, which is why the animals are small. The imbalance between these antagonizing states underlies the growth retardation seen in HSD. Any genetic means to reset the imbalance either when ACC is over-expressed in immune cells or bmm is down-regulated or when Ldh is down-regulated in HSD immune cells, are sufficient to recover animal sizes on HSD. Thus, alluding to imbalance between growth promoting (lipogenic) versus growth repressing (glycolysis, lipolysis) immune metabolic states on HSD that cause smaller sized adults. These findings are now included (HSD (Fig. 4, 5, 6, Fig. S9 and Pg 27).

Collectively, the data obtained from RF and HSD screening, has allowed us to conclude on immune metabolic state and their control of animal growth specifically, and then interpret it based on the diet. We thank the reviewer for this very important comment.

3. In this regard, and in light of their previous published data, one explanation for any changes could be that the knockdowns primarily alter the number of macrophages, rather than their activity and that this influences adult growth. The authors test four 'randomly selected' knockdowns (Fig. S2) to look at this, but they should test all the knockdowns with phenotypes, especially the ones that they follow up in the manuscript, to ascertain whether in some cases the effects could be explained by a simple change in cell number. Furthermore, knockdowns that reduce adult size via their effects on macrophages could do so for trivial reasons by making the macrophages non-viable. It doesn't appear that this is considered in the manuscript (although there are a couple of examples where knockdown and overexpression produce the opposite effects, which indicates there is some specificity to these specific changes; though again there is no analysis of Hml+ macrophage number).

Author Response: We thank the reviewer for raising this point. We would like to point out our past work (P et al, 2020) where we identified immune-cell state as opposed to their number as a key determinant of growth paradigm. This work set the precedence for the screen, which the current study has undertaken. We have now assayed immune cell numbers for all genotypes relevant to the study, as suggested by the reviewer.

Doing this, we failed to identify any correlation between immune cell numbers and animal size regulation. In most conditions, we did not observe any change in immune cell numbers (Fig. S10v''', w'''). Specifically, gain of ACC expression was as its over-expression in immune cells which is associated with larger animals, showed fewer immune cells in them (Fig. S10 w'''). Even LdhRNAi condition, that leads to bigger animals, no change in blood cell numbers was evident (Fig. S10 w'''). These data are in alignment with our previous observations where the internal state changes and activity of immune cells emerges as key determinants of adult body size regulation as opposed to just their numbers. These findings are now presented in detail with figure panels showing bleeds from all conditions and included in results section (Fig. S9, Pg 30).

4. One important point raised in Fig. S2 that makes the data very difficult to interpret is that at least in the case of the four knockdowns in Fig. S2, larval size during the entire period of HSD appears to be unchanged relative to HSD controls. It is not clear whether the larvae analysed in this figure are at early third instar or late stages, but if the former, it is also important to know what size the larvae have reached as they pupate. Is it third instar larval growth that is altered if the larvae are permitted to continue growing to adults (presumably on HSD), or probably less likely, is there a defect just in the organs, such as imaginal discs, that will form the adult structures? These seem critical questions, particularly since the timings would suggest that the knockdown effects on HSD-induced growth may be primarily happening after the transcriptional analysis of macrophages is performed in this study, despite the fact that the RNAi treatments are taking place through all larval instars when the authors suggest there is no obvious effect on larval growth. There needs to be much more clarity about when growth changes occur following knockdown, and both late third instar body and imaginal disc sizing to work out where the key problem lies.

I found it somewhat difficult to work out from the text when the growth reductions induced by HSD in the absence of knockdown are taking place, but looking at Fig 1c, there seems to be a significant growth reduction by early third instar (is this about 4 hrs after moulting?) following Ct-HSD, which is during the time when the RNAi knockdowns presented in Fig. S2 did not have an effect. So how can this be consistent with these genes playing a role in those early HSD-induced growth changes?

Author Response: A very thoughtful comment and we have addressed this. Accordingly, as suggested we analyzed the late wandering 3rd instar larval body lengths for the genotypes that impacted adult growth in high sugar conditions. We found that, compared to HSD controls, changes in larval body lengths was observed for only half of the genetic conditions and not for all conditions (Fig S10 a-g and v,w). As rightly pointed out by the reviewer, this observation necessitated further investigation into changes that might be seen in the growth of larval epithelial structures that give rise to adult structures or more specifically the imaginal discs which might help explain any corresponding changes in adult body growth under HSD. Indeed, we found that immune manipulations that affected adult growth on HSD showed striking and consistent changes in respective larval wing disc area measurements (Fig S10 h-n and v',w') across all genotypes. Even though changes in larval

size wasn't consistently observed, the changes in wing imaginal disc development was evident across all genotypes. Indicating a specific immune-imaginal disc cross-talk that coordinated adult sizing. Additionally, we characterized larval fat body cell areas and we observed changes in it as well, which implied an organismal level cross-talk initiated by immune metabolic changes (Fig S10 o-u and v'', w'').

We would like to speculate a specific crosstalk that operates between immune cell metabolic states and larval imaginal discs, that finally determines the adult size to be achieved. The directness of this cross-talk however remains to be discerned and these new data are included in results section (Fig.S10 Pg 29).

5. I believe the control for RNAi knockdown experiments is a cross to a non-transgenic line. The authors should use another RNAi line, eg. luciferin, to at least activate the RNAi pathway in macrophages as a proper control.

Author Response: We have now added respective background RNAi controls for both 2nd and 3rd chromosome. These have been used for statistical analysis across all data sets. We mention this in the methods specifically (Pg 50) and thank the reviewer for this suggestion.

6. I was concerned about how the data were being analysed and what some of the terms actually meant in the figure legends. As examples of the graphs in question (but there are several others; Fig S6f reveals a reduction and an increase in different experiments, but the change is marked as significant), Fig. 1f, i and particularly l, where there is an N of three independent experiments (N = 3), each presumably involving multiple animals or groups of animals (n = between 30 and 105 total?). The individual mean values for each experiment (data points plotted) often look very different, with some of the means overlapping with controls. I think this might be expected because it is difficult to reproduce all the conditions for each HSD experiment consistently. But this means that the N used for the statistics calculation must be 3, because any individual values for specific flies are not independent of the experiment in which they were performed; if this N value is used, I cannot see how many of the results highlighted as significant can be significant with such a low N. Furthermore, given the variation in means, I doubt that an ANOVA is an appropriate test for the very low N. In some cases, the N is reduced to 2, eg. Fig. 1k, Fig. 4a (some overlaps with an N of 2 are still marked as significant) - it does not seem to be valid to undertake statistics with an N of 2, and this brings into question the basic metabolic data presented in the manuscript.

Author Response: We acknowledge the overall comment made and indeed the statistical analysis were not rightly done. The use of ANOVA as pointed out by the reviewer is not the accurate choice.

We have now redone the stats and have added more repeats. Now with more than 3 experimental repeats, in all cases (N=5 at least, for Fig 1 and N=3-4 batches for all the adult fly and larval data sets) we have redone the stats to compare the mean changes or their medians. We have referred the following papers (Texada et al, 2019; Kubrak et al, 2022) where similar data sets have been compared, and have used the same statistical method to consider significance. The changes in animal sizes in Figures 4, 5, 6 and Sup. Fig 5,6,7,8, larval sizes, Wing disc size, fat body cell area Fig. S10 are now represented as fold changes against their respective controls and have been analysed using Mann Whitney test, for Fig.1 Unpaired t-test and Welch's t-test for pairwise comparison, Fig.1d and S1 a'', b''

Two-Way Anova in Graphpad Prism 10 software. The use of these statistical methods are in alignment with the degree of change that is noticed.

Although, some data sets may still seem to have overlapping means closer to controls, but now with the addition of more experimental batches, the differences are better evident and the statistical analysis utilized takes into consideration these new data points. The methods section mentions the analysis in detail (Pg 50).

Even with the new stats performed, the data overall holds its significance and the major conclusions remain unchanged. The data are rather more-stronger and thoroughly done, and we are satisfied with the revisions, we sincerely thank the reviewer for the same.

7. There are also some surprising results in the analysis. Both macrophage-specific hh and Smo knockdown induce recovery of growth defects in the HSD condition. Does this mean that these cells are being regulated by autocrine Hh signalling, which seems rather surprising? No comment is provided concerning this.

Author Response: We agree with the reviewer that finding both hh and Smo knockdown to induce growth recovery is surprising. Our genetic findings would argue for an autocrine mode of activation which is worth exploring which is supported by recent by Yin et.al., 2022, where autocrine Hh signaling becomes essential for controlling proliferation and differentiation of tracheal progenitor cells. Specifically, depletion of hh or Smo in epithelial cells affected their proliferation and differentiation implying an autocrine mode of signaling. Thus, this is possible in our case.

Alternatively, as part of our ongoing studies, we have performed single cell analysis of HSD immune cells with the Hml positive population. Compared to homeostasis, changes in immune cell clusters is evident in HSD, and we do find a few immune clusters that are ligand producing. This analysis is however still in progress, but it is conceivable to propose clusters of hh producing cells that signal to control Smo signaling in a paracrine manner whose outcome is control of systemic growth in HSD. This is entirely speculative as of now, but we hope in the future, as our analysis becomes more clearer, we will be able to reveal the nature of such ligand producing and signaling cross-talk.

We have now added a comment in the results section on this finding (Page no. 13).

Minor comments:

1. I think it would be helpful for a native English speaker to go through the text, if submitting a revised version in the future. I found some of the wording difficult to interpret in places and the manuscript seemed over-long in terms of setting the scene for each dataset and explaining the data. Regarding clarity, for example, the abstract didn't really explain the experiments that had been undertaken in the manuscript, eg. did 'sugar-induced' mean 'induced by a high sugar diet'?, Did 'Stimulating pro-lipogenic immune state' refer to knockdown/overexpression experiments under conditions of HSD?

Author Response: We have now made extensive changes to the draft in terms of its writing and overall description of the experimental results. Also, the suggested changes in the abstract to describe the experiments have been made. We had previously kept it succinct to be

within the word limit, but now provide a much more descriptive abstract. The revised draft is significantly shortened and describes the metabolic data across dietary regimes of regular food and HSD. We have removed the details on Notch and hedgehog that was provided in the previous version. While we feel those data added value to the context of lipogenesis, but should the data be needed, we will be happy to provide it back.

2. When the authors assess GO terms enriched in Ct.HSD and 4hr.HSD in macrophages and whole larvae, they do not indicate in the whole larvae whether the altered macrophage-specific GO terms flagged in Tables S2 and S3 are also altered in whole larvae; particularly for Ct.HSD, they just list GO terms that are altered, which have much lower P values. The effect (down-regulated or up-regulated) and P values for the equivalent GO terms identified in macrophages should be provided for whole larvae, so that the authors can demonstrate the specific rewiring described in the manuscript (line 396).

Author Response: We have now provided a table (Table S4) comparing the common GO terms (up or down regulated along with the p values) enriched for both macrophage and whole larvae. To avoid confusion, we have removed the previous tables depicting GO terms for whole larvae only (Table S4 and S5). The new table clearly indicates macrophage-specific metabolic rewiring that either promote or repress growth while showing an overall metabolic dampening in the context of whole larvae on HSD.

References

- Agrawal T, Sadaf S & Hasan G (2013) A genetic RNAi screen for IP₃/Ca²⁺ coupled GPCRs in *Drosophila* identifies the PdfR as a regulator of insect flight. *PLoS Genet* 9: e1003849
- Janardan V, Sharma S, Basu U & Raghu P (2020) A Genetic Screen in *Drosophila* To Identify Novel Regulation of Cell Growth by Phosphoinositide Signaling. *G3 (Bethesda)* 10: 57–67
- Kubrak O, Koyama T, Ahrentlöv N, Jensen L, Malita A, Naseem MT, Lassen M, Nagy S, Texada MJ, Halberg KV, *et al* (2022) The gut hormone Allatostatin C/Somatostatin regulates food intake and metabolic homeostasis under nutrient stress. *Nat Commun* 13: 692
- Madhwal S, Shin M, Kapoor A, Goyal M, Joshi MK, Ur Rehman PM, Gor K, Shim J & Mukherjee T (2020) Metabolic control of cellular immune-competency by odors in *Drosophila*. *eLife* 9: e60376
- Mishra S, Manohar V, Chandel S, Manoj T, Bhattacharya S, Hegde N, Nath VR, Krishnan H, Wendling C, Di Mattia T, *et al* (2024) A genetic screen to uncover mechanisms underlying lipid transfer protein function at membrane contact sites. *Life Sci Alliance* 7: e202302525
- Musselman LP, Fink JL, Narzinski K, Ramachandran PV, Hathiramani SS, Cagan RL & Baranski TJ (2011) A high-sugar diet produces obesity and insulin resistance in wild-type *Drosophila*. *Dis Model Mech* 4: 842–849

P P, Tomar A, Madhwal S & Mukherjee T (2020) Immune Control of Animal Growth in Homeostasis and Nutritional Stress in *Drosophila*. *Frontiers in Immunology* 11

Texada MJ, Jørgensen AF, Christensen CF, Koyama T, Malita A, Smith DK, Marple DFM, Danielsen ET, Petersen SK, Hansen JL, *et al* (2019) A fat-tissue sensor couples growth to oxygen availability by remotely controlling insulin secretion. *Nat Commun* 10: 1955

Dear Dr. Mukherjee,

Thank you for the re-submission of your revised manuscript to EMBO reports. I have now received the reports from the two referees that were asked to re-evaluate your study, which can be found at the end of this email.

As you will see, original referee #1 is satisfied by the revision and supports publication. Referee #3 acknowledges that the revision has greatly improved the study, but has remaining concerns and suggestions to improve the study further.

I would thus like to invite you to revise your manuscript further with the understanding that the remaining concerns of referee #3 must be addressed in the revised manuscript and in a further point-by-point response.

When submitting your further revised manuscript, we will require:

1) a .docx formatted version of the final manuscript text (including legends for main figures, EV figures and tables), but without the figures included. Figure legends should be compiled at the end of the manuscript text.

2) individual production quality figure files as .eps, .tif, .jpg (one file per figure), of main figures and EV figures. Please upload these as separate, individual files upon re-submission.

The Expanded View format, which will be displayed in the main HTML of the paper in a collapsible format, has replaced the Supplementary information. You can submit up to 6 images as Expanded View. Please follow the nomenclature Figure EV1, Figure EV2 etc. The figure legend for these should be included in the main manuscript document file in a section called Expanded View Figure Legends after the main Figure Legends section. Additional Supplementary material should be supplied as a single pdf file labeled Appendix. The Appendix should have page numbers and needs to include a table of content on the first page (with page numbers) and legends for all content. Please follow the nomenclature Appendix Figure Sx, Appendix Table Sx etc. throughout the text, and also label the figures and tables according to this nomenclature.

3) a .docx formatted letter INCLUDING the reviewers' reports on the revision and your detailed point-by-point responses to their remaining concerns. As part of the EMBO Press transparent editorial process, the point-by-point response is part of the Review Process File (RPF), which will be published alongside your paper.

4) a complete author checklist, which you can download from our author guidelines

(<https://www.embopress.org/page/journal/14693178/authorguide>). Please insert page numbers in the checklist to indicate where the requested information can be found in the manuscript. The completed author checklist will also be part of the RPF.

5) that primary datasets produced in this study (e.g. RNA-seq, ChIP-seq, structural and array data) are deposited in an appropriate public database. If no primary datasets have been deposited, please also state this in a dedicated section (e.g. 'No primary datasets have been generated and deposited'), see below.

The accession numbers and database should be listed in a formal "Data Availability" section that follows the model below. This is now mandatory (like the COI statement). Please note that the Data Availability Section is restricted to new primary data that are part of this study. This section is mandatory. As indicated above, if no primary datasets have been deposited, please state this in this section

Data availability

6) We now request the publication of original source data with the aim of making primary data more accessible and transparent to the reader. You will receive a separate email with instructions for providing source data with your revised manuscript, including information how to upload and organize the files.

8) Regarding data quantification and statistics, please make sure that the number "n" for how many independent experiments were performed, their nature (biological versus technical replicates), the bars and error bars (e.g. SEM, SD) and the test used to calculate p-values is indicated in the respective figure legends (also for EV and Appendix figures). Please also check that all the p-values are explained in the legend, and that these fit to those shown in the figure. Please provide statistical testing where applicable. Please avoid the phrase 'independent experiment', but clearly state if these were biological or technical replicates. Please also indicate (e.g. with n.s.) if testing was performed, but the differences are not significant. In case n=2, please show the data as separate datapoints without error bars and statistics. See also: <http://www.embopress.org/page/journal/14693178/authorguide#statisticalanalysis>

9) Please add scale bars of similar style and thickness to microscopic images, using clearly visible black or white bars (depending on the background). Please place these in the lower right corner of the images themselves. Please do not write on or near the bars in the image but define the size in the respective figure legend.

10) Please also note our reference format:

12) We now use CRediT to specify the contributions of each author in the journal submission system. CRediT replaces the author contribution section. Please use the free text box to provide more detailed descriptions and do NOT provide your final manuscript text file with an author contributions section. See also our guide to authors: <https://www.embopress.org/page/journal/14693178/authorguide#authorshippinguidelines>

13) All Materials and Methods need to be described in the main text using our 'Structured Methods' format, which is required for all research articles. According to this format, the Methods section should include a Reagents and Tools Table (listing key reagents, experimental models, software, and relevant equipment and including their sources and relevant identifiers), uploaded as separate file, and a Methods section in which we encourage the authors to describe their methods using a step-by-step protocol format with bullet points, to facilitate the adoption of the methodologies across labs. More information on how to adhere to this format as well as downloadable templates (.doc) for the Reagents and Tools Table can be found in our author guidelines (section 'Structured Methods'):

14) Please order the sections like this, using these names:

Title page - Abstract - Keywords - Introduction - Results - Discussion - Methods - Data availability section - Acknowledgements (please include here also the funding information) - Disclosure and Competing Interests Statement - References - Figure legends - Expanded View Figure legends

15) Please make sure that all the funding information is also entered into the online submission system and that it is complete and similar to the one in the acknowledgement section of the manuscript text file.

16) Please have your final manuscript text file carefully proofread by a native speaker.

I look forward to seeing a revised form of your manuscript when it is ready.

Yours sincerely,

Referee #1:

The authors of the manuscript have convincingly addressed all of my concerns and comments. In my opinion, the manuscript is very interesting and thoroughly elaborated. During the review process, the authors significantly expanded on many details, and the work is now of a very high standard.

Referee #3:

Mahanta et al have produced a substantially revised version of their manuscript, responding to my comments on the previous version and for the most part, introducing changes that have improved the quality of the manuscript. This is particularly true for point 2 (assessing the effects of RNAis under normal dietary conditions) and point 4 (effects on imaginal disc growth), where the findings have provided clarification and some mechanistic insights regarding two outstanding questions in their original manuscript. I believe that the additional experiments, added controls and better statistical analysis that they have also included have significantly improved the robustness of the study.

I have a number of points, which are primarily related to my original comments, which I believe still need to be addressed.

1. Supp. Figs. S9 and S10 now present important data concerning the specificity of immune cell metabolic response to HFD diet and the underlying developmental growth defects in imaginal discs - the latter features in the Abstract. I think these figures, or at the very least, Supp. Fig. S10, should be promoted to main figures.

2. In my opinion, there are still quite significant issues with the text in the manuscript, which will make it difficult to follow for many readers (highlighted in minor point 1 of original review). As suggested previously, the manuscript really needs to be copy-edited by a native English speaker to correct grammatical errors, organise the content of the sections better and compress the text. A few examples (though there are many others, including several incomplete sentences, throughout the manuscript) are:

a. Title: 'stress-induced', and more normally, titles of research articles are sentences. I.e. 'Dietary stress-induced macrophage metabolic reprogramming regulates animal growth'. The current title seems more like a review title.

b. The abstract includes far too many inappropriately hyphenated words (though the authors did not hyphenate 'dietary stress-induced') with several undefined abbreviations - were the authors trying to reduce the word count?

c. Each Results section starts with a description of the approach and methods, much of which could move to the Methods section. These sections then end with a quite detailed discussion of the implications of the results, a lot of which would be better placed in the discussion. For me, the discussion is too speculative - I have no issue in drawing parallels with other systems, but this seems too extensive, at the expense of considering the data in the manuscript (which currently are discussed in the Results). Overall, the manuscript seems too long for the amount of data presented and it will not be straightforward to read for anyone who is dipping into it rather than reading it in detail.

d. The text of the figure legends is again very long, starting with Data information, prior to describing the panels. I don't think this fits with the recommended style for EMBO Reports and I found it difficult to rapidly interpret what was being presented, because the explanation was buried in the middle of the legend.

3. In Supp Fig. S1c', the image provided seems to show much higher ROS in Ct.HSD contrary to the bar chart shown in Figure 1k. Is this data set represented in the bar chart?

Overall, I think this study is much improved and does make a valuable contribution to the field, presenting some interesting and novel findings, but it does require further attention to make it accessible to the readership of EMBO Reports.

POINT BY POINT RESPONSE TO REVIEWER

We sincerely appreciate the editors for giving us the opportunity to further refine our manuscript, and we are thankful to both reviewers for their insightful and constructive feedback. We are grateful for Referee #1's positive evaluation and clear support for publication. We also thank Referee #3 for recognizing the significant improvements made in our revision and for offering thoughtful suggestions to further strengthen the manuscript's clarity and presentation.

In response to Referee #3's remaining points, we have thoroughly revised the manuscript once again.

Referee #3:

Mahanta et al have produced a substantially revised version of their manuscript, responding to my comments on the previous version and for the most part, introducing changes that have improved the quality of the manuscript. This is particularly true for point 2 (assessing the effects of RNAis under normal dietary conditions) and point 4 (effects on imaginal disc growth), where the findings have provided clarification and some mechanistic insights regarding two outstanding questions in their original manuscript. I believe that the additional experiments, added controls and better statistical analysis that they have also included have significantly improved the robustness of the study.

Author Response:

We are grateful to Referee #3 for the thoughtful follow-up comments and have carefully revised the manuscript to address each remaining concern, as detailed below.

I have a number of points, which are primarily related to my original comments, which I believe still need to be addressed.

1. Supp. Figs. S9 and S10 now present important data concerning the specificity of immune cell metabolic response to HFD diet and the underlying developmental growth defects in imaginal discs - the latter features in the Abstract. I think these figures, or at the very least, Supp. Fig. S10, should be promoted to main figures.

Author Response:

In line with this comment, we have moved Supp. Fig. S10 to the main figure Fig. 6, to better highlight these important findings. The figure has been renumbered accordingly, and all relevant text and references have been updated in the manuscript.

2. In my opinion, there are still quite significant issues with the text in the manuscript, which will make it difficult to follow for many readers (highlighted in minor point 1 of original review). As suggested previously, the manuscript really needs to be copy-edited by a native English speaker to correct grammatical errors, organise the content of the sections better and compress the text. A few examples (though there are many others, including several incomplete sentences, throughout the manuscript) are:

a. Title: 'stress-induced', and more normally, titles of research articles are sentences. I.e. 'Dietary stress-induced macrophage metabolic reprogramming regulates animal growth'. The current title seems more like a review title.

Author Response: We appreciate the reviewer's feedback. In response, we have revised the title to better reflect the main findings and to align with the conventions of research article titles. The new title is: "*Dietary stress-induced macrophage metabolic reprogramming regulates organ growth and adult body size in Drosophila.*"

Additionally, the manuscript has undergone substantial language editing to improve grammar, readability, and overall structure, while preserving the scientific content.

b. The abstract includes far too many inappropriately hyphenated words (though the authors did not hyphenate 'dietary stress-induced') with several undefined abbreviations - were the authors trying to reduce the word count?

Author Response: We apologize for the many hyphenated words. We have now fully rewritten the abstract, removed unnecessary hyphenations, clarified all abbreviations, and ensured the text reads more clearly.

c. Each Results section starts with a description of the approach and methods, much of which could move to the Methods section. These sections then end with a quite detailed discussion of the implications of the results, a lot of which would be better placed in the discussion. For me, the discussion is too speculative - I have no issue in drawing parallels with other systems, but this seems too extensive, at the expense of considering the data in the manuscript (which currently are discussed in the Results). Overall, the manuscript seems too long for the amount of data presented and it will not be straightforward to read for anyone who is dipping into it rather than reading it in detail.

Author Response: We thank the reviewer for this valuable comment. We have revised the Results sections to minimize detailed methodological descriptions, which have now been moved to the Methods section where appropriate. We have also shifted interpretive and speculative statements to the Discussion to better distinguish between the presentation of data and its interpretation, thereby improving the overall clarity and structure of the manuscript. We have substantially shortened the Discussion to reduce speculative content and focus more directly on the data presented. We believe these changes have improved the overall readability and balance of the manuscript.

d. The text of the figure legends is again very long, starting with Data information, prior to describing the panels. I don't think this fits with the recommended style for EMBO Reports and I found it difficult to rapidly interpret what was being presented, because the explanation was buried in the middle of the legend.

Author Response: We have shortened and restructured the legend to maintain the style of EMBO reports. They start by describing each panel clearly and succinctly, followed by the data representation.

3. In Supp Fig. S1c', the image provided seems to show much higher ROS in Ct.HSD contrary to the bar chart shown in Figure 1k. Is this data set represented in the bar chart?

Author Response: We acknowledge the issue and have now replaced the image in Supp. Fig. S1c' with one that accurately matches the data shown in the bar chart in Figure 1k.

Overall, I think this study is much improved and does make a valuable contribution to the field, presenting some interesting and novel findings, but it does require further attention to make it accessible to the readership of EMBO Reports.

Author Response: We thank the reviewer for the support of our work and for providing the necessary and constructive critique and inputs. With the introduction of all the necessary and relevant changes, the current version over two rounds of revisions has improved significantly in its scientific rigor, presentation and overall quality and will be of considerable interest to the broad readership of EMBO Reports.

Dear Dr. Mukherjee,

Thank you for the submission of your further revised manuscript to our editorial offices. I now went through this and you final p-b-p-response and I consider the remaining points of referee #3 as adequately addressed.

Before we can proceed with formal acceptance, I have these editorial requests I ask you to address in a final revised manuscript:

- I know you have adjusted the title upon request of referee #3. Nevertheless, we need a final title with not more than 100 characters including spaces. Please provide one.
- Please reduce the number of keywords to five and order the manuscript sections like this, using these names: Title page - Abstract - Keywords - Introduction - Results - Discussion - Methods - Data availability section - Acknowledgements - Disclosure and Competing Interests Statement - References - Figure legends - Expanded View Figure legends
- There are name discrepancies. It is Ajit Bhowmick in the manuscript text file, but Ajith Bhowmick in the submission system, and Preethi P in the manuscript, but Preethi Parupalli in the submission system. Please check and make sure that the same names are used.
- Please prepare the figure legends according to our format (capital letters - also in the figures):

- Please make sure that all figure panels (main and EV figures) are called out separately and sequentially. Presently, there seem to be no separate callouts for Fig. 1K, 1L, 1M, 1N, 4B and 6R. Please check.
- Please check again that the number "n" for how many independent experiments were performed, their nature (biological versus technical replicates), the bars and error bars (e.g. SEM, SD) and the test used to calculate p-values is indicated in the respective figure legends (main and EV figures). Please also check that all the p-values are explained in the legend, and that these fit to those shown in the figure. Please provide statistical testing where applicable. Please avoid the phrase 'independent experiment', but clearly state if these were biological or technical replicates. Please also indicate (e.g. with n.s.) if testing was performed, but the differences are not significant. In case n=2, please show the data as separate datapoints without error bars and statistics. See also:
<http://www.embopress.org/page/journal/14693178/authorguide#statisticalanalysis>

If n<5, please show single datapoints for diagrams. Moreover:

- Please define the annotated p values ****/***/**/* as well as provide the exact p-values for the same in the legend of figure S2 A as appropriate.
- Please note that the exact p values are not provided in the legends of figures 1D, F, I, L, M, N; 4A', G, H; 5E, P, Q; 6V'-W'; EV1 A', B'; EV3 E, F; EV4 A1, A2, A5-A7; EV5 H, I, M, N; EV6 B, B', D, D', F, F', H, H'; S3 C, H, H', K, K'
- Please indicate the statistical test used for data analysis in the legends of figure S2 A
- Please note that the box plots need to be defined in terms of minima, maxima, centre, bounds of box and whiskers, and percentile in the legends of figures 4G, H; 5P, Q; 6V-V' W-W'; EV3 E, F; EV4 L, M; EV5 H, I, M, N; EV6 A-H'; S3 H, H', K, K'
- Please note that the error bars are not defined in the legends of figures 1D, F, I, K, L, M, N; 4A', 5E, 6V', W'; EV1 A', B'; EV4A1-A7; S2A, B; S3C
- All Materials and Methods need to be described in the main text using our 'Structured Methods' format, which is required for all research articles. According to this format, the Methods section should include a Reagents and Tools Table (listing key reagents, experimental models, software, and relevant equipment and including their sources and relevant identifiers), uploaded as separate file, and a Methods section in which we encourage the authors to describe their methods using a step-by-step protocol format with bullet points, to facilitate the adoption of the methodologies across labs. More information on how to adhere to this format as well as downloadable templates (.doc) for the Reagents and Tools Table can be found in our author guidelines (section 'Structured Methods'):

- Please move the information shown in Tables 5 and 6 to the Reagents and Tools table and update the callouts. Moreover, please add callouts to the reagents and tools table in the methods section where applicable.
- Please add the Tables 1, 2, 3, 4, and 7 to the Appendix. Please use the name Appendix Table Sx and update any callouts. Finally, please remove the file with the tables from the manuscript files.
- Please remove the author list from the Appendix title page. It is sufficient to state 'Appendix for' followed by the final title of the

paper. Moreover, please add page numbers to the Appendix file.

- BioRender and other software should be acknowledged in the Methods section in the following way:

Graphics:

(some of the... OR Figure #... OR synopsis) Graphics were created with BioRender.com. Then please remove their mention from the Acknowledgements.

- Please make sure that all the funding information is also entered into the online submission system and that it is complete and similar to the one in the acknowledgement section of the manuscript text file. Presently, the grants 'USIAS Indo French grant, Council of Scientific & Industrial Research (CSIR)-Fellowship, DST INSPIRE' are missing in the submission system. Please check.

- During our internal figure integrity checks, we noted the reuse of three images:

Fig. 1e' seems to be part of EV1c'; Fig. 1e' seems to be part of EV1c'; Fig. 1j seems to be part of EV1f (but has been rotated). Please check. In case this reuse is intentional (the experimental conditions seem to be the same) please indicate this clearly on the figure legends and mark the part of the image shown in higher magnification in Fig. 1 with a box in Fig. EV1. And explain this in the legend of EV1. In case there are more similar reuses we did not detect, please treat these in a similar way.

In addition, I would need from you uploaded separately:

- a short, two-sentence summary of the manuscript (not more than 35 words).

- two to four short (!) bullet points highlighting the key findings of your study (two lines each).

- a schematic summary figure as separate file that provides a sketch of the major findings (not a data image) in jpeg or tiff format (with the exact width of 550 pixels and a height of not more than 400 pixels) that can be used as a visual synopsis on our website.

Best,

The authors have addressed all minor editorial requests.

Dr. Tina Mukherjee
Institute for Stem Cell Science and Regenerative Medicine
Regulation Of cell fate
GKVK Campus
Bellary Road
Bangalore, Karnataka 560065
India

Dear Dr. Mukherjee,

I am very pleased to accept your manuscript for publication in the next available issue of EMBO reports. Thank you for your contribution to our journal.

Yours sincerely,
